

**Principal component analysis of summertime ground site measurements in the Athabasca oil sands:**
**Sources of IVOCs**
Travis W. Tokarek[1], Charles A. Odame-Ankrah[1], Jennifer A. Huo[1], Robert McLaren[2], Alex K. Y. Lee[3, 4],
Max G. Adam[4], Megan D. Willis[5], Jonathan P. D. Abbatt[5], Cristian Mihele[6], Andrea Darlington[6],
Richard L. Mittermeier[6], Kevin Strawbridge[6], Katherine L. Hayden[6], Jason S. Olfert[7], Elijah. G. Schnitzler[8],
Duncan K. Brownsey[1], Faisal V. Assad[1], Gregory R. Wentworth[5, a], Alex G. Tevlin[5], Douglas E. J. Worthy[6],
Shao-Meng Li[6], John Liggio[6], Jeffrey R. Brook[6], and Hans D. Osthoff[1*]
[1] Department of Chemistry, University of Calgary, Calgary, Alberta, T2N 1N4, Canada
[2] Centre for Atmospheric Chemistry, York University, Toronto, Ontario, M3J 1P3, Canada
[3] Department of Civil and Environmental Engineering, National University of Singapore, Singapore
117576, Singapore
[4] NUS Environmental Research Institute, National University of Singapore, Singapore
[5] Department of Chemistry, University of Toronto, Toronto, Ontario, M5S 3H6, Canada
[6] Air Quality Research Division, Environment and Climate Change Canada, Toronto, Ontario, M3H 5T4,
Canada
[7] Department of Mechanical Engineering, University of Alberta, Edmonton, Alberta, T6G 1H9, Canada
[8] Department of Chemistry, University of Alberta, Edmonton, Alberta, T6G 2G2, Canada
[a] Now at: Environmental Monitoring and Science Division, Alberta Environment and Parks, Edmonton,
Alberta, T5J 5C6, Canada
* Corresponding author



**Abstract**
In this paper, measurements of air pollutants made at a ground site near Fort McKay in the Athabasca
oil sands region as part of a multi-platform campaign in the summer of 2013 are presented. The
observations included measurements of selected volatile organic compounds (VOCs) by a gas
chromatograph – ion trap mass spectrometer (GC-ITMS). This instrument observed a large, analytically
unresolved hydrocarbon peak (with retention index between 1100 and 1700) associated with
intermediate volatility organic compounds (IVOCs). However, the activities or processes that contribute
to the release of these IVOCs in the oil sands region remain unclear.
Principal component analysis (PCA) with Varimax rotation was applied to elucidate major source types
impacting the sampling site in the summer of 2013. The analysis included 28 variables, including
concentrations of total odd nitrogen ($NO_y$), carbon dioxide ($CO_2$), methane ($CH_4$), ammonia ($NH_3$), carbon
monoxide (CO), sulfur dioxide ($SO_2$), total reduced sulfur compounds (TRS), speciated monoterpenes
(including α- and β-pinene and limonene), particle volume calculated from measured size distributions
of particles less than 10 μm and 1 μm in diameter ($PM_{10-1}$ and $PM_1$), particle-surface bound polycyclic
aromatic hydrocarbons (pPAH), and aerosol mass spectrometer composition measurements, including
refractory black carbon (rBC) and organic aerosol components. The PCA was complemented by bivariate
polar plots showing the joint wind speed and direction dependence of air pollutant concentrations to
illustrate the spatial distribution of sources in the area. Using the 95% cumulative percentage of
variance criterion, ten components were identified and categorized by source type. These included
emissions by wet tailings ponds, vegetation, open pit mining operations, upgrader facilities, and surface
dust. Three components correlated with IVOCs, with the largest associated with surface mining and is
likely caused by the unearthing and processing of raw bitumen.



## 1. Introduction

The Athabasca oil sands region of Northern Alberta, Canada, has seen extraordinary expansion of its oil

sands production and processing facilities (CAPP, 2016) and associated emissions of air pollutants over

the last several decades (Englander et al., 2013; Bari and Kindzierski, 2015). Air emissions from these

facilities have been impacting surrounding communities, including the city of Ft. McMurray and the

community of Ft. McKay (WBEA, 2013). To assess the impact of these emissions on human health,

visibility and climate, and the ecosystems downwind, it is critical to obtain an understanding of the

source types from all activities associated with oil sands operations (ECCC, 2016).

Prior to 2013, there had been only a single industry-independent study of trace gas emissions from the

Athabasca oil sands mining operations (Simpson et al., 2010; Howell et al., 2014). The data showed

elevated concentrations in n-alkanes (30% of the total quantified hydrocarbon emissions), cycloalkanes

(49%), and aromatics (15%) in plumes from an oil sands surface mining facility intercepted from a single

aircraft flight. These compounds are associated with oil and gas developments including mining,

upgrading, and transportation of bitumen (Siddique et al., 2006). Specifically, these activities involve the

use of naphtha, a complex mixture of aliphatic and aromatic hydrocarbons in the range of $C_3$ to $C_{14}$

containing n-alkanes (e.g., n-heptane, n-octane, and n-nonane) and benzene, toluene, ethylbenzene,

and xylenes (BTEX).

In August 2013, a comprehensive air quality study as a part of the Joint Oil Sands Monitoring (JOSM)

plan (JOSM, 2012), referred to here as the 2013 JOSM intensive study was conducted. This study was

performed in northern Alberta at two ground sites in and near Fort McKay and from a National Research

Council of Canada (NRC) Convair 580 research aircraft to characterize oil sands emissions and their

downwind physical and chemical transformations (Gordon et al., 2015; Liggio et al., 2016; Li et al., 2017).

One ground site, located at the Wood Buffalo Environmental Association (WBEA) air monitoring station





(AMS) 13 (Fig. 1), was equipped with a comprehensive set of instrumentation to measure
concentrations of trace gases and aerosols (Table 1). As part of this effort, a gas chromatograph
equipped with an ion trap mass spectrometer (GC-ITMS) was deployed at AMS 13. When air masses
passing over regions with industrial activities were observed (as judged from a combination of local wind
direction and tracer measurements), the total ion chromatogram showed an analytically unresolved
hydrocarbon signal associated with intermediate volatile organic compounds (IVOCs) with saturation
concentration ($C^*$) in the range $10^5$ µg m$^{-3}$< $C^*$ < $10^7$ µg m$^{-3}$ (Liggio et al., 2016).
Emission estimates for analytically unresolved hydrocarbons range from $5\times10^6$ kg year$^{-1}$ to $14\times10^6$ kg
year$^{-1}$ for the two facilities that reported such emissions  (Li et al., 2017).  Using aircraft measurements
during the 2013 study, Liggio et al. (2016) showed that IVOCs contributed to the majority of the
observed secondary organic aerosol (SOA) mass production in a similar fashion as anthropogenic VOCs
contributed to SOA production during the Deepwater Horizon oil spill (de Gouw et al., 2011) and rivaling
the magnitude of SOA formation observed downwind of megacities (Liggio et al., 2016),  though
ultimately it has remained unclear which activities are associated with IVOC emissions.
In this paper, concurrent measurements of air pollutants at the AMS 13 ground site during the 2013
JOSM intensive study are presented and analyzed using principal component analysis (PCA) to elucidate
the origin of the IVOCs in the Athabasca oil sands. The analysis presented here is a receptor analysis
focusing on the normalized variability of pollutants impacting the AMS 13 ground site and hence does
not constitute a comprehensive emission profile analysis of the oil sands facilities as a whole, for which
aircraft-based measurements and/or direct plume or stack measurements are more suitable. The PCA
was complemented by bivariate polar plots (Carslaw and Ropkins, 2012; Carslaw and Beevers, 2013) to
show the spatial distribution of sources in the region as a function of locally measured wind direction
and speed. A second PCA was performed to investigate which components correlate with (and generate)
secondary pollutants, i.e., pollutants that are formed by atmospheric processes. Potential sources and



processes contributing to each of the components identified by PCA are discussed.

**2. Experimental**
**2.1 Measurement location**
Measurements of air pollutants were made at AMS 13 routine air monitoring station (Fig. 1), which is
operated by WBEA. The site is located at 111.6423° W longitude and 57.1492° N latitude about 3 km
from the southern edge of the community of Fort McKay, 300 m west from a public road, and 1 km west
of the Athabasca river. The immediate vicinity of the site consisted of mixed-leaf boreal forest with a
variety of tree species, including poplar, aspen, pine and spruce trees (Smreciu et al., 2013). The site was
accessible via a gravel road; traffic on this road was restricted during the study period (August -
September, 2013).
The site is impacted by emissions from nearby oil sands facilities (Table 1 and Fig. 1), including a large
surface mining site operated by Syncrude Canada whose northeastern corner is located 3.5 km to the
south of AMS 13 (and which is adjacent to the 5 km long Syncrude – Mildred Lake (SML) tailings pond)
and from a large upgrader stack facility operated by Suncor Energy Inc. located to the Southeast. There
are additional oil sands facilities operated (during the study period) by Canadian Natural Resources
Limited, Imperial Oil, and Shell Canada to the North and Northeast.




**Table 1**. Oil sands facilities located within 30 km of AMS 13. Distances were estimated using coordinates
provided in the National Pollutant Release Inventory (NPRI, 2013) and do not account for the size of
each facility whose boundaries may be considerably closer to (or further away from) AMS 13. PACPRM =
Petroleum and coal products refining and manufacturing; OGPS = Oil and gas pipelines and storage.

| Company | Name | Type | Direction | Distance (km) |
|---|---|---|---|---|
| Syncrude Canada Ltd. | Mildred Lake Plant Site | PACPRM | S | 12.2 |
| Athabasca Minerals Inc. | Susan Lake Gravel Pit | Mining and Quarrying | N | 15.5 |
| Syncrude Canada Ltd. | Aurora North Mine Site | PACPRM | NE | 18.7 |
| Suncor Energy | Suncor Energy Inc. Oil Sands | PACPRM | SE | 19.4 |
| Enbridge Pipelines Inc. | MacKay River Terminal | OGPS | WSW | 19.7 |
| Suncor Energy | MacKay River, In-Situ, Oil Sands Plant | PACPRM | WSW | 19.9 |
| Enbridge Pipelines Inc. | Athabasca Terminal | OGPS | SE | 21.2 |
| Williams Energy | Fort McMurray Hydrocarbon Liquids Extraction Facility | Conventional oil and gas extraction | SE | 21.6 |
| Canadian Natural Resources Limited | Horizon Oil Sands Processing Plant and Mine | PACPRM | NNW | 21.8 |
| Shell Canada Energy | Muskeg River Mine and Jackpine Mine | PACPRM | NNE | 23.7 |



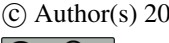




**Figure 1**. Map of oil sands facilities showing locations of surface mines and tailings ponds, downloaded
from the Oil Sands Information Portal (Alberta, 2017). The red star indicates the location of AMS 13.

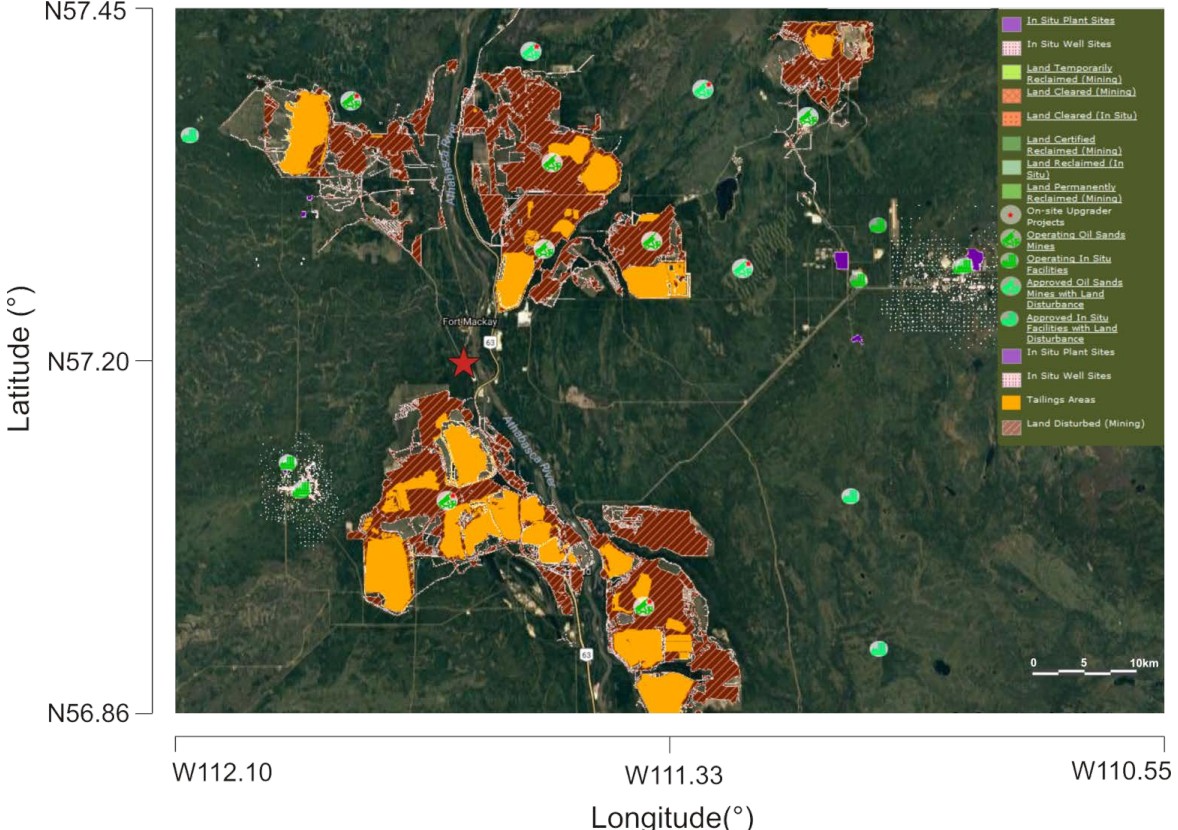


**2.2 Instrumentation**
A large number of instruments was deployed for this study; a partial list whose data were utilized in this
manuscript is given in Table 2. A detailed description of these instruments is given in the S.I. Sample
observations of analytically unresolved hydrocarbons by GC-ITMS and how these data were used in the
analysis are described in section 2.2.1 below.



**Table 2**. Instruments used to measure ambient gas-phase and aerosol species during the 2013 JOSM
intensive study at AMS 13.

| Instrument and Model | Species measured | Operated by | Reference |
| --- | --- | --- | --- |
| Picarro CRDS G2401 | CO, $CO_2$, $CH_4$ | York University and ECCC | (Chen et al., 2013; Nara et al., 2012) |
| Thermo Scientific, Model 42i | $NO_y$ | University of Calgary | (Tokarek et al., 2014; Odame-Ankrah, 2015) |
| Blue diode cavity ring-down spectroscopy | $NO_2$ | University of Calgary | (Paul and Osthoff, 2010; Odame-Ankrah, 2015) |
| Thermo Scientific Model 49i | $O_3$ | University of Calgary | (Tokarek et al., 2014; Odame-Ankrah, 2015) |
| Griffin/FLIR, model 450 GC-ITMS | VOCs | University of Calgary | (Tokarek et al., 2017; Liggio et al., 2016) |
| Thermo Scientific CON101 | TS | ECCC | n/a |
| Thermo Scientific 43iTLE | $SO_2$ | ECCC | n/a |
| AIM-IC | $NH_{3(g)}$, $NH_4^+{}_{(p)}$ | University of Toronto | (Markovic et al., 2012) |
| Aerodyne SP-AMS | rBC, $NH_4^+{}_{(p)}$, $SO_4^{2-}{}_{(p)}$, $NO_3^-{}_{(p)}$, $Cl^-{}_{(p)}$, organics | University of Toronto and ECCC | (Onasch et al., 2012) |
| TSI APS 3321 | $PM_{10-1}$ size distribution | University of Calgary | (Peters and Leith, 2003) |
| TSI SMPS (3081 DMA, 3776 CPC) | $PM_1$ size distribution | University of Alberta | (Wang and Flagan, 1990) |
| EcoChem Analytics PAS 2000CE | pPAH | ECCC | (Wilson et al., 1994; Burtscher et al., 1982) |




### 2.2.1 Analytically unresolved hydrocarbon signature

As previously reported (Liggio et al., 2016), the total ion chromatogram of the GC-ITMS occasionally

showed elevated and analytically unresolved hydrocarbons in the volatility range of $C_{11} - C_{17}$ with

saturation vapor concentration ($C^*$) from $10^5$ µg m$^{-3}$ < $C^*$ < $10^7$ µg m$^{-3}$. An example is shown in Fig. 2.

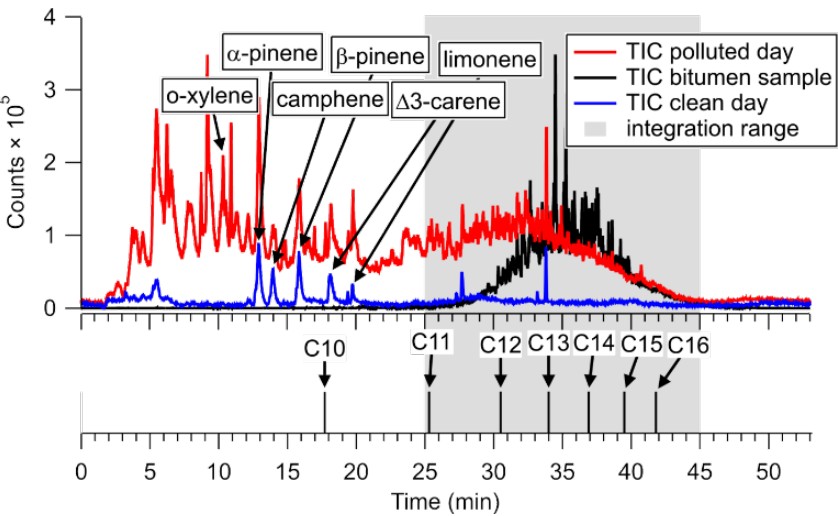

**Figure 2. (Top)** Total ion chromatograms of air samples collected on August 27, 2013 from 18:04 to

18:14 UTC (red) and on August 28, 2013 from 13:43 to 13:53 UTC (blue). The TIC of a head space sample

of ground-up bitumen collected post-campaign is superimposed (black). The gray area indicates the

range over which IVOC signal was integrated. **(Bottom)** Retention times of n-alkanes.

This unresolved signal was integrated in all ambient air chromatograms from a retention time of 25 min

to 45 min (gray area in Fig. 2). A qualitatively similar unresolved signal was observed in an offline

analysis of the headspace above ground-up bitumen (Fig. 2, black trace). In this particular case, the

ambient air chromatogram also shows enhancements of lower molecular weight hydrocarbons (possibly

from naphtha) that were not observed in the bitumen sample.




### 2.3 Principal Component Analysis

The PCA was carried out using the "Statistical Analysis System" (SAS™) Studio 3.4 software (SAS, 2015)
using a method similar to that described by Thurston et al. (2011; 1985). The source-related
components and their associated profiles are derived from the correlation matrix of the input trace
constituents. This approach assumes that the total concentration of each "observable" (i.e., input
variable) is made up of the sum of contributions from each of a smaller number of pollution sources and
that variables are conserved between the points of emission and observation.

### 2.3.1 Selection of variables

22 variables whose ambient concentrations are dominated by primary emissions or which are formed
very shortly after emission (such as the less oxidized oxygenated organic aerosol (LO-OOA) factor
observed by the SP-AMS, see below) were included in the PCA (Table 3). These variables included $CO_2$,
$CH_4$, $NO_y$, CO, and $SO_2$, which are known to be emitted in the oil sands region from stacks, the mine fleet
and faces, tailings ponds, and by fugitive emissions (Percy, 2013). The median $NO_x$ (= NO + $NO_2$) to $NO_y$
ratio was 0.85, consistent with the close proximity of the measurement site to emission sources and
limited chemical processing. Because $NO_x$ constituted a large fraction of $NO_y$, its temporal variation was
captured by the latter, and it was not included as a separate variable in the PCA analysis.
For this work, mixing ratios of all non-methane hydrocarbons (NMHCs) that were quantified (i.e., o-
xylene, the n-alkanes decane and undecane, the aromatics 1, 2, 3- and 1, 2, 4-TMB, as well as limonene
and α- and β-pinene) were included as variables. In addition, the aforementioned unresolved signal
associated with IVOCs was included as a variable by integrating total GC-ITMS ion counts ($m/z$ 50 –425)
over a retention time range of 25-45 min (retention index range of 1100 to 1700).





Gas-phase ammonia was included as a variable because elevated reduced nitrogen concentrations have
been observed in the region and were linked to the use of ammonia on an industrial scale, for example
as a floating agent and for hydrotreating (Bytnerowicz et al., 2010). Total sulfur and total reduced sulfur
were added as tracers of upgrader stack $SO_2$ emissions and of "odours", believed to be emitted from oil
sands tailings ponds which continue to be of concern in surrounding communities (Small et al., 2015;
Percy, 2013; Holowenko et al., 2000).
Refractory black carbon was added as a variable since it is present in Diesel truck exhaust and in biomass
burning plumes and, hence, a combustion tracer (Wang et al., 2016; Briggs and Long). pPAHs were
included because of their association with facility stack emissions and combustion particles in the area
(Allen, 2008; Grimmer et al., 1987). Hydrocarbon-like organic aerosol (HOA) was included as a surrogate
for fossil fuel combustion by vehicles (Jimenez et al., 2009). The LO-OOA factor was included as it is
unique to the Alberta oil sands and appears to form rapidly after emission of precursors (Lee et al., In
prep). Supermicron aerosol volume ($PM_{10-1}$, i.e., the volume of particles between $PM_{10}$ and $PM_1$) was
also included as a tracer of coarse particles from primary sources, which are expected to be dominated
by dust emissions.



**Table 3.** Variables observed at the AMS 13 ground site during the 2013 JOSM campaign used for PCA.

| Variable | Unit | Median[a] | Average[a,b] | Standard deviation[a,b] | LOD[e] | Min.[a] | Max.[a] |
|---|---|---|---|---|---|---|---|
| **Anthropogenic VOCs** | | | | | | | |
| o-xylene | pptv[f] | 5 | 30 | 69 | 1 | < LOD | 635 |
| 1,2,3 - TMB | pptv | 1.7 | 4.3 | 7.9 | 0.2 | < LOD | 67 |
| 1,2,4 - TMB | pptv | 2.1 | 7.7 | 14.7 | 0.2 | < LOD | 107 |
| decane | pptv | 0.5 | 8.5 | 18.2 | 0.1 | < LOD | 125 |
| undecane | pptv | 0.4 | 3.0 | 6.3 | 0.1 | < LOD | 37 |
| **Biogenic VOCs** | | | | | | | |
| α-pinene | pptv | 477 | 542 | 401 | 1 | 19 | 1916 |
| ß-pinene | pptv | 390 | 467 | 334 | 1 | 18 | 1594 |
| limonene | pptv | 150 | 179 | 158 | 2 | < LOD | 711 |
| **Combustion tracers** | | | | | | | |
| $NO_y$ | ppbv | 1.79 | 4.00 | 5.44 | 0.01 | 0.13 | 41.6 |
| rBC | µg m$^{-3}$ | 0.13 | 0.20 | 0.10 | 0.02 | < LOD | 0.90 |
| CO | ppbv | 117.6 | 120.0 | 18.2 | 5.7[h] | 90.9 | 241.2 |
| $CO_2$ | ppmv | 420.2 | 433.2 | 39.5 | 0.4[h] | 386.0 | 577.7 |
| **Aerosol species** | | | | | | | |
| pPAH | ng m$^{-3}$ | 1 | 2 | 2 | 1[c] | < LOD | 14 |
| $PM_{10-1}$ | µm$^3$ cm$^{-3}$ | 11.2 | 14.4 | 12.9 | 0.003 | 1.0 | 79.5 |
| HOA | µg m$^{-3}$ | 0.31 | 0.43 | 0.35 | N/A[g] | 0.04 | 2.32 |
| LO-OOA | µg m$^{-3}$ | 1.19 | 2.00 | 2.26 | N/A[g] | 0.11 | 15.6 |
| **Sulfur species** | | | | | | | |
| Total sulfur (TS) | ppbv | 0.22 | 1.41 | 4.27 | 0.13 | < LOD | 33.3 |
| $SO_2$ | ppbv | < LOD | 1.0 | 4.0 | 0.2 | < LOD | 33.5 |
| Total reduced sulfur (TRS) | ppbv | 0.26 | 0.38 | 1.05 | 0.2 | < LOD | 14.8 |
| **Other** | | | | | | | |
| IVOCs | Counts × min | 1.8×10$^7$ | 3.4×10$^7$ | 4.2×10$^7$ | N/A[g] | 1.4×10$^6$ | 2.5×10$^8$ |
| $CH_4$ | ppbv | 1999.2 | 2065.5 | 169.6 | 1.8[h] | 1880 | 2959 |
| $NH_3$ | µg m$^{-3}$ | 0.79 | 1.10 | 1.03 | 0.05 | 0.06 | 5.75 |

[a] Values were determined only from data points included in PCA analysis, not from entire campaign.

[b] Average and standard deviation were calculated before zeros were replaced with 0.5×LOD.

[c] Estimated.

[e] LOD = limit of detection.

[f] parts-per-trillion by volume (10$^{-12}$)

[g] N/A = data not available

[h] calculated using 3 × standard deviation at ambient background levels




To assess which components have the greatest impact on secondary product formation, a second PCA
was performed which included variables mainly formed through atmospheric chemical processes and
whose concentrations more strongly depend on air mass chemical age than those variables selected
initially. In this PCA, odd oxygen ($O_x = O_3 + NO_2$), submicron aerosol $SO_4^{2-}{}_{(p)}$, $NO_3^-{}_{(p)}$, $NH_4^+{}_{(p)}$, a second,
more-oxidized OOA factor (MO-OOA), and $PM_1$ volume were included, increasing the total number of
variables to 28 (Table 4).

**Table 4.** Variables added in the second PCA. Particle-phase concentrations, i.e., $SO_4^{2-}{}_{(p)}$, $NO_3^-{}_{(p)}$, $NH_4^+{}_{(p)}$
and MO-OOA were made by aerosol mass spectrometry and account for $PM_1$ only.

| Variable | Unit | Median | Average | Standard deviation | LOD | Min. | Max. |
|---|---|---|---|---|---|---|---|
| $O_x$ | ppbv | 7.35 | 11.1 | 10.6 | 1 | <LOD | 41.1 |
| $SO_4^{2-}{}_{(p)}$ | µg m$^{-3}$ | 0.3 | 0.8 | 1.1 | 0.1 | <LOD | 6.6 |
| $NO_3^-{}_{(p)}$ | µg m$^{-3}$ | 0.08 | 0.13 | 0.13 | 0.01 | 0.01 | 0.72 |
| $NH_4^+{}_{(p)}$ | µg m$^{-3}$ | 0.13 | 0.28 | 0.37 | 0.05 | <LOD | 2.21 |
| MO-OOA | µg m$^{-3}$ | 1.65 | 1.83 | 0.960 | N/A | $1.41 \times 10^{-6}$ | 4.65 |
| $PM_1$ volume | µm$^3$ cm$^{-3}$ | 2.48 | 3.77 | 3.72 | N/A | 0.35 | 20.9 |




**2.3.2 Treatment of input data**
Data used in the PCA analysis were averaged to match the time resolution of the GC-ITMS VOC and IVOC
measurements, i.e. over 10 minute long periods (spaced ~ 1 hr apart) set by the start and stop times of
the GC-ITMS pre-concentration period. When concentrations were below their respective limit of
detection (LOD; values are given in Table 3), half the reported LOD was used to minimize bias (Harrison
et al., 1996; Buhamra et al., 1998). Prior to PCA, input variables were standardized to eliminate unit
differences by subtracting the mean concentration $\overline{C_i}$ of pollutant $i$ from the concentration of sample $k$
($C_{i,k}$) and dividing by the standard deviation ($s_i$) of all samples included in the PCA analysis.
$$Z_{i,k} = \frac{C_{i,k} - \overline{C_i}}{s_i}$$     (1)
Here, $Z_{i,k}$ is the standardized pollutant concentration. In total, 218 data points from all identified species
over the period of the campaign were used for the main PCA analysis.

**2.3.3 PCA solutions**
In this work, the Varimax method (Kaiser, 1958) was used to rotate the loading matrix. This method is an
orthogonal rotation (i.e., components are not expected to correlate) which minimizes the impact of high
loadings, making the results easier to interpret (Kaiser, 1958). Several criteria (Table S-10) were
considered for component selection: the latent root criterion, i.e., on the basis that rotated eigenvalues
must be greater than unity, the (cumulative) percentage of variance criterion, where the extracted
components accounts for >95% of the variance, and the Scree test (Fig. S-1) (Thurston and Spengler,
1985; Guo et al., 2004; Hair et al., 1998; Cattell, 1966). For the optimal solution presented in the main
manuscript, the 95% variance criterion was chosen, providing a 10-component solution for the PCA with



only primary variables and an 11-component solution for the PCA with both primary and secondary
variables. Solutions with fewer and more components are presented in the supplemental material
section.
Time series of each of the components were calculated by multiplying the original standardized matrix
by the rotated loading matrix and were used to generate bivariate polar plots (section 2.4).

**2.4 Bivariate polar plots**
The PCA was complemented by bivariate polar plots showing the wind speed and direction dependence
of air pollutant concentrations. The use of these representations implies a linear relationship between
local wind conditions and air mass origin, which may not be always the case (for example, during or after
stagnation periods). In addition, local topography, such as the Athabasca river valley, complicates
regional air flow patterns and limit the interpretability of polar plots in general and in particular to the E
of AMS 13, where the river valley is located. The plots were generated with the Openair software
package (Carslaw and Ropkins, 2012; Carslaw and Beevers, 2013) using the R programming language and
the open-source software "RStudio: Integrated development environment for R" (RStudio Boston,
2017). The default setting (100) was used as the smoothing function.



## 3. Results

### 3.1. Overview of the data set

Time series of the 22 pollution tracers chosen for PCA analysis are presented in Fig. 2, grouped

approximately by source type. Statistics of the data (i.e., median, average, maxima, minima, etc.) are

summarized in Table 3.





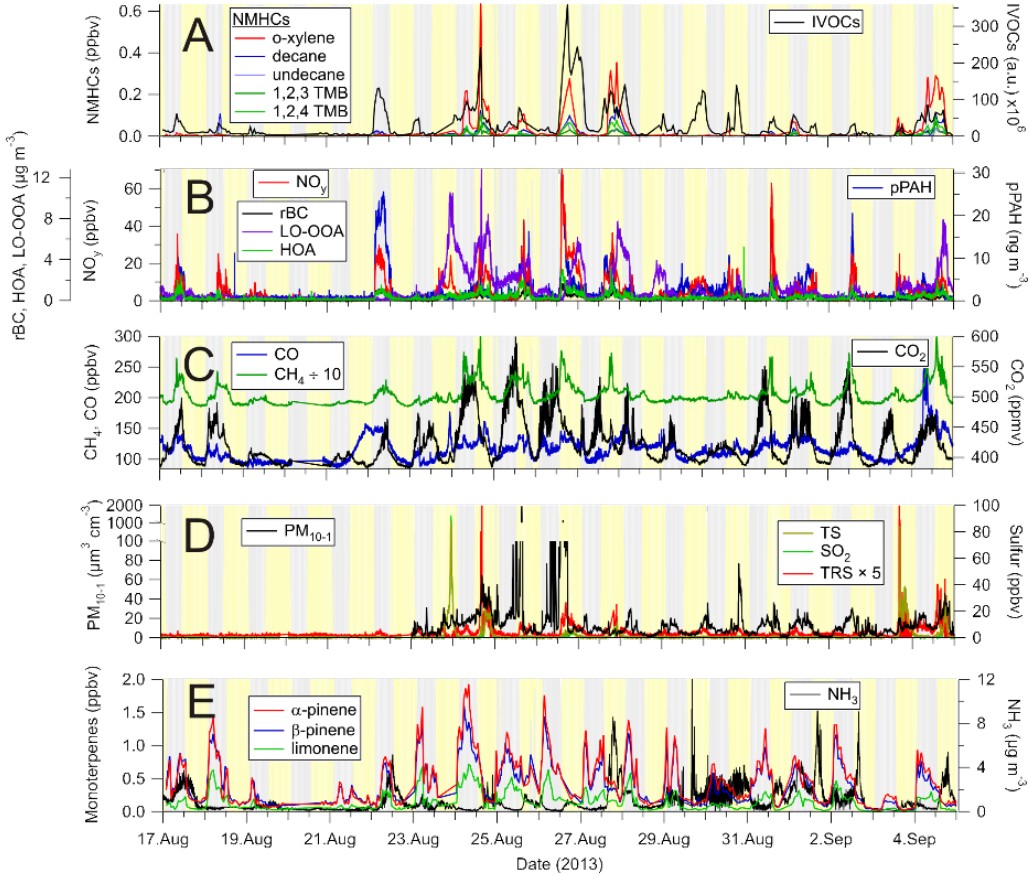


**Figure 3**. Time series of selected pollution tracers observed at the AMS 13 ground site in the Athabasca

oil sands during the 2013 JOSM measurement intensive. The gray and yellow backgrounds represent
night and day, respectively. (**A**) Selected non-methane hydrocarbons (NMHCs) and IVOCs. (**B**)
Combustion product tracers: refractory black carbon (rBC), total odd nitrogen (NO$_y$) and particle surface
bound polycyclic aromatic hydrocarbons (pPAH), and organic aerosol components: hydrocarbon-like
organic aerosol (HOA) and less oxidized oxygenated organic aerosol (LO-OOA). (**C**) Methane (CH$_4$),
carbon dioxide (CO$_2$) and monoxide (CO). (**D**) Total sulfur (TS), sulfur dioxide (SO$_2$), and total reduced
sulfur (TRS) and PM$_{10}$ particle volume. (**E**) Biogenic VOCs (α-pinene, ß-pinene and limonene) and
ammonia (NH$_3$).



Time series of VOCs of primarily anthropogenic origin (i.e., o-xylene, 1, 2, 3- and 1, 2, 4-TMB, etc.) as
well as the IVOC signature are shown in Fig. 3A. The abundances of these species, as well as the other
compounds, were highly variable and varied as a function of time of day (i.e., boundary layer mixing
height) and air mass origin, with higher VOC concentrations generally observed during daytime. The VOC
concentrations varied between nearly pristine, remote conditions, with concentrations below
detectable limits, to mixing ratios of aromatic species exceeding 100 pptv. The concentration range of o-
xylene is within the extremes reported by WBEA in their 2013 annual report (WBEA, 2013), exemplifying
that the data set is representative of typical pollutant levels in this region.
While there is some obvious covariance between variables (i.e., when the mixing ratios of one particular
VOC increases, so do others), the ratios of hydrocarbons varied considerably. For example, on August
18, 10:50 UTC, the n-decane to o-xylene ratio was ~22:1, whereas on August 24, 07:40 UTC it was ~1:5.7.
The IVOC magnitude also varied greatly and often increased and decreased in tandem with the other
VOCs (e.g., on Aug 24, 16:30 UTC) but also increased independently from the other VOC abundances
(e.g., on Aug 30, 01:20 UTC, and on the night of Aug 22). This behaviour suggests the presence of
multiple sources with distinct signatures that are being sampled to a varying extent at different times.
This, coupled with the intermittency of the highly elevated signals, presents an analysis problem
frequently encountered in environmental analysis that is usually investigated through a factor or
principal component analysis (Thurston et al., 2011; Guo et al., 2004).
Presented in Fig. 3B are the time series of $NO_y$, rBC and pPAH abundances, all of which are combustion
byproducts. For example, rBC is emitted from combustion of fossil fuels, biofuels, open biomass burning,
and burning of urban waste (Bond et al., 2004). Similar to the VOCs, the abundances of these species
varied greatly, from very low, continental background levels (i.e., <100 pptv of $NO_y$, < LOD for rBC and
pPAHs) to polluted concentrations (i.e., > 60 ppbv of $NO_y$, > 1 µg m$^{-3}$ rBC, > 10 ng m$^{-3}$ pPAHs)
characteristic of polluted urban and industrial areas. When high concentrations of $NO_y$ were observed,



its main component was $NO_x$ (data not shown), which is a combustion byproduct usually associated with
automobile exhaust. In the Alberta oil sands, emissions from off-road mining trucks as well as the
upgrading processes are the main contributors to the $NO_y$ burden (Percy, 2013; Watson et al., 2013).
Shown in Fig. 3C are the mixing ratios of the greenhouse gases $CH_4$ and $CO_2$ along with CO. Abundances
of $CO_2$ were clearly attenuated by photosynthesis and respiration of the vegetation near the
measurement site, as judged from the strong diurnal cycle in its concentration (not shown). Maxima
typically occurred shortly after sunrise, coincident with the expected break-up of the nocturnal
boundary layer. In addition to biogenic emissions from vegetation and soil, $CO_2$ originates from a variety
of point and mobile sources in this region, including off-road mining trucks (Watson et al., 2013) and the
extraction, upgrading, and refining of bitumen and on-road vehicle sources in the area (Nimana et al.,
2015a, b). Concentrations of $CO_2$ spiked whenever these emissions were transported to the
measurement site.
Concentrations of $CH_4$ also exhibit a diurnal cycle, with higher concentrations generally observed at
night and peaking in the early morning hours. While $CH_4$ and $CO_2$ mixing ratios frequently correlated in
plumes, their ratios were variable overall, suggesting they originated from distinct sources. Potential
methane point sources in the region include microbial production in tailings ponds (Siddique et al.,
2012) and fugitive emissions associated with the mining and processing of bitumen (Johnson et al.,
2016). Indeed, a recent analysis shows tailings ponds and open pit mining sources to be the largest
sources of $CH_4$ in the region (Baray et al., submitted, 2017).
Similar to the anthropogenic VOCs, the abundances of $CH_4$ and $CO_2$ were highly variable and ranged
from minima of 1.88 and 384 ppmv to maxima of 2.96 and 578 ppmv, corresponding to maximum
enhancements of 1.63 and 1.47 relative to tropospheric global monthly means of 1.806±0.001 and
394.3±0.1 ppmv for July, 2013 (Dlugokencky, 2017b, a), respectively.





Mixing ratios of CO also varied with time but generally were not elevated greatly (median 118 ppbv)
above background levels (minimum 91 ppbv), except for occasional spikes in concentration (Fig. 3C).
Carbon monoxide is a tracer of biomass burning and fossil fuel combustion, in particular in automobiles
with poorly performing or absent catalytic converters, but is also a byproduct of the oxidation of VOCs,
in particular of methane and isoprene which are oxidized over a wide area upwind of AMS 13 (Miller et
al., 2008).
Time series of sulfur species and $PM_{10-1}$ volume are shown in Fig. 3D. The TS and $SO_2$ data are dominated
by intermittent plumes containing $SO_2$ mixing ratios exceeding 5 ppbv. The highest mixing ratio
observed was 92.5 ppbv (in between the preconcentration periods of the GC-ITMS). Mixing ratios of $SO_2$
exhibited the most variability of all pollutants, as judged from the standard deviation of each of the
measurements (Table 3). TRS levels were generally small (< 1 ppbv) and variable, except for plumes; TRS
abundances in plumes, however, are more uncertain since they were calculated by subtraction of two
large numbers. When TS and $SO_2$ abundances were low (< 1 ppbv), TRS abundances were variable and
occasionally exhibited spikes that did not show any obvious correlation with other variables, suggesting
the presence of one or more distinct TRS sources. $PM_{10}$ volume concentrations varied a lot as well and,
just like TRS, did not show an obvious correlation with other variables. Fugitive dust emissions likely
contributed to much of the $PM_{10}$ volume in the Athabasca oil sands region (Wang et al., 2015).
Time series of monoterpene mixing ratios are shown in Fig. 3E. α-Pinene was generally the most
abundant monoterpene, followed by β-pinene. Their ratio, averaged over the entire campaign was
1:0.85, though occasionally the α- to β-pinene ratio was below 1:2 (e.g., on Aug 28, 14:50 UTC and Sept
5, 12:40 UTC). Terpene mixing ratios were generally higher at night than during the day, with maxima of
1.9 and 1.6 ppbv, respectively, a diurnal pattern consistent with what has been observed at other forest
locations (Fuentes et al., 1996). Monoterpenes are emitted by plants via both photosynthetic and non-
photosynthetic pathways (Fares et al., 2013; Guenther et al., 2012); at night, their emissions accumulate



in a shallow nocturnal boundary layer, whereas during daytime, they are entrained aloft (above the
canopy) and oxidized by the hydroxyl radical (OH) and $O_3$, which are more abundant during the day than
at night (Fuentes et al., 1996). α- and β-pinene mixing ratios were lowest mid-day (median values at
noon of 140 and 133 pptv, respectively). The largest daytime concentrations were observed on Aug 25, a
cloudy day (as judged from spectral radiometer measurements of the $NO_2$ photolysis frequency): on this
particular day, mixing ratios at noon were 687 and 850 pptv, respectively.
Also shown in Fig. 3E is the time series of ammonia. These data were dominated by spikes which were
observed sporadically and did not correlate with other variables, suggesting the presence of nearby
ammonia point sources. Ammonia was not as variable as some of the other pollutants (e.g., the
anthropogenic VOCs, sulfur species) as judged from its standard deviation (Table 3), which suggests a
geographically more disperse source or sources similar to CO or $CH_4$, which have a "background". This is
consistent with a recent study by Whaley et al. (2017) that estimated over half (~57%) of the near-
surface $NH_3$ during the study period originated from $NH_3$ bi-directional exchange (i.e. re-emission of $NH_3$
from plants and soils), with the remainder being from a mix of anthropogenic sources (~20%) and forest
fires (~23%).

**3.2. Principal component analysis**
**3.2.1. PCA analysis with primary variables**
The loadings of the optimum solution are presented in Table 5. The 10-component solution accounts for
a cumulative variance of 95.5%. The communalities for the analysis, i.e., the fraction of total pollutant
observations accounted for by the PCA are all greater than 85%, with the lowest communality obtained
for the IVOCs (0.86).
In the following, an overview of the observed components is presented. Associations with r>0.7, r>0.3,



and r>0.1 are referred to as "strong", "moderate", and "weak", respectively. Hypothesized
identifications are given in section 4 and are summarized in Table 6 and Fig. 4.
The component accounting for most of the variance of the data, component 1, is strongly associated
with the anthropogenic VOCs (r > 0.87), moderately associated with $CH_4$ (r = 0.59), TRS (r = 0.59), HOA (r
= 0.40), LO-OOA (r = 0.45), CO (r = 0.41), and the IVOCs (r = 0.31), and weakly associated with $NO_y$ (r =
0.27) and rBC (r = 0.30). Component 2 is strongly associated with the combustion tracers $NO_y$ (r = 0.82),
rBC (r = 0.77), HOA (r = 0.74), and pPAH (r = 0.94), moderately associated with $CH_4$ (r = 0.39) and IVOCs (r
= 0.39), and weakly associated with ammonia (r = 0.20), CO (r = 0.18) and undecane and decane (r = 0.27
and 0.22, respectively). Component 3 is strongly associated (r > 0.9) with the biogenic VOCs and
moderately associated with $CO_2$ (r = 0.48) and shows weak negative correlations with $NO_y$ (r = -0.26),
ammonia (r = -0.24), and $SO_2$ (r = -0.15). Component 4 is strongly associated with $SO_2$ and TS (r = 0.97
and 0.93, respectively) and weakly with $NO_y$ (r = 0.21) and LO-OOA (r = 0.28).
Components 1 through 4 emerged regardless of the number of components used to represent the data,
whereas the structure of components 5 through 10 only fully emerged in the 10-component solution
(see S.I.). Hence, components 6 through 10 are somewhat tentative as many (i.e., 7 – 9) are single
variable components and have eigenvalues close to or below unity, i.e., account for less variance than
any single variable. For the purpose of this manuscript, this is inconsequential as components 6 – 10 are
not associated with IVOCs.





**Table 5.** Loadings for the 10-factor, optimal solution (primary variables only). Coefficients with Pearson
correlation coefficients r>0.3 are interpreted as being moderately or strongly associated with a
component and are shown in bold font.

| | 1 | 2 | 3 | 4 | 5 | 6 | 7 | 8 | 9 | 10 | Commu-nalities |
|---|---|---|---|---|---|---|---|---|---|---|---|
| **Anthropogenic VOCs** | | | | | | | | | | | |
| o-xylene | **0.88** | 0.08 | 0.02 | 0.10 | 0.14 | 0.13 | 0.07 | -0.04 | 0.16 | **0.32** | 0.95 |
| 1,2,3 - TMB | **0.93** | 0.16 | 0.07 | 0.05 | 0.05 | 0.11 | 0.04 | -0.02 | 0.18 | -0.01 | 0.95 |
| 1,2,4 - TMB | **0.94** | 0.14 | 0.01 | 0.10 | 0.11 | 0.08 | 0.07 | -0.03 | 0.18 | 0.13 | 0.98 |
| decane | **0.92** | 0.22 | -0.02 | 0.15 | 0.23 | 0.01 | 0.05 | 0.04 | 0.04 | 0.03 | 0.97 |
| undecane | **0.87** | 0.27 | -0.08 | 0.23 | 0.20 | -0.06 | 0.12 | 0.07 | -0.04 | -0.10 | 0.96 |
| **Biogenic VOCs** | | | | | | | | | | | |
| $\alpha$-pinene | -0.03 | -0.08 | **0.98** | -0.11 | 0.02 | 0.04 | 0.01 | -0.08 | 0.02 | 0.01 | 0.98 |
| ß-pinene | -0.02 | -0.08 | **0.98** | -0.12 | 0.02 | 0.03 | 0.02 | -0.07 | 0.00 | 0.01 | 0.98 |
| limonene | 0.07 | -0.03 | **0.92** | -0.08 | 0.12 | 0.24 | 0.05 | -0.11 | 0.03 | -0.05 | 0.95 |
| **Combustion tracers** | | | | | | | | | | | |
| $NO_y$ | 0.27 | **0.82** | -0.26 | 0.21 | 0.22 | -0.04 | 0.02 | 0.10 | -0.08 | 0.01 | 0.92 |
| rBC | **0.30** | **0.77** | 0.03 | 0.05 | **0.44** | 0.10 | 0.09 | 0.13 | 0.12 | -0.10 | 0.94 |
| CO | **0.41** | 0.18 | 0.04 | 0.02 | 0.09 | 0.09 | 0.08 | 0.06 | **0.87** | -0.01 | 0.99 |
| $CO_2$ | 0.09 | 0.08 | **0.48** | -0.12 | -0.03 | **0.77** | 0.25 | -0.14 | 0.05 | -0.08 | 0.95 |
| **Aerosol species** | | | | | | | | | | | |
| pPAH | 0.06 | **0.94** | -0.07 | -0.13 | -0.11 | 0.07 | 0.01 | 0.13 | 0.10 | 0.04 | 0.95 |
| $PM_{10-1}$ | 0.18 | 0.14 | 0.08 | 0.09 | 0.11 | 0.17 | **0.93** | -0.03 | 0.07 | 0.08 | 0.98 |
| HOA | **0.40** | **0.74** | 0.02 | 0.12 | 0.25 | 0.15 | 0.23 | -0.06 | 0.16 | 0.09 | 0.90 |
| LO-OOA | **0.45** | 0.11 | 0.12 | 0.28 | **0.72** | 0.05 | 0.25 | 0.00 | 0.10 | 0.04 | 0.91 |
| **Sulfur** | | | | | | | | | | | |
| TS | 0.25 | 0.04 | -0.16 | **0.93** | 0.08 | -0.05 | 0.07 | -0.02 | 0.01 | 0.12 | 1.00 |
| $SO_2$ | 0.12 | 0.03 | -0.15 | **0.97** | 0.02 | -0.04 | 0.03 | -0.03 | 0.01 | -0.05 | 0.99 |
| TRS | **0.59** | 0.04 | -0.08 | 0.11 | 0.26 | -0.04 | 0.16 | 0.04 | -0.04 | **0.71** | 0.96 |
| **Other** | | | | | | | | | | | |
| IVOCs | **0.31** | **0.39** | 0.12 | -0.08 | **0.74** | -0.02 | -0.02 | -0.06 | 0.02 | 0.20 | 0.86 |
| $NH_3$ | 0.01 | 0.20 | -0.24 | -0.05 | -0.02 | -0.08 | -0.03 | **0.94** | 0.04 | 0.02 | 0.99 |
| $CH_4$ | **0.59** | **0.39** | 0.10 | -0.05 | 0.12 | **0.59** | 0.11 | 0.00 | 0.17 | 0.14 | 0.93 |
| **Eigenvalues** | 5.72 | 3.32 | 3.23 | 2.16 | 1.64 | 1.13 | 1.13 | 0.99 | 0.96 | 0.74 | |
| **% of variance** | 25.99 | 15.08 | 14.69 | 9.80 | 7.46 | 5.14 | 5.13 | 4.51 | 4.36 | 3.35 | |
| **Cumulative variance** | 25.99 | 41.07 | 55.76 | 65.56 | 73.02 | 78.16 | 83.30 | 87.81 | 92.17 | 95.52 | |




**Table 6**. Hypothesized identification of principal components.

| Component | Key observations | Possible source(s) | Relevant references |
|---|---|---|---|
| 1 | Enhancements of aromatics, n-alkanes, TRS, $NO_y$, rBC, HOA, LO-OOA, CO and $CH_4$ | Wet tailings ponds and associated facilities | (Simpson et al., 2010; Small et al., 2015; Percy, 2013; Holowenko et al., 2000; Howell et al., 2014) |
| 2 | Enhancements of $NO_y$, rBC, pPAH and HOA due to engine exhaust | Mine fleet and operations | (Wang et al., 2016; Grimmer et al., 1987; Allen, 2008; Briggs and Long, 2016) |
| 3 | Enhancements of monoterpenes and $CO_2$, weak anticorrelation with $NO_y$ and absence of anthropogenic VOCs | Biogenic emission and respiration | (Guenther et al., 2012; Helmig et al., 1999) |
| 4 | Enhancements of $SO_2$ and TS, weak correlation with $NO_y$ and LO-OOA | Upgrader facilities | (Simpson et al., 2010; Kindzierski and Ranganathan, 2006) |
| 5 | Enhancements of IVOCs, rBC, LO-OOA, $NO_y$, and TRS | Surface exposed bitumen and hot-water based bitumen extraction | this work |
| 6 | Enhancements of $CO_2$ and $CH_4$, absence of combustion tracers | Mine face and soil | (Johnson et al., 2016; Rooney et al., 2012) |
| 7 | Enhancement of $PM_{10-1}$ | Wind-blown dust | (Wang et al., 2015) |
| 8 | Enhancement of ammonia | Fugitive emissions from storage tanks and natural soil/plant emissions | (Bytnerowicz et al., 2010; Whaley et al., 2017) |
| 9 | Enhancement of CO | VOC oxidation | (Marey et al., 2015) |
| 10 | Enhancements of TRS and o-xylene, weak association with $CH_4$ | Composite tailings | (Small et al., 2015; Warren et al., 2016) |






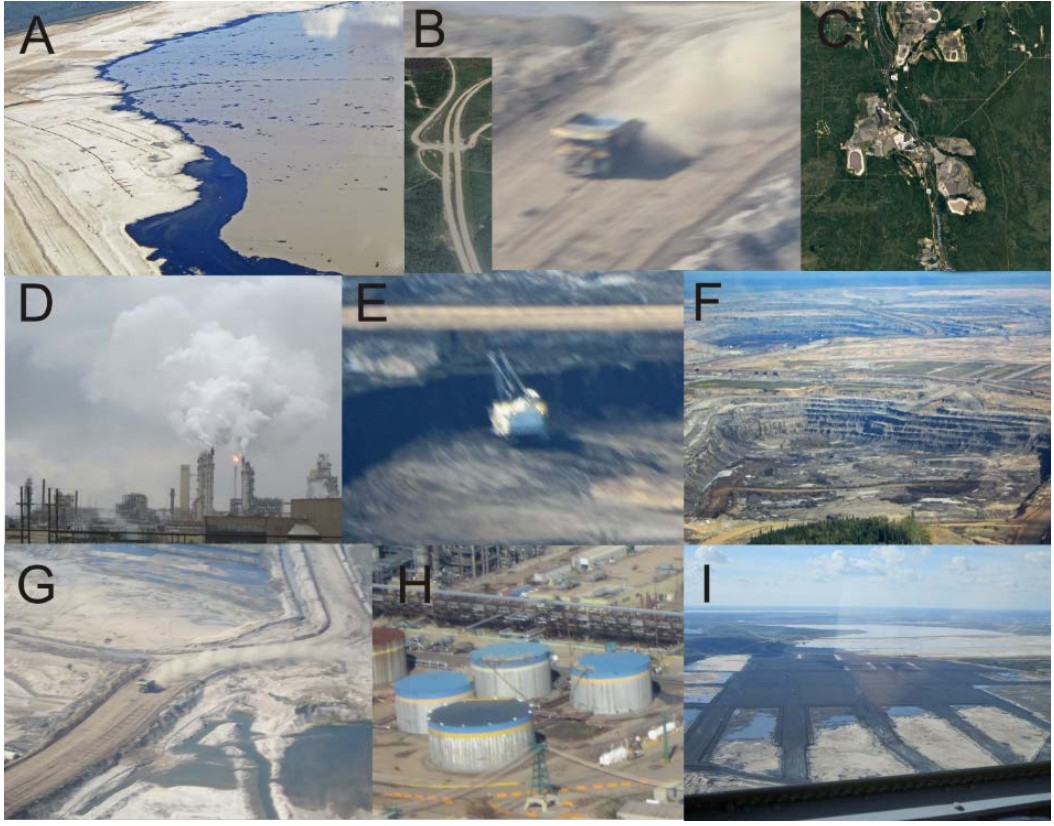


**Figure 4.** Images of likely sources associated with each of the principal components. From top left to

bottom: **(A)** Wet tailings ponds (component 1). **(B)** Mine truck fleet and highway traffic emissions

(component 2). **(C)** Biogenic emissions from vegetation (component 3). **(D)** Upgrader facilities

(component 4). **(E)** Exposed bitumen on mined surfaces (component 5). **(F)** Fugitive greenhouse gas

emissions from mine faces (component 6). **(G)** Wind-blown dust from exposed sand (component 7). **(H)**

Fugitive emissions of ammonia from storage tanks (Component 8). **(I)** Composite (dry) tailings

(component 10). No image is shown for production CO from oxidation of VOCs (component 9).

374





### 3.2.2. Extended PCA analysis with added secondary variables

The loadings of the optimum solution that includes primary and secondary variables are shown in Table

7. In this 11-component solution, the 10 components originally identified were preserved, though their

relative order was changed, with the upgrader component moving from the $4^{th}$ to $2^{nd}$ position. There

was one new component (#6), which encompassed only secondary species, including MO-OOA (r =

0.92), $O_x$ (r = 0.33), $NO_3^-{}_{(p)}$ (r = 0.36), $PM_1$ (r = 0.31) and LO-OOA (r = 0.31).

$NH_4^+{}_{(p)}$, $SO_4^{2-}{}_{(p)}$, and $NO_3^-{}_{(p)}$ are associated with the stack emissions component (#2, with r = 0.84, 0.84

and 0.44, respectively), which also moderately correlated with $PM_1$ (r = 0.44) and $O_x$ (r = 0.36). The

association of secondary variables with the primary components suggests rapid formation of these

secondary products on a time scale that is similar to the transit time of the pollutants to the

measurement site. $PM_1$ and $O_x$ correlated strongly with the major IVOC component (component 5, r =

0.80), which also moderately associated with LO-OOA (r=0.66) and $NO_3^-{}_{(p)}$ (r = 0.59), as well as $NH_4^+{}_{(p)}$

and $SO_4^{2-}{}_{(p)}$ (r = 0.32 and 0.33, respectively).



389    **Table 7.** Loadings for the 11-component solution with the inclusion of variables associated with

390    secondary processes.

| | 1 | 2 | 3 | 4 | 5 | 6 | 7 | 8 | 9 | 10 | 11 | Commu-nalities |
|---|---|---|---|---|---|---|---|---|---|---|---|---|
| **Anthropogenic VOCs** | | | | | | | | | | | | |
| o-xylene | **0.89** | 0.16 | 0.04 | 0.04 | 0.15 | 0.00 | 0.10 | 0.07 | -0.04 | 0.17 | 0.24 | 0.94 |
| 1,2,3 - TMB | **0.91** | 0.13 | 0.10 | 0.16 | 0.09 | 0.07 | 0.11 | 0.03 | -0.03 | 0.16 | -0.08 | 0.95 |
| 1,2,4 - TMB | **0.93** | 0.19 | 0.02 | 0.13 | 0.13 | 0.05 | 0.06 | 0.07 | -0.03 | 0.17 | 0.06 | 0.99 |
| decane | **0.89** | 0.25 | 0.00 | 0.22 | 0.26 | 0.05 | -0.01 | 0.05 | 0.01 | 0.00 | 0.01 | 0.98 |
| undecane | **0.81** | **0.35** | -0.08 | 0.27 | 0.21 | 0.15 | -0.07 | 0.08 | 0.04 | -0.12 | -0.10 | 0.96 |
| **Biogenic VOCs** | | | | | | | | | | | | |
| α-pinene | 0.00 | -0.08 | **0.98** | -0.07 | 0.05 | 0.03 | 0.01 | 0.01 | -0.07 | 0.02 | 0.01 | 0.98 |
| ß-pinene | 0.01 | -0.08 | **0.98** | -0.08 | 0.05 | 0.05 | 0.01 | 0.03 | -0.06 | 0.01 | 0.02 | 0.98 |
| limonene | 0.11 | -0.02 | **0.92** | -0.02 | 0.14 | 0.09 | 0.21 | 0.02 | -0.10 | 0.02 | -0.03 | 0.95 |
| **Combustion tracers** | | | | | | | | | | | | |
| NO$_y$ | 0.23 | 0.20 | -0.27 | **0.82** | 0.21 | -0.06 | -0.07 | 0.03 | 0.10 | -0.10 | 0.01 | 0.92 |
| rBC | 0.22 | 0.15 | 0.05 | **0.80** | 0.43 | 0.15 | 0.10 | 0.05 | 0.09 | 0.07 | 0.00 | 0.95 |
| CO | **0.40** | 0.09 | 0.08 | 0.20 | 0.09 | 0.22 | 0.08 | 0.06 | 0.03 | **0.83** | -0.02 | 0.97 |
| CO$_2$ | 0.12 | -0.07 | **0.50** | 0.08 | -0.03 | 0.09 | **0.75** | 0.28 | -0.12 | 0.03 | -0.08 | 0.95 |
| **Aerosol species** | | | | | | | | | | | | |
| pPAH | 0.06 | -0.10 | -0.06 | **0.93** | -0.07 | -0.06 | 0.07 | 0.03 | 0.15 | 0.13 | -0.05 | 0.94 |
| PM$_{10-1}$ | 0.19 | 0.16 | 0.08 | 0.16 | 0.13 | 0.08 | 0.18 | **0.91** | -0.03 | 0.05 | 0.07 | 0.99 |
| PM$_1$ | 0.24 | **0.44** | 0.00 | 0.17 | **0.70** | 0.31 | -0.06 | 0.11 | -0.04 | 0.07 | -0.14 | 0.90 |
| NH$_4^+$(p) | 0.28 | **0.84** | 0.02 | 0.12 | **0.32** | 0.22 | 0.06 | 0.07 | -0.04 | 0.14 | -0.04 | 0.97 |
| SO$_4^{2-}$(p) | 0.29 | **0.84** | 0.03 | 0.12 | **0.33** | 0.19 | 0.06 | 0.06 | -0.05 | 0.12 | -0.05 | 0.97 |
| NO$_3^-$(p) | **0.30** | **0.44** | 0.09 | 0.23 | **0.59** | **0.36** | 0.08 | 0.15 | -0.13 | 0.02 | 0.24 | 0.92 |
| HOA | **0.37** | 0.18 | 0.02 | **0.77** | 0.25 | 0.10 | 0.10 | 0.18 | -0.08 | 0.13 | 0.14 | 0.93 |
| LO-OOA | **0.37** | **0.40** | 0.12 | 0.16 | **0.66** | 0.31 | 0.03 | 0.12 | -0.06 | 0.00 | 0.27 | 0.97 |
| MO-OOA | 0.10 | 0.15 | 0.09 | 0.00 | 0.10 | **0.92** | 0.05 | 0.07 | 0.10 | 0.16 | -0.03 | 0.95 |
| **Sulfur** | | | | | | | | | | | | |
| TS | 0.27 | **0.90** | -0.20 | 0.03 | 0.04 | -0.04 | -0.09 | 0.07 | 0.00 | -0.04 | 0.18 | 0.98 |
| SO$_2$ | 0.09 | **0.96** | -0.19 | 0.02 | -0.03 | -0.01 | -0.08 | 0.03 | -0.02 | -0.03 | 0.00 | 0.98 |
| TRS | **0.65** | 0.14 | -0.10 | 0.05 | 0.23 | -0.08 | -0.07 | 0.17 | 0.06 | -0.04 | **0.63** | 0.95 |
| **Other** | | | | | | | | | | | | |
| IVOCs | **0.34** | -0.01 | 0.12 | **0.33** | **0.80** | -0.23 | -0.02 | 0.02 | 0.02 | 0.06 | 0.06 | 0.94 |
| NH$_3$ | -0.03 | -0.08 | -0.22 | 0.21 | -0.04 | 0.09 | -0.07 | -0.03 | **0.93** | 0.02 | 0.02 | 0.99 |
| O$_x$ | 0.07 | **0.36** | **-0.62** | 0.01 | 0.27 | **0.33** | **-0.41** | -0.07 | -0.03 | -0.14 | 0.12 | 0.91 |
| CH$_4$ | **0.60** | 0.00 | 0.14 | **0.42** | 0.10 | 0.08 | **0.57** | 0.08 | -0.04 | 0.13 | 0.16 | 0.94 |
| **Eigenvalues** | 5.85 | 4.30 | 3.71 | 3.51 | 2.78 | 1.58 | 1.24 | 1.09 | 1.01 | 0.94 | 0.75 | |
| **% of variance** | 20.90 | 15.34 | 13.25 | 12.52 | 9.92 | 5.65 | 4.43 | 3.88 | 3.59 | 3.37 | 2.66 | |
| **Cumulative variance** | 20.90 | 36.24 | 49.49 | 62.02 | 71.94 | 77.59 | 82.03 | 85.90 | 89.50 | 92.87 | 95.53 | |



### 3.3 Spatial distribution of IVOC sources

Bivariate polar plots were generated for all components and their dominant, associated variables and are shown in the supplemental material section (Figs. S2-S11). Winds were predominantly from the SW but were also observed often from the S and N. Fig. 5A shows the plot for IVOCs. The highest concentrations were observed when the local wind direction was from the NE, where several facilities including the Aurora North, Musket River and Jackpine mines and large swaths of disturbed and cleared land are located in close proximity to each other (Table 1 and Fig. 1). The second highest IVOC signal intensity was observed when local wind direction was from the SSE.

The bivariate polar plots of the 3 components associated with IVOCs are shown in Fig. 5B-D. These components are associated with winds from the NE, E, SE and S at low to moderate speeds (1-3 m s$^{-1}$). Component 5 (Fig. 5B) was the most strongly correlated with IVOCs and shows the most spatial overlap with the distribution of the IVOC source; however, the intensities differ owing to the association of component 5 with other variables such rBC and LO-OOA.




404

**Figure 5.** Bivariate polar plots related to IVOCs: **(A)** IVOCs from the complete data set. **(B)** Component 5

extracted from the main PCA (Table 5). **(C)** Component 1 extracted from the main PCA. **(D)** Component 2

extracted from the main PCA analysis. Wind direction is binned into 10° intervals and wind direction into

30° intervals. The polar axis indicates wind speed (m s⁻¹). a.u. = arbitrary units.



## 4. Discussion

The main objective of this work is to elucidate the origin of the IVOC signature observed at the AMS 13

ground site downwind from the AB oil sands mining operations (Fig. 2) through a principal component

analysis. The optimum PCA solution identified 10 components, of which three were associated with the

IVOC signature: 1, 2, and 5 (Table 5). The assignments of these components to source types in the oil

sands are given in Table 6 and are discussed below.

Emission inventories show that the facilities that process the mined bitumen are by far the largest

anthropogenic point sources in the oil sands region (NPRI, 2013), consistent with recent aircraft

measurements (Howell et al., 2014; Li et al., 2017; Baray et al., submitted, 2017; Simpson et al., 2010)

which have shown substantial emissions of $NO_y$, $SO_2$, CO, VOCs, $CO_2$, and $CH_4$, from these facilities and

associated mining activities. No single component correlates with all of these variables, suggesting that

the PCA is able to distinguish between source types within the facilities such as tailings ponds

(component 1), stack emissions (component 4), and mining (component 2).

Close-up overflights (Howell et al., 2014; Li et al., 2017; Baray et al., submitted, 2017) were able to

spatially resolve various oil sands facility emission sources (i.e., tailings ponds from upgraders, fluid

coking reactors, hydrocrackers and –treaters); the PCA presented in this manuscript is not expected to

do this in all cases because some emissions would have frequently merged into a single plume by the

time of observation at AMS 13; unless their emissions vary considerably in time, these sources could be

interpreted as originating from a single source in the PCA.

The discussion below focuses on components that are associated with IVOCs (section 4.1), followed by

those that are not (section 4.2). The PCA analysis that included 6 secondary products is discussed in

section 4.3. Components which are not associated with IVOCs and have only tentatively been identified

(i.e., components 6 – 10) are discussed in the S.I.





### 4.1 Sources associated with IVOCs

#### 4.1.1. Component 1: Tailings ponds (wet tailings)

Component 1 is strongly associated with anthropogenic VOCs (r > 0.87) and moderately with TRS (r =

0.59), and $CH_4$ (r = 0.59). These pollutants originate from tailings ponds (Small et al., 2015), though it is

unclear from this analysis how large a source tailings ponds are compared to fugitive emissions of these

pollutants from the nearby processing (e.g., bitumen separation and mining) facilities.

Tailings ponds cover large areas of land and are used to slowly (on a time scale of years to decades)

separate solid components, or tailings, from water used in bitumen extraction. Residual bitumen often

floats to the top of the settling basins. Most tailings ponds are "wet" (as they contain residual naphtha

that is used as a diluent during the transfer of tailings to the ponds) and emit VOCs, $CH_4$, and $CO_2$ (Small

et al., 2015). The presence of o-xylene, TMB and the n-alkanes in component 1 is consistent with the

fugitive release of VOCs from residual naphtha, which contains these compounds (Siddique et al., 2008;

Siddique et al., 2011; Small et al., 2015). Furthermore, the observation of TRS and $CH_4$ from this source is

consistent with the presence of anaerobic sulfur reducing bacteria and methanogens within the ponds,

which degrade not only the residual bitumen (Holowenko et al., 2000; Percy, 2013; Quagraine et al.,

2005) but also the various components of naphtha (Shahimin and Siddique, 2017; Small et al., 2015).

Overall, tailings ponds emissions explain much of the TRS and $CH_4$ concentration variability in this data

set (Table 5) and in a recent aircraft study (Baray et al., submitted, 2017).

While component 1 correlates with $CH_4$ (r = 0.59), it does not correlate with $CO_2$ (r = 0.09). Emissions of

$CH_4$ from tailings ponds due to methanogenic bacterial activity are well-documented (Small et al., 2015;

Yeh et al., 2010) and hence the correlation with $CH_4$ is not unexpected. On the other hand, the lack of

correlation with $CO_2$ seems inconsistent with emission inventories that generally present tailings ponds

as large $CO_2$ sources (Small et al., 2015). One plausible explanation is that tailings ponds are a relatively



small $CO_2$ source overall in the region and that other, larger $CO_2$ sources and sinks (such as
photosynthesis and respiration by the vegetation surrounding the site) dominate the variance impacting
the PCA results. It may also indicate that, at least on aggregate and for the particular ponds detected in
this work, the emissions are in a regime where the release of $CH_4$ dominates over $CO_2$, i.e., the ponds
have, perhaps, become more anoxic than believed to be the case in previous studies and hence emit
more $CH_4$ (Holowenko et al., 2000). For example, Small et al. (2015) showed that older tailings ponds
(those without the addition of fresh froth or thickening treatments) tended to emit more $CH_4$, while
newer ponds are associated with higher VOC emissions. It is likely that component 1 is dominated by the
nearest pond (the Mildred Lake settling basin, 6 – 11 km SSE of AMS 13) and other tailings in the SE
where the majority of air samples originated from. The Mildred Lake settling basin is one of the oldest in
the region and is still actively being used; the correlation with $CH_4$ and VOC emissions is hence expected.
Component 1 is also associated with $NO_y$, rBC, CO, and HOA, though these correlations are relatively
modest (r = 0.27, 0.30, 0.41, and 0.40, respectively). These species typically originate from combustion
sources, such as generators, motor vehicles, including Diesel powered engines powering generators or
pumps; it is not obvious if and to what extent these are operated on or near tailings ponds, though.
Satellite observations have shown elevated concentrations of $NO_2$ above on-site upgrader facilities,
likely a result of emissions from extraction and transport sources (McLinden et al., 2012). In addition,
one of the major highways of the region is located adjacent to the Mildred Lake settling basin and other
major ponds in the region; highway traffic emissions (of CO, $NO_y$, rBC, and HOA) may hence also be
partially included in component 1.
The bivariate polar plot shows that component 1 was observed when local wind speeds were from the
SE and E of the measurement site (Fig. 5C), which is consistent with the notion that the Mildred Lake
settling basin and emissions along Highway 63 and, potentially, more distant facilities are sources
contributing to this component.





Component 1 is associated with the IVOC signature, though to a lesser degree than components 2 and 5.
The association of the IVOC signal with component 1 is slightly weaker (r = 0.31) than the association
with component 2 (r = 0.39), but significantly weaker than component 5 (r = 0.74). The association of
IVOCs with tailings ponds vapor can be explained by the presence of bitumen in the ponds that was not
separated from the sand during the separation stage (Holowenko et al., 2000). Tailings ponds contain
anywhere from 0.5% - 5% residual bitumen by weight (Chalaturnyk et al., 2002; Holowenko et al., 2000;
Penner and Foght, 2010). As illustrated in Fig. 4A, some of this material floats on the ponds' surfaces,
where IVOCs can partition to the air. Emission of IVOCs from bitumen floating on tailings ponds would
be a function of many variables (e.g., diluent composition, extraction methodology, settling rate,
temperature, etc.) and is thus not expected to be as persistent as $CH_4$ partitioning from the ponds to the
above air or from exposed bitumen on the mine surface, leading to a lower overall correlation.
Component 1 is also associated with the less oxidized oxygenated organic aerosol factor, LO-OOA (r =
0.45). Liggio et al. (2016) found that the observed secondary organic aerosol is dominated by an OOA
factor whose mass spectrum was similar to those of aerosols formed from oxidized bitumen vapours.
The organic aerosol budget in this study was also dominated by an OOA factor, the LO-OOA (Lee et al.,
In prep). The association of LO-OOA with component 1 is thus consistent with its association with IVOCs.
**4.1.2. Component 2: Mine fleet and vehicle emissions**
Component 2 strongly correlates with $NO_y$ (r = 0.82), rBC (r = 0.77), pPAH (r = 0.94), and HOA (r = 0.74),
which suggests a combustion source such as Diesel engines. In the AB oil sands, there is a sizeable off-
road mining truck fleet consisting of heavy aggregate haulers. In addition, there are Diesel engine
sources associated with generators, pumps and land moving equipment, i.e., graders, dozers, hydraulic
excavators, and electric rope shovels (Watson et al., 2013; Wang et al., 2016). Most of these non-road
applications have been exempt from highway fuel taxes, on-road fuel formulation requirements and



after-engine exhaust treatment (Watson et al., 2013). Emissions from the hauler fleet and the stationary
sources would fit the profile of component 2. Other Diesel engines operated in the region include a
commuter bus fleet, pickup and delivery trucks, tractor-trailers, and privately owned Diesel powered
automobiles used to commute from the work sites to the major residential areas around Fort
McMurray, whose emissions are likely captured by component 2 as well, though the magnitude of these
relative to the mining truck fleet is not known. Consistent with component 2 being associated with an
anthropogenic source is its weak correlation with undecane (r =0.27), likely arising from fugitive fuel
emissions.
The bivariate polar plot (Fig. 5D) for component 2 and $NO_y$ in particular (Fig. S-3A) match the location of
Highway 63 which crosses the river to the SE of AMS 13 and bends to the E and is indicative of a line
source. At the same time, some of the largest mining operations in the region, the Susan Lake Gravel Pit,
Aurora North, Muskeg river, and Millennium mines are located to the NE and SE of AMS 13 as well. $NO_y$,
rBC, and HOA (Fig. S-3A, B and D) all appear to have dominating point sources to the S and E when wind
speeds are 1-2 m $s^{-1}$. These directions are the same as the Fort McKay industrial park to the E and the
Syncrude Mildred Lake facility parking lot to the S which would have a higher concentration of vehicles
emitting these pollutants in a smaller area, whose emissions would be in addition to those from
industrial activities.
Component 2 is associated with the IVOCs signature and $CH_4$ (both r = 0.39). The mining activities bring
bitumen to the surface; similar to what we had observed in lab experiments (Fig. 2, black trace), the
surface exposure of bitumen during mining and on-site processing is expected to be associated with
fugitive emissions of $CH_4$ (Johnson et al., 2016) and IVOCs.
Fine-fraction particle-surface bound PAHs (pPAH) are associated strongly with component 2, but no
other components. Measurements of individual PAHs in snow and moss downwind from the oil sands



facilities have identified multiple sources of PAHs in the Athabasca oil sands, which include wind-blown
petroleum coke dust (also referred to as petcoke for short), a carbonaceous residual product from the
upgrading of crude petroleum that is stockpiled on mine sites, and emissions from fine tailings, oil sands
ore, and naturally exposed bitumen (Zhang et al., 2016; Jautzy et al., 2015; Parajulee and Wania, 2014).
Given this diversity of known sources, the associations of PAHs with only a single component is
surprising, though indicates that emissions from the mining fleet (which would include Diesel and,
perhaps, wind-blown emissions from petcoke that is being transported) gave rise to most of the
variability in surface-bound PAH concentrations in this data set. The petcoke emissions identified in the
studies mentioned above are likely mainly associated with larger, supermicron sized particles, whose
PAH content would not be detected by the pPAH measurement in this data set.
Component 2 is not significantly associated with LO-OOA ($r = 0.11$), even though IVOCs are associated
with this component. This feature may indicate that the IVOCs emitted in component 2 are qualitatively
different from those emitted by components 1 and 5, in that they are less likely to yield organic aerosol
on the time scale of transport from emission to observation. One reason for the difference could be that
the bitumen that is transported by the mining fleet is relatively freshly exposed, whereas the IVOCs
released by bitumen in tailings ponds has been processed by microbes and that released by mine faces
(component 5) may have been photochemically oxidized to a greater extent and hence more prone to
rapid aerosol formation.
There is little to no association of component 2 with either CO or $CO_2$ ($r = 0.18$ and $0.08$, respectively).
This is somewhat unexpected as the trucks are expected to release both (Wang et al., 2016) but could be
due to significantly larger $CO_2$ sources in the area dominating the observed $CO_2$ variability at AMS 13
(e.g., components 3 and 6).



### 4.1.3. Component 5: Surface-exposed bitumen and hot-water bitumen extraction

Component 5 correlates more strongly with the IVOCs (r = 0.74) than with any other component and correlates strongly with LO-OOA (r = 0.72), moderately with rBC (r = 0.44), and weakly with HOA (r = 0.25), $NO_y$ (r = 0.22), decane (r = 0.23), undecane (r = 0.20), and TRS (r = 0.26). We interpret this profile as emissions from surface-exposed bitumen which outgases IVOCs.

One possibility is that these emissions occur on mine faces, where previously unexposed bitumen is brought to the surface as a result of mining. Only a relatively small portion of the mine faces is actively mined; those parts give rise to rBC and $NO_y$ emissions from combustion engines in heavy haulers or generators powering equipment. The weak association of component 5 with TRS could be due to sulfur reducing bacteria found on the surface of bitumen. However, most of the variability of TRS at AMS 13 is attributed to composite or "dry" tailings ponds given their more conducive environment to microbial activity.

Component 5 does not correlate with $CO_2$ (r = -0.03) and only weakly with $CH_4$ (r = 0.12), which is somewhat at odds with the notion of mine faces as the main source of IVOCs. The mine faces give rise to substantial fugitive emissions of $CO_2$ and $CH_4$ (Johnson et al., 2016) – these emissions are likely captured by component 6 in this analysis (see S.I.). It is unclear to what extent these greenhouse gases are released relatively quickly from "hot spots" (i.e., from a small number of locations) through surface cracks and fissures or by slow release from new material that is exposed and then releases greenhouse gases during material handling, transport and processing (Johnson et al., 2016). IVOCs from surface-exposed bitumen are likely released by the latter mechanism and are temperature-dependent. If the mine faces are indeed the main IVOC source, the analysis results presented here suggest that the IVOCs emissions from surface-exposed bitumen on mine faces are decoupled from $CH_4$ emissions in time and appear as a distinct component and hence corroborate the "hot spots" or fast release hypothesis,



though clearly, more work is needed to characterize greenhouse gas emissions from oil sands mine
faces.
The association of IVOCs with component 5 may also be a result of fugitive emissions during the hot
water-based extraction of bitumen sand slurries during the separation phase of bitumen treatment.
Generally, bitumen is extracted in a weak alkaline environment by aeration of the solution to optimize
the separation of sand and bitumen (Masliyah et al., 2004). Unrecovered bitumen and naphtha then end
up in tailings. The recovered bitumen and naphtha are moved to upgrader facilities where they undergo
further treatment (such as coking or hydrotreatment). The magnitude of fugitive emissions during these
downstream extraction processes could be large, considering the bitumen is heated and actively
aerated. Future work should investigate IVOC fluxes near extraction plants and on mine faces.
Finally, it is conceivable that a "natural" background of IVOCs exists in the region (since bitumen can be
found at or near the surface in many parts of the region); such a natural background would also be
included in component 5. However, this "natural" bitumen would have been exposed at the surface for
geological time scales and, unlike unexposed, buried bitumen, likely would have lost most of its volatile
content over that period. Furthermore, the mine faces occupy large swaths of land in the region (as
evident from satellite imagery). Thus, the IVOCs emissions are more likely due to anthropogenic activity
than due to a natural phenomenon.

**4.2. Sources not associated with IVOCs**
**4.2.1. Component 3: Biogenic emissions and respiration**
Component 3 is strongly correlated with the monoterpenes α-pinene (r = 0.98), ß-pinene (r = 0.98) and
limonene (r = 0.92) and is hence identified as a biogenic emissions source. This component is also



moderately associated with $CO_2$ (r = 0.48).
At AMS 13, $CO_2$ and the monoterpenes exhibit a very similar diurnal cycle: they are present in higher
concentrations during the night than during the day (Fig. 3) due to a decrease in the boundary layer
height (BLH) at night coupled with plant respiration of $CO_2$ and non-photochemical emission of
monoterpenes (Fares et al., 2013; Guenther et al., 2012). During the day, mixing ratios of $CO_2$ are lower
due to plant uptake and photosynthesis, and mixing ratios of terpenes are lower due to higher mixing
heights and vertical entrainment and due to oxidation by $O_3$ and OH (Fuentes et al., 1996). Hence, the
PCA gives a *positive* correlation of monoterpenes with $CO_2$ even though the physical processes,
photosynthesis and respiration, work in opposite direction.
The bivariate polar plots (Fig. S-4A-C) show that the monoterpenes and $CO_2$ were observed in highest
concentrations when the wind speeds were low (< 1 m s$^{-1}$), consistent with formation of a stable
nocturnal boundary layer.
To corroborate this interpretation, the PCA was repeated with BLH estimated by a light detection and
ranging (LIDAR) instrument (Strawbridge et al., in prep.) added as a variable (Table S-9 in the S.I.). Since
BLH is not "emitted" by any source, it appears as a single variable component (r = 0.90). The only other
component that BLH (anti)correlates with is the biogenic component 3 (r = -0.35).
The dominant monoterpene species observed was α-pinene, followed by ß-pinene and limonene,
though occasionally there was twice as much ß-pinene than α-pinene in the sampled air. Some
variability of this ratio is expected since emission factors vary considerably between tree species (Geron
et al., 2000) which are not homogeneously distributed throughout the region (e.g., Fig. S1 of Rooney et
al. (2012)).
Simpson et al. (2010) observed enhancements of α-pinene and, to a greater extent, β-pinene over the
oil sands (up to 217 pptv and 610 pptv) compared to background levels of 20±7 and 84±24 pptv,





respectively, during mid-day overflights (which occurred between 11:00 and 13:00 local time). Similar
enhancements were also reported by Li et al. (2017) who observed emissions of biogenic hydrocarbons
in the four facilities sampled, three of which showed a higher β- than α-pinene concentration. The PCA
analysis (Table 5) showed no significant correlation of α- and β-pinene with any of the anthropogenic
sources, which implies that the biogenic source strength is simply too large for any anthropogenic
emissions of terpenes to be picked up in the analysis, especially considering that terpenes are relatively
short-lived.
The biogenic source shows weak anticorrelations with $NO_y$ (r = -0.26), $NH_3$ (r = -0.24), and $SO_2$ (r = -0.15).
Many $NO_y$ species (i.e., $NO_2$, HONO, peroxycarboxylic nitric anhydrides or PAN, and $HNO_3$) and $SO_2$
deposit to the forest canopy (Hsu et al., 2016; Min et al., 2014; Fenn et al., 2015); at night, when mixing
heights are lower, their concentrations are expected to decrease faster than during the day and are thus
out of phase with the $CO_2$ and terpene concentrations. In addition, there is a time-of-day observation
bias for $SO_2$ and, to lesser extent, $NO_y$, which are found in upgrader plumes (see 4.2.2.). The weak
anticorrelation with $NH_3$ likely arises because the $NH_3$ emissions from plants are mainly stomatal and
scale with temperature and are hence larger during the day than at night, anticorrelated with the
terpene source (Whaley et al., 2017).

**4.2.2 Component 4: Upgrader emissions**

Component 4 is strongly correlated with $SO_2$ (r = 0.97) and total sulfur (r = 0.93). By far the largest
source of $SO_2$ in the region are upgrader facilities, which emit as much as $6 \times 10^7$ kg annually according to
emission inventories (ECCC, 2013). Significant $SO_2$ emissions from upgrader facilities have recently been
confirmed by aircraft studies (Simpson et al., 2010; Howell et al., 2014; Liggio et al., 2016). Component 4
is also weakly correlated with $NO_y$ (r = 0.21) but not with rBC (r = 0.05), consistent with a non-sooty (i.e.,
lean) combustion source such as upgrader stacks. Strong enhancements in $SO_2$ were only observed



intermittently as "spikes", which is expected when sampling emissions from relatively few and discrete
point sources.
Component 4 is weakly anticorrelated with $CO_2$ (r = -0.12), even though inventories indicate that the
upgrading facilities are the largest $CO_2$ source in the region (Furimsky, 2003; Englander et al., 2013; Yeh
et al., 2010). In this data set, the lack of correlation of component 4 with $CO_2$ (and to some extent with
$PM_{10-1}$ as well) likely arises mainly from a sampling bias as stack emissions were only observed during
daytime, likely due to diurnal variability of the atmospheric boundary layer structure as explained
below.
Most of the variability in $CO_2$ concentration at AMS 13 is due to surface-based sources that originate
from large areas, especially biogenic processes (photosynthesis during the day and respiration at night,
component 3) and anthropogenic surface sources such as those captured by component 6 (section
4.2.3). Other anthropogenic pollutants, such as $SO_2$, $NO_y$, and $CH_4$, are not subject to large biogenically
driven processes and are less affected than $CO_2$.
In contrast to surface sources, emissions from the > 100 m tall stacks are comparatively under sampled
and observed mainly during daytime, when vertical mixing brings elevated plumes to the surface, yet
$CO_2$ concentrations are generally much lower than during the night due to uptake by vegetation. At
night, pollutants emitted from stacks are injected above the likely very shallow nocturnal surface layer
and were hence not observed at the surface. Vertical profile measurements of $SO_2$ stack plumes by a
Pandora spectral sun photometer at Fort McKay during daytime have shown considerable vertical
gradients and only occasional transport of $SO_2$ all the way to the surface (Fioletov et al., 2016).
The association of component 4 with $CO_2$ is negative because the stack emission source is observed only
during the day when the large biogenic sink dominates and effectively masks the relatively small
increase due to anthropogenic $CO_2$. In contrast, background concentrations of $SO_2$ are comparatively





low, and the increase in $SO_2$ concentrations is readily picked up the PCA.
It would be interesting to conduct a future study in winter when biogenic activities decrease; a
wintertime PCA analysis of surface measurements might be able to associate $CO_2$ enhancements with
upgraders, though boundary layer mixing heights would decrease as well, which would make a PCA
analysis using surface data even more challenging.
Component 4 does not correlate with $PM_{10\text{-}1}$ volume (r = 0.09). It is clear that the emitted $SO_2$ will
contribute to secondary aerosol formation downwind, such that a correlation of stack emissions with
$PM_{10\text{-}1}$ volume might be expected. However, these secondary contributions will likely mostly be in the
submicron aerosol fraction, which adds relatively little to $PM_{10\text{-}1}$ volume. Further, $PM_{10\text{-}1}$ volume is
dominated by coarse particles from other primary sources, mostly wind-blown emission of sand from
the mine surfaces, roadways and, perhaps, bioaerosol (component 7, see S.I.). These effects make $PM_{10\text{-}1}$
volume from stacks appear comparatively small, such that the variability of the larger, surface-based
sources likely masks the contribution of stacks emissions to $PM_{10\text{-}1}$ variability.
The bivariate polar plot of component 4 (Fig. S-5D) shows that the largest magnitudes were observed
when local winds were from the SE. The corresponding plot of $SO_2$ (Fig. S-5A) reveals two more distinct
sources: a larger one from the E and a smaller one from the SSE. However, only two facilities (Sunrise
and Firebag) are located to the E at relatively large distances of 37 km and 47 km respectively. The
largest known upgraders and $SO_2$ sources in the area (i.e., upgraders located at the Mildred Lake and
Suncor base plants) are located to the S and SE of AMS 13. Considering that the stack emissions are only
observed intermittently, we speculate that there exists a mesoscale transport pattern in the Athabasca
river valley which channel emissions, such that the local wind direction and speed may be misleading as
to the true location of these sources. For more extensive data sets, such phenomena may very well
average out but perhaps did not in this case.



**4.3. Extended PCA with added secondary variables**
The extended analysis (Table 7) qualitatively preserves the structure (with the exception of an added
"Aged" component, # 6) of the original 10-component solution but allows an assessment of which
components most result in formation of secondary products such as SOA, which has implications for
health (Bernstein, 2004 ) and climate (Charlson et al., 1992). Secondary products vary considerably as a
function of air mass chemical age (which depends, amongst other components, on time of day and
synoptic conditions, including wind speed) and are hence expected to add considerable noise and
scatter to the results leading to lower correlations. On the other hand, the distance between the
measurement site and sources is fixed, such that this variability should average out over time. This
indeed appears to have happened in this data set in spite of the relatively low sample size.
The analysis indicates that the strongest IVOC source (Component 5) has the largest impact on $PM_1$
(Table 7).  Aircraft measurements combined with a modelling study have required a group of IVOC
hydrocarbons to explain the significant SOA formation and growth downwind of the oil sands region
(Liggio et al., 2016). The association of IVOCs with $PM_1$ volume is consistent with the hypothesis that
oxidation of IVOCs observed at AMS 13 leads to SOA generation and appears to have a significant impact
on the variation in $PM_1$ mass.
The second component influencing $PM_1$ is that from stack emissions (Component 4 in the primary PCA;
Component 2 in the secondary PCA) (Tables 5 and 7). It is well established that the oxidation of $SO_2$ to
sulfate will lead to formation of fine particulate matter. This apparently occurs, at least partially, on the
time scale between the point of emission and the AMS 13 site (assuming a wind speed of 3 m/s and a
distance of 11 km, the transit time is 1 hour), though some fraction of $SO_4^{2-}(p)$ is likely directly emitted.



**5. Summary and conclusions**
A PCA was applied to continuous measurements of 22 primary pollutant tracers at the AMS 13 ground
site in the Athabasca oil sands during the 2013 JOSM intensive study to elucidate the origins of airborne
analytically unresolved hydrocarbons that were observed by GC-ITMS. The analysis identified 10
components. Three components correlated with the IVOC signature and were assigned to mine faces
and, potentially, hot-water bitumen extraction facilities, the mine hauler fleet, and wet tailings ponds
emissions. All three are anthropogenic activities that involve the handling of raw bitumen, i.e., the
unearthing, mining and transport of crude bitumen, and the disposal of processed material that contains
residual bitumen in wet tailings ponds. The PCA results are consistent with our previous interpretation
that the unresolved hydrocarbons originate from bitumen, which was based on the similarity of the
chromatograms with those obtained in a head space vapor analysis of ground-up bitumen in the
laboratory.
Liggio et al. (2016) showed that these hydrocarbons constitute a group of IVOCs in the saturation vapor
concentration ($C^*$) range $10^5$ µg m$^{-3}$ < $C^*$ < $10^7$ µg m$^{-3}$ that contribute significantly to secondary organic
aerosol formation and growth downwind of the oil sands facilities. The correlation of LO-OOA with 2 of
the 3 IVOC components in the main PCA analysis and with $PM_1$ in the extended analysis corroborates the
high SOA formation potential of IVOCs and suggests that further differentiation may be needed and
stresses the need for IVOCs to be routinely monitored. In particular, direct measurements of emissions
throughout the processing of raw bitumen are needed to pinpoint source contributions more accurately
and aid in the development of potential mitigation strategies.
The PCA analysis in this study suffered from a several limitations. For instance, PCA does not provide
insight into the magnitudes of emissions, though it does capture what conditions change ambient
concentrations the most. Further, the receptor nature of PCA did not always discern between large



source areas that may have many individual point sources coming together at the point of observation.
For example, component 1 contains an obvious tailings pond signature because of its high correlations
with anthropogenic VOCs, methane and TRS, but also includes several combustion sources, making
interpretation of this IVOC source location more challenging. A longer continuous data set with a greater
number of variables would have perhaps been able to resolve these different sources, including the
various tailings ponds, of which there are 19 in the region, all with slightly different emission profiles
(Small et al., 2015) .
Another limitation is the bias of this (and most) ground site data set towards surface-based emissions
and the undersampling of stack emissions. Facility stacks were only observed in the daytime because at
night the mixing height is so low that the stacks are emitting directly into the residual layer. These
emissions could be quantified using aircraft based platforms (Baray et al., submitted, 2017; Howell et al.,
2014; Li et al., 2017). The PCA struggled most with the allocation of greenhouse gases. Mixing ratios of
$CO_2$, in particular, were difficult to reconcile in this analysis due to a high background and large
attenuation by biogenic activity and boundary layer meteorology. Forests greatly affected $CO_2$ levels in
the region because it is taken up during the day when plants are photosynthetically active and emitted
at night when plants undergo cellular respiration. This $CO_2$ source and sink appears to dominate the
PCA, effectively masking relatively small emissions from tailings ponds, facilities, and tail pipes in
particular from the mine hauling fleet.
Finally, there is a need for improved monitoring methods for IVOCs. For instance, future studies should
focus on characterizing the VOCs in the above mentioned volatility range using a greater mass and time
resolution instrument, such as a time-of-flight mass spectrometer (TOF-MS) or higher resolution
separation methods (e.g., multi-dimensional gas chromatography).





**Acknowledgments**
Funding for this study was provided by Environment and Climate Change Canada and the Canada-
Alberta Oil Sands Monitoring program. The GC-ITMS used in this work was purchased using funds
provided by the Canada Foundation for Innovation and matching funds by the Alberta government.
TWT, JAH, DKB, FVA and GRW acknowledge financial support from the Natural Sciences and Engineering
Research Council of Canada (NSERC) Collaborative Research and Training Experience Program (CREATE)
program Integrating Atmospheric Chemistry and Physics from Earth to Space (IACPES).



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
