# Peer review of "Principal component analysis of summertime ground site measurements in the Athabasca oil sands: Sources of IVOCs"

_Atmospheric Chemistry and Physics, 2017_

## Referee Comment (RC1) · Anonymous Referee #1 · 27 Feb 2018

This manuscript measured a variety of air pollutants at a surface site in the oil sands region in summer 2013. PCA approach was used to elucidate major sources of these air pollutants, in particular, IVOCs. Using the 95% cumulative percentage of variance criterion, ten components were categorized, among which 3 components correlated with IVOCs, most likely related to the unearthing and processing of raw bitumen. The authors also found the association of secondary variables with the primary components, implying rapid formation of these secondary products on a time scale that is similar to the transit time of the pollutants to the measurement site. The study was specifically carried out in an oil sand region, which constrained the extension of the implication of the findings to other regions. The PCA with different number of sources was tested,

and the identification of source types was discussed and elucidated in detail. Though this manuscript was for a special issue of ACP, this manuscript still needs significant improvement, and could only be considered for potential publication after the following major concerns are well addressed. 1) The aim of this study was to identify IVOC sources. However, none of the peaks of IVOC species in Ion Chromatograms was identified, which subsequently biased or even wronged the identification of the IVOC sources. 2) It is doubtful that CO was mainly generated from VOC oxidation given many industrial sources in the oil sand region. 3) The criteria for the degree of correlations, i.e., correlation coefficient r value, are unclear. r= 0.3 indicated very poor correlation to me. Specific comments: 1. Page 9, Figure 2. The gray area showed unsolved HCs in the volatility range of C11-C17 with saturation vapor concentration from 105 $\mu$g m-3< C* < 107 $\mu$g m-3. It is not clear how the retention time for C11, C12, ..., C16 was obtained in the figure. The marked retention time for C11-C16 n-alkanes is actually doubtful, because for n-alkanes IVOCs, the distance between C11 and C12, C12-C13, C13-C14, ...C15-C16 in the chromatograms should be generally equal. Also, their retention time in Figure 2 seems too high (25 min later) as they usually appear at ~10-15 mins (of course it depends on the methods and columns used). Furthermore, why were some IVOC species not identified (even could not be quantified) when this study was designed because the focus of this manuscript is about IVOC sources? If the focus was based on something unknown, the speculation could be very wrong. In this study, the headspace sample of bitumen indeed showed an unsolved signal in the IVOC range, but this does not mean bitumen is definitely the IVOC source in this study because n-alkanes IVOCs have many other sources such as vehicular emissions, biogenic IVOCs (e.g. sesquiterpenes) and petroleum enterprises. In fact, there are mature analytical methods such as TAG-GC-ToF-MS for gas- and particle-phase IVOCs, SVOCs and SOA tracers. 2. I have to admit that the PCA results in this study were quite impressive no matter how many factors were extracted (5-11 factors) because it is common that PCA often gives collocated factors and it is very difficult to identify 10 or above individual sources with this method, especially when 22-28 variables were input. Certainly

more than 200 samples as input could enhance the PCA extraction. 3. Page 12, Table 3. 4 AVOC and 3 BVOC species were quantified. What were the reasons to select these specific VOC species in this study, due to the limit of GC-ITMS or selection of tracers? In particular, isoprene was not measured (used) in this study as its level in the atmosphere is usually much higher than other BVOCs. 4. Lines 284-289 stated that $CH_4$ and $CO_2$ originated from distinct sources while both correlated well with Factor 6 in Table 5, indicating their similar sources/patterns. They are contradictory. Please clarify it. 5. Table 5 claims that coefficients with Pearson correlation coefficients r>0.3 are interpreted as being moderately or strongly associated with a component, which I do not agree. What are the criteria to select r>0.3 to indicate moderate or strong association? To me, r value around 0.3 means very poor correlation. At least r value should be >0.5 to indicate somewhat correlations. 6. Page 24, Table 6. Component 5 was identified as "surface exposed bitumen and hot-water based bitumen extraction". This is only based on the headspace sample of bitumen without any identification of IVOC species. Could it be other sources? If TAG-GC-ToF-MS was used, this source could be better identified. Furthermore, IVOCs correlated well with LO-OOA (Table 5), implying that some IVOCs might be secondarily formed or the PCA results were collocated. Please comment on these. 7. Page 24, Table 6. Component 9 "enhancement of CO" was categorized as "VOC oxidation". This is questionable. Why? It is apparent that on some days the nighttime CO was quite high, which should not be formed via VOC oxidation due to the fact that nighttime VOC oxidation chemistry is weaker than daytime photochemistry. Moreover, if CO was from VOC oxidation, it would destroy the correlations among VOCs (due to various photochemical reactivity of VOC species), leading to poor factor loadings, while the correlations among AVOCs and BVOCs were strong in Table 5. Though CO did not have correlations with other combustion tracers, could it be caused by the weakness of the PCA method, or the correlation of CO with other combustion tracers is the "must" for the identification of a combustion souce? 8. Page 25, Figure 4. (B) (component 2) vehicles emit aromatics and n-alkanes but component 2 did not in Table 5. (D) (component 4) Upgrader facilities emitted TS and

SO2 only? There was flare stack. Were there no VOC emissions (Table 5)? 9. Page 26, lines 382-385. Not ture. r=0.27 only for Ox in component 5! Moreover, why didn't Ox have high correlations with any variables in Table 7? 10. Page 43, lines 718-720. Could any measurements be done downwind to verify this hypothesis? 11. Page 44, lines 740-742. This also fits CO. But no discussion at all. 12. Page 44, lines 749-750. TAG-GC-ToF-MS technique has already been applied to identify and quantify many IVOC species. Many papers have been published in the society. In the "Supplement" document 13. Lines 54-55. Again, what are the selection criteria for these specific VOCs? Can GC-ITMS measure more VOCs such as C2-C10 HCs and so on for better source identification? 14. Line 330. No solid evidence for this. 15. Figs S2-S4, S7-S11 caption: Table 4 should be Table 5 in the text? 16. Figure S5 caption: Table 4 should be other Table in the text? 17. Figure S6 caption: Table 4 should be Table 5 in the text?

---

## Referee Comment (RC2) · Anonymous Referee #2 · 24 Apr 2018

This study investigates the sources of IVOCs in the Athabasca oil sands by applying PCA on air pollutants measured at a ground site. IVOCs have been indicated as an important class of SOA precursors. Identification of the major sources of IVOCs is needed in order to make effective measures to reduce their emissions. The objective of this study is interesting, but the data and presentation are too broad, lack of the focus on IVOCs given that the term of "IVOCs" is highlighted in the title.

Measurements of IVOCs were carried out using GC-ITMS. Atmospheric IVOCs are composed of both primary IVOCs, dominated by hydrocarbons, and oxygenated IVOCs, oxidation products of primary IVOCs and VOCs. The elution of oxygenated

[Figure]

IVOCs from the GC column is likely incomplete. However, in this study, the split between hydrocarbon-IVOCs and oxygenated-IVOCs was neither performed nor discussed. In addition, the collection efficiency and recovery of IVOCs was not described, either. In the light of the description of chemical analysis of IVOCs provided in this study, it is unclear whether the measurements capture the variability of atmospheric IVOCs, which is critical for PCA.

This manuscript mentioned that PCA is unable to determine the fractional contributions of sources to IVOCs in the section of "5. Summary and conclusions". Why is the PMF analysis not applied in this study? The number of samples is sufficient to provide a robust solution.

As mentioned above, ambient IVOCs includes oxygenated IVOCs, oxidation products of IVOCs and VOCs. However, the PCA in this study did not identify a component for oxygenated IVOCs.

Liggio et al. (2016) finds that the evaporation and atmospheric oxidation of low-volatility organic vapors from mined oil sands is directly responsible for the majority of observed SOA mass. Does this mean that the contribution of the component 2 to IVOCs shall be small given that this component is likely related to "Mine fleet and vehicle emissions". More discussion is needed.
* * *

---

## Referee Comment (RC3) · Anonymous Referee #3 · 5 May 2018

This paper describes principle component analysis (PCA) of ambient air quality data set collected in the Canadian oil sands region over a month or so in the summer of 2013. The data collection was part of a large field project to characterize the impact of oil sand operations on local air quality and climate. This is interesting question given the magnitude of the facilities. Recent work suggests dramatic secondary organic aerosol (SOA) formation downwind of these facilities. This SOA formation was attributed to emissions of low volatility organics, IVOCs. This was described in the Nature paper of Liggio et al. 2016.

This paper is focused on better understand the sources of IVOCs. This is an important

question that is of interest to readers of ACP. They have done this by performing PCA on an ambient dataset of mainly traditional pollutants (a handful of anthropogenic and biogenic VOCs plus other species). They decompose the data into 10 or 11 factors (depending on if they are looking at secondary species). Three of the factors have some association with IVOCs. The paper contains some relatively qualitative description of the sources of the factors that generally make sense. Although I am interested in this topic, after reading it I did not find the paper particularly interesting and did not feel like it made much of contribution to our understanding of the sources. I felt like going in the authors' attributed the IVOCs to bitumen. It was not clear that the paper did anything to support that hypothesis. There is nothing "wrong" with the paper, it is well written and has long descriptions. The paper would be much more interesting if it could quantitatively attribute IVOCs to sources (even the discussion of components seemed pretty speculative and qualitative).

Larger comments

The paper uses a criteria of r>0.3 to indicate moderate correlation. I view 0.3 as almost no correlation. I think the threshold should be much higher.

Presumably the GC-ITMS technique used to measure the VOCs and IVOCs could measure a much broader suite of compounds (or even break the chromatogram into subareas) that might yield more insight into what the sources of the IVOCs.

Specific comments

Figure 1 – Including a wind rose for the study period as an inset panel on this figure would be very helpful.

Line 125 – Please add one or two sentences on calibration and QA procedures.

Table 2. Who operated instruments is not that interesting. Time resolution would be more useful.

Line 164 – What is the recovery and calibration of the IVOCs?

Line 198 – What fraction of the data are below LOD?

Line 216 – I did not wade through the other solutions. Text on line Line 353 suggests they do not material change conclusions – maybe state that here.

Line 345 – Moderately associated with IVOC (r=0.31) versus weakly associated with rBC (r=0.3). All these values seem very weak (to almost no) correlation.

Figure 5. Add labels of directions to different facilities labeled in earlier table.

Line 498 (and other locations) – diesel should not be capitalized.

Line 529 – This is not total PAH but particle bound PAH. Seems like surprising they are associated with combustion and not other components (e.g. 1 or 5).

Line 544 – But diesel engines are not a major source of CO (gasoline engines are).

Line 552 – Associating the IVOCs bitumen seemed to be the hypothesis going in. It is not clear how this analysis reinforces or tests that hypothesis. It seems pre-conceived and they are just interpreting the data that way.

Line 695 – What is the evidence for this claim?

---

## Author Comment (AC1) · 15 Jun 2018

We thank the reviewers for their time and effort reviewing this manuscript. All reviewer comments are reproduced below in **bold, italicized font**. Our responses are shown in regular font. Changes to the text are indicated as underlined text for insertions or are  for deletions. Line numbers given below are for the revised version with all markups shown. We numbered the reviewer comments for easier cross-referencing.

Anonymous Referee #1
*This manuscript measured a variety of air pollutants at a surface site in the oil sands region in summer 2013. PCA approach was used to elucidate major sources of these air pollutants, in particular, IVOCs. Using the 95% cumulative percentage of variance criterion, ten components were categorized, among which 3 components correlated with IVOCs, most likely related to the unearthing and processing of raw bitumen. The authors also found the association of secondary variables with the primary components, implying rapid formation of these secondary products on a time scale that is similar to the transit time of the pollutants to the measurement site. The study was specifically carried out in an oil sand region, which constrained the extension of the implication of the findings to other regions. The PCA with different number of sources was tested, and the identification of source types was discussed and elucidated in detail.*

*Though this manuscript was for a special issue of ACP, this manuscript still needs significant improvement, and could only be considered for potential publication after the following major concerns are well addressed.*

*1) The aim of this study was to identify IVOC sources. However, none of the peaks of IVOC species in Ion Chromatograms was identified, which subsequently biased or even wronged the identification of the IVOC sources.*

The goal of this manuscript was to identify sources of IVOCs, but not to identify the compounds that make up the IVOCs. The identities of individual IVOCs do not need to be known to determine their origin as a group, through PCA, and does not imply, as the reviewer suggests, that our analysis was biased or wrong.

We agree with the reviewer, however, that identification of individual IVOC may provide useful information; however, this was not possible for this measurement campaign, since the IVOCs did not resolve on the chromatographic column. The more important feature is the volatility range of these species, which was constrained based on their retention times.

We note on line 779 that "future studies should focus on characterizing the VOCs in the above mentioned volatility range using a greater mass and time resolution instrument, such as a time-of-flight mass spectrometer (TOF-MS) or higher resolution separation methods (e.g., multi-dimensional gas chromatography)".

No changes were made in response to reviewer's comment.

**2) It is doubtful that CO was mainly generated from VOC oxidation given many industrial sources in the oil sand region.**

We apologize for the misunderstanding, which was probably triggered by the entry in Table 6 and the discussion of component 9 in the S.I.

The main source of CO in the atmosphere is undoubtedly the incomplete oxidation of hydrocarbons, to which a variety of sources contribute – e.g., fuel combusted in car engines, generators, forest fires, etc. or VOCs oxidized photochemically. For the oil sands region, it is, as Marey et al. (2015) state, generally "assumed that CO spatial variations in northern Alberta are associated with oil sands industrial activities and forest fires". We certainly agree that in the vicinity of large industrial sources such as the AOS industrial sources ought to be the largest source of CO. Our analysis suggests that factor 9 is dispersed over a wide geographical area, which could be due to a combination of numerous CO point sources (generators, pump stations, motor vehicles, etc.), i.e., the size of the geographical area covered by oil sands operations, but could also include the oxidation of biogenic and anthropogenic VOCs. Accordingly, we state in the main text that "Carbon monoxide is a tracer of biomass burning and fossil fuel combustion, in particular in automobiles with poorly performing or absent catalytic converters, but is also a byproduct of the oxidation of VOCs, in particular of methane and isoprene which are oxidized over a wide area upwind of AMS 13 (Miller et al., 2008)."

The assignment to component 9 to a particular source is somewhat tentative as noted on lines 378-380 that "components 6 through 10 are somewhat tentative as many (i.e., 7 – 9) are single variable components and have eigenvalues close to or below unity, i.e., account for less variance than any single variable."

In response to the reviewer's comment, we have modified the text as follows:

In Table 6, component 9 (Enhancement of CO) now lists as possible source(s) "incomplete hydrocarbon oxidation" instead of "VOC oxidation".

We have also modified the paragraph starting on line 376 as follows:

"Components 1 through 4 emerged regardless of the number of components used to represent the data, whereas the structure of components 5 through 10 only fully emerged in the 10-component solution (see S.I.). Hence, components 6 through 10 are somewhat tentative as many (i.e., 7 – 9) are single variable components and have eigenvalues close to or below unity, i.e., account for less variance than any single variable. As a result, the interpretations of these components are subject to more uncertainty and are more speculative but are presented in the S.I. for the sake of completeness and transparency. For the purpose of this manuscript, this is inconsequential as components 6 – 10 are not associated with IVOCs."

We have also slightly modified a paragraph in the S.I. (starting on line 366):

"Component 9: Incomplete hydrocarbon oxidation
Component 9 is another single variable component and strongly correlates with CO (r = 0.87). The variables with the next largest correlation coefficients are $CH_4$ (r = 0.17), 1,2,3- and 1,2,4-TMB (both r = 0.18), and o-xylene (r = 0.16).

The conventional interpretation of CO is as a byproduct of incomplete hydrocarbon oxidation, as it is found in fossil fuel combustion exhaust or in biomass burning plumes. Component 9, however, is not associated with $NO_y$ (r = -0.08) or $CO_2$ (r = 0.05), which rules out this conventional interpretation. Recently, Marey et al. (2015) examined the spatial distribution of CO in Northern Alberta using a combination of satellite and ground station data and found that most CO is derived from anthropogenic sources, biomass burning and the photochemical oxidation of methane and other VOCs."

**3) The criteria for the degree of correlations, i.e., correlation coefficient r value, are unclear.  r= 0.3 indicated very poor correlation to me.**

The criteria for the degree of correlations were stated on lines 362-364 in the text: 'Associations with r>0.7, r>0.3, and r>0.1 are referred to as "strong", "moderate", and "weak", respectively.'
We chose a cut-off value of 0.3 because of the non-negligible correlations of IVOCs with components 1 and 2 (Table 5). The labels "weak, "strong", and "moderate" were used for the purpose of keeping the discussion consistent when referencing degrees of correlation.

In light of the reviewer's concern (which was echoed by reviewer 3), we have modified the text, starting with the sentence on lines 362-364, as follows:

'Associations with r>0.7, r>0.3, and r>0.1 are referred to as "strong", "weak", and "poor", respectively.'

**Specific comments:**

**1a. Page 9, Figure 2. The gray area showed unsolved HCs in the volatility range of C11-C17 with saturation vapor concentration from 105 _g m-3< C* < 107 _g m-3. It is not clear how the retention time for C11, C12, ..., C16 was obtained in the figure.**

Our apologies – this important experimental detail should have been provided. The retention times were obtained by analyzing a standard hydrocarbon mixture with the same temperature program and carrier gas flow rate as deployed in the field. We have added the following to the figure caption:

"Figure 2. (Top) .... (Bottom) Retention times of n-alkanes, determined after the measurement intensive by sampling a VOC mixture containing a $C_{10} - C_{16}$ n-alkane ladder."

**1b. The marked retention time for C11-C16 n-alkanes is actually doubtful, because for n-alkanes IVOCs, the distance between C11 and C12, C12-C13, C13-C14, ... C15-C16 in the chromatograms should be generally equal. Also, their retention time in Figure 2 seems too high (25 min later) as they usually appear at 10-15 mins (of course it depends on the methods and columns used).**

The reviewer is assuming that the GC was operated using a single temperature ramp. However, we used a somewhat unusual temperature program that resulted in uneven spacing of the n-alkanes; the temperature program is described in the S.I. on line 87:

"The GC oven was programmed as follows: hold at 40° C for 3.00 min, heat at 1.5° C min$^{-1}$ to 70° C, heat at 5° C min$^{-1}$ to 200 °C and hold for 4 min (total 53.00 min)."

We have modified this description slightly as follows:

"The GC oven was programmed as follows: hold at 40 °C for 3.00 min, heat at 1.5 °C min$^{-1}$ to 70° C (reached at 23.00 min), heat at 5° C min$^{-1}$ to 200 °C (reached at 49.00 min) and hold for 4 min (total 53.00 min)."

No further changes were made in response to reviewer's comment.

***1c. Furthermore, why were some IVOC species not identified (even could not be quantified) when this study was designed because the focus of this manuscript is about IVOC sources? If the focus was based on something unknown, the speculation could be very wrong.***

Please see our response above to major comment #1 above.

***1d. In this study, the headspace sample of bitumen indeed showed an unsolved signal in the IVOC range, but this does not mean bitumen is definitely the IVOC source in this study because nalkanes IVOCs have many other sources such as vehicular emissions, biogenic IVOCs (e.g. sesquiterpenes) and petroleum enterprises.***

The reviewer is correct that IVOCs may originate from all of these sources. However, our analysis (see Table 5) suggests that the IVOCs in this study originated mainly from a standalone component (#5), were not associated with a biogenic source (#3) and only very weakly with vehicular sources (#2). Since these associations are described and discussed at length in the manuscript, no changes were made in response to reviewer's comment.

***1e. In fact, there are mature analytical methods such as TAG-GC-ToF-MS for gas- and particle-phase IVOCs, SVOCs and SOA tracers.***

This is correct. However, identification of individual compounds that make up the IVOC signature was not the goal of this work, nor was it possible with the instrumentation on hand during the time of the study. We agree that future TAG-GC-ToF-MS measurements at this site would be useful.
In response to the reviewer's comment, we modified a paragraph in the conclusion section:

"Finally, there is a need for improved monitoring methods for IVOCs. For instance, future studies should focus on characterizing the VOCs in the above mentioned volatility range using a greater mass and time resolution instrument, such as a time-of-flight mass spectrometer (TOF-MS), higher resolution separation methods (e.g., multi-dimensional gas chromatography), and also include measurement of speciated aerosol organic composition by, for example, thermal desorption aerosol GC (TAG) analysis (Williams et al., 2006). "

*2a. I have to admit that the PCA results in this study were quite impressive no matter how many factors were extracted (5-11 factors) because it is common that PCA often gives collocated factors and it is very difficult to identify 10 or above individual sources with this method, especially when 22-28 variables were input.*

We agree that this is a somewhat unusual, though note that the sources in this region are somewhat unusual as well. In support of the number of components extracted, all solutions are shown in the supplementary information for this reason.

No changes were made in response to this comment.

*2b. Certainly more than 200 samples as input could enhance the PCA extraction.*

We agree. Unfortunately, we are limited by the temporal resolution of the various instruments on site, data outages, etc. A longer field study over an entire season would be very beneficial for this type of approach. Since we already state on line 764 that "A longer continuous data set with a greater number of variables would have perhaps been able to resolve these different sources", that is, additional components, no changes were made to the manuscript in response to the reviewer's comment.

*3. Page 12, Table 3. 4 AVOC and 3 BVOC species were quantified. What were the reasons to select these specific VOC species in this study, due to the limit of GC-ITMS or selection of tracers? In particular, isoprene was not measured (used) in this study as its level in the atmosphere is usually much higher than other BVOCs.*

As stated on lines 183-185, we included **all** quantified non-methane hydrocarbons in the analysis. We agree that inclusion of a larger number of hydrocarbons would have been advantageous; however, the GC-ITMS was the only time-resolved VOC instrument on site that reported final data. Isoprene was not on this list.

The manuscript was not changed in response to the reviewer's comment.

*4. Lines 284-289 stated that CH4 and CO2 originated from distinct sources while both correlated well with Factor 6 in Table 5, indicating their similar sources/patterns. They are contradictory. Please clarify it.*

There are many sources of $CH_4$ and $CO_2$ in the region. This is reflected by five different components that correlate with one of $CH_4$ or $CO_2$ (but not the other); these are # 1, 2, and 3. Factor 6 is the only one associated with both $CH_4$ and $CO_2$ (Table 5). To improve the clarity of the manuscript, we have modified the text on lines 306-307:

"While $CH_4$ and $CO_2$ mixing ratios frequently correlated in plumes, their ratios were variable overall, suggesting they often originated from distinct sources."

**5. Table 5 claims that coefficients with Pearson correlation coefficients r>0.3 are interpreted as being moderately or strongly associated with a component, which I do not agree.**

We apologize for the confusion and have modified the table caption (which contradicted what was stated in the main text) as follows:

"Table 5. Loadings for the 10-factor, optimal solution (primary variables only). Coefficients with Pearson correlation coefficients r>0.3 are  shown in bold font."

**What are the criteria to select r>0.3 to indicate moderate or strong association? To me, r value around 0.3 means very poor correlation. At least r value should be >0.5 to indicate somewhat correlations.**

Please see our response to major comment #3 above.

**6. Page 24, Table 6. Component 5 was identified as "surface exposed bitumen and hot-water based bitumen extraction". This is only based on the headspace sample of bitumen without any identification of IVOC species. Could it be other sources?**

In Table 6, we show "Hypothesized identifications" and "possible source(s)", i.e., they are not definitive identifications. Other potential sources were considered and discussed on lines 579-618.

In brief, we concluded that surface exposed bitumen and hot-water based bitumen extraction was a likely source contributing to component 5. This is supported by the observations of hydrocarbons that encompassed the same volatility space and produced similar ion fragments in the electron impact mass spectra as observed in the head space analysis above bitumen (Figure 2) and the absence of correlations with variables expected to be associated with other activities, such as $CO_2$.

As already stated in our response to major comment #1, identification of individual IVOC species is not necessary in this context.

No changes were made in response to this comment.

**If TAG-GC-ToF-MS was used, this source could be better identified.**

Please see our responses to major comment #1 and minor comment #1e above.

**Furthermore, IVOCs correlated well with LO-OOA (Table 5), implying that some IVOCs might be secondarily formed or the PCA results were collocated. Please comment on these.**

This is discussed on lines 516-521:

"Component 1 is also weakly associated with the less oxidized oxygenated organic aerosol factor, LO-OOA (r = 0.45). Liggio et al. (2016) found that the observed secondary organic aerosol is dominated by an OOA factor whose mass spectrum was similar to those of aerosols formed from oxidized bitumen vapours. The organic aerosol budget in this study was also dominated by an OOA factor, the LO-OOA (Lee et al., 2018). The association of LO-OOA with component 1 is thus consistent with its association with IVOCs."

It is possible that the IVOCs included species formed by secondary processes. However, given the close proximity to sources (and a bias of the measurement towards non-oxygenated hydrocarbons - see our response to question 2 by reviewer 2), it is reasonable to assume that most of what was observed in this study was primary. To make such a distinction would have required more advanced instrumentation (see comment #1e above).

No changes were made in response to the reviewer's comment.

*7a. Page 24, Table 6. Component 9 "enhancement of CO" was categorized as "VOC oxidation". This is questionable. Why?*

Please see our response to major comment #2 above.

*7b It is apparent that on some days the nighttime CO was quite high, which should not be formed via VOC oxidation due to the fact that nighttime VOC oxidation chemistry is weaker than daytime photochemistry.*

The reviewer is correct that at night, CO mixing ratios are enhanced mainly due to anthropogenic emissions of partially oxidized hydrocarbons into a shallow boundary layer. Since we clarified that we view CO as originating from the incomplete oxidation of hydrocarbons by both anthropogenic and natural sources (see response to major comment #2 above), no further changes were made in response to the reviewer's comment.

*7c. Moreover, if CO was from VOC oxidation, it would destroy the correlations among VOCs (due to various photochemical reactivity of VOC species), leading to poor factor loadings, while the correlations among AVOCs and BVOCs were strong in Table 5.*

A diffuse source such as VOC oxidation would not likely correlate with the (strong) AVOC and BVOC emissions, but likely show up as a stand-alone component similar to what is observed for component #9.

Please also see our response to major comment #2 above.

*7d. Though CO did not have correlations with other combustion tracers, could it be caused by the weakness of the PCA method, or the correlation of CO with other combustion tracers is the "must" for the identification of a combustion souce?*

As stated in our response to major comment #2, the assignment to component 9 to a particular source is tentative as noted on lines 378-380 that "components 6 through 10 are somewhat tentative as many (i.e., 7 – 9) are single variable components and have eigenvalues close to or below unity, i.e., account for less variance than any single variable." and is certainly not a major result of the PCA presented in this paper.

**8a. Page 25, Figure 4. (B) (component 2) vehicles emit aromatics and n-alkanes but component 2 did not in Table 5.**

We interpret component 2 as being mainly due to mine fleet emissions as it is consistent with a sooty combustion source (strong correlations with rBC, pPAH, HOA as well as association with $NO_y$).

There is no doubt that non-road mining truck emit aromatics such as o-xylene and alkanes (Watson et al., 2013). However, one has to keep in mind that a PCA only gives insight into which components contribute to the variability of a specific variable's concentration with time. In this particular case, decane and undecane correlated with factor 2, alas poorly (r=0.22 and 0.27). However, these correlations were the second and third highest of all components. In other words, the results shown in Table 5 do not show the absence of n-alkanes (or xylenes) in component 2, but that another component (#1) that emitted higher concentrations of alkanes and xylenes dominated their variability at this measurement location.

This is not an unusual result at all for PCA. For example, Lan et al. (2014) observed 4 factors in Taiwan and found an association of n-alkanes with a vehicular traffic component, whereas the xylenes correlated with a different factor (labeled "industrial solvent").

We added the following to the discussion section on line 573:

"Furthermore, one would expect an association of non-road mining truck emissions with aromatics and alkanes. Component 2 exhibited only poor correlations with decane (r = 0.22) and undecane (r = 0.27) and negligible correlation with o-xylene (r = 0.08), suggesting that other components (i.e., component 1) explained most of the variability of their concentrations at this site."

**8b. (D) (component 4) Upgrader facilities emitted TS and SO2 only? There was flare stack. Were there no VOC emissions (Table 5)?**

As stated on line 683, we believe that we under sampled emissions from stacks. Flaring (combustion) would likely oxidize most VOCs, such it is doubtful that these emissions would or could be observed.

No changes were made to the manuscript in response to the reviewer's comment.

**9a. Page 26, lines 382-385. Not ture. r=0.27 only for Ox in component 5!**

Our apologies - this was an error. We have made the following change:

"$PM_1$  correlated strongly with the major IVOC component (component 5, r = 0.80), which also moderately associated with LO-OOA (r=0.66) and $NO_{3^-(p)}$ (r = 0.59), as well as $NH_4^+_{(p)}$ and $SO_4^{2-}_{(p)}$ (r = 0.32 and 0.33, respectively)."

*9b Moreover, why didn't Ox have high correlations with any variables in Table 7?*

A discussion about odd oxygen budgets during this study is beyond the scope of this manuscript, but since the reviewer asked:

The communality for $O_x$ was 0.91, suggesting that ~91% of the variability of $O_x$ is accounted for. However, $O_x$ is a complex variable whose concentration is affected by many processes, including air mass "age", i.e., extent of photochemical $O_3$ production, vertical entrainment of $O_3$ and $NO_2$, direct emission of $NO_2$, and dry deposition of $O_3$ and $NO_2$. Its mixing ratio also exhibits a strong diurnal cycle. As a result, no single component dominates its temporal variability. However, there are weak positive correlations of $O_x$ with components 2 (r = 0.36) and 6 (r = 0.33) and sizeable negative correlations with components 3 (r = -0.62) and 7 (r = -0.41). These can be rationalized as follows:

Component 3 is associated with biogenic emissions; low ozone concentrations generally occurred when biogenics accumulated (i.e., low boundary layer height and low $O_3$ abundance at night) such that the anticorrelation is expected. The negative correlation of $O_x$ with component 7 is consistent with the notion of it being a surface source. Component 6 is a factor associated with secondary production of aerosol, which goes hand-in-hand with photochemical $O_3$ production. Finally, the correlation of $O_x$ with component 3 is likely a reflection of the diurnal cycle, since sulfur species were mainly observed during daytime.

No changes were made to the manuscript in response to the reviewer's comment.

*10. Page 43, lines 718-720. Could any measurements be done downwind to verify this hypothesis?*

The text on these lines reads: "Liggio et al. (2016) showed that these hydrocarbons constitute a group of IVOCs in the saturation vapor concentration (C*) range $10^5$ μg m$^{-3}$ < C* < $10^7$ μg m$^{-3}$ that contribute significantly to secondary organic aerosol formation and growth downwind of the oil sands facilities."

Since this hypothesis was verified by Liggio et al. (2016), no changes were made to the manuscript in response to the reviewer's comment.

*11. Page 44, lines 740-742. This also fits CO. But no discussion at all.*

The text on these lines reads: "The PCA struggled most with the allocation of greenhouse gases. Mixing ratios of $CO_2$, in particular, were difficult to reconcile in this analysis due to a high background and large attenuation by biogenic activity and boundary layer meteorology."

Please see our response to major comment #2.

*12. Page 44, lines 749-750. TAG-GC-ToF-MS technique has already been applied to identify and quantify many IVOC species. Many papers have been published in the society. In the "Supplement" document*

Please see our response to minor comment #1e.

*13a. Lines 54-55. Again, what are the selection criteria for these specific VOCs?*

We are not sure what specific lines the reviewer is referring to. The selection criteria for specific VOC monitored as stated in our response to minor question #3.

*13b. Can GC-ITMS measure more VOCs such as C2-C10 HCs and so on for better source identification?*

In principle, yes. However, the GC-ITMS was on site to quantify monoterpenes and set up for measurements of C9 and higher hydrocarbons. Hydrocarbons up to an approximate volatility of toluene did not resolve on our column. It may be possible with column cooling, a different column and longer adsorption times that we might be able to see compounds as low as C2. However, this would be at the expense of C9 and higher time resolution and at the time of measurement was not feasible.

No changes were made to the manuscript in response to the reviewer's comment.

*14. Line 330. No solid evidence for this.*

We are not sure what the reviewer questions here, since the sentence that encompasses line 330 reads: "This is consistent with a recent study by Whaley et al. (2017) that estimated over half (~57%) of the near-surface $NH_3$ during the study period originated from $NH_3$ bi-directional exchange (i.e. re-emission of $NH_3$ from plants and soils), with the remainder being from a mix of anthropogenic sources (~20%) and forest fires (~23%)."

Whaley have published their results in Atmos. Chem. Phys., 18, 2011-2034, 10.5194/acp-18-2011-2018, 2017, and we have no reason to doubt their results.

No changes were made to the manuscript in response to the reviewer's comment.

*15. Figs S2-S4, S7-S11 caption: Table 4 should be Table 5 in the text?*

Thank you for noticing this. All captions have been corrected.

**16. Figure S5 caption: Table 4 should be other Table in the text?**

We believe the reviewer is referring to Table 7 in the main text. The following correction has been made:
"Bivariate polar plots associated with component 4 for the optimum secondary pollutant solution (Table 7)"

**17. Figure S6 caption: Table 4 should be Table 5 in the text?**

We believe the reviewer is referring to Table 7 in the main text. The following correction has been made:
"Bivariate polar plots associated with component 5 for the optimum secondary pollutant solution (Table 7)"

*Anonymous Referee #2*
*This study investigates the sources of IVOCs in the Athabasca oil sands by applying PCA on air pollutants measured at a ground site. IVOCs have been indicated as an important class of SOA precursors. Identification of the major sources of IVOCs is needed in order to make effective measures to reduce their emissions. The objective of this study is interesting, but the data and presentation are too broad, lack of the focus on IVOCs given that the term of "IVOCs" is highlighted in the title.*

Principal component analysis is a commonly used tool that allows characterization of major sources contributing to variability of pollutants at a particular measurement site. The AOS is an interesting region as there are many classes of pollutants emitted in high concentration (greenhouse gases, nitrogen oxides, flame retardants, heavy metals, etc.). This makes it necessary to focus the paper. The addition of "Sources of IVOCs" following "Principal component analysis of summertime ground site measurements in the Athabasca oil sands" in the title alerts the reader that the paper's discussion will focus primarily on factors that IVOCs are associated with. If "Sources of IVOCs" were omitted, the discussion ought to have been much broader and covered other (climate change advocates might argue more important) pollutants as well. However, many these other pollutants were measured close-up by aircraft, which provided more concise information (e.g., (Baray et al., 2018)). In contrast, IVOCs were not observed by the aircraft.

As it is, the observation IVOCs by GC-ITMS at this site was unexpected, as we were not aware a priori that these even existed let alone what processes or facilities would contribute to their release to the gas phase. The approach taken in this paper is to put the observed IVOC signature in context of other temporarily resolved measurements made concurrently, through PCA. We have therefore chosen to keep the title as is and have not altered the manuscript in response to the reviewer's comment.

*1. Measurements of IVOCs were carried out using GC-ITMS. Atmospheric IVOCs are composed of both primary IVOCs, dominated by hydrocarbons, and oxygenated IVOCs, oxidation products of primary IVOCs and VOCs. The elution of oxygenated IVOCs from the GC column is likely incomplete. However, in this study, the split between hydrocarbon-IVOCs and oxygenated-IVOCs was neither performed nor discussed.*

*2. In addition, the collection efficiency and recovery of IVOCs was not described, either.*

The reviewer makes two important points and is correct that oxygenated compounds (alcohols and acids) will likely not elute from the analytical column. They are, therefore, not included in the IVOC signature detected. For the second, we know that calibration curves for n-alkanes, which encompass the bulk of IVOCs observed, were linear, but we do not know what the collection efficiency and recovery of late eluting compounds would be, though we assumed it to be reproducible. We also observed minimal carry-over between chromatograms (see also our reply to comment #3 below).

We hadn't considered oxygenated compounds since bitumen contains very little oxygen (Yoon et al., 2009) and the extent of oxidative processing was assumed to be negligible. IVOCs, however, could have formed through chemical aging, though we'd expect them to be oxygenated (and hence not observed) and in relatively small abundance compared to primary IVOCs.

In response to these concerns, we have expanded section 2.2.1 (Analytically unresolved hydrocarbon signature) and added the following starting on line 142:
"An offline analysis of the headspace above ground-up bitumen gave a similarly unresolved hydrocarbon signal (Fig. 2, black trace). In this particular case, the ambient air chromatogram also shows enhancements of lower molecular weight hydrocarbons (possibly from naphtha) that were not observed in the bitumen sample.
The major ions contributing to the unresolved signals in Figure 2 are associated with alkanes (i.e., $m/z$ 55, 57, 67, 69, etc. – see Fig. S-1). In contrast, counts at masses associated with aromatics (i.e., $m/z$ 115, $C_9H_7^+$, and $m/z$ 91, $C_7H_7^+$) as reported by Cross et al. (2013) were negligible in both the bitumen head space and polluted day samples. The strong resemblance of the unresolved hydrocarbon feature in ambient air with the bitumen head space sample both in terms of volatility (i.e., elution time) and electron impact mass fragmentation is consistent with bitumen as the source of IVOCs at this site.
In the interpretation of the integrated IVOC signal, it is assumed that it is of primary origin, i.e., emitted directly from point sources in the vicinity of the measurement site. For the PCA analysis, the unresolved signal was integrated from a retention time of 25 min to 45 min (gray area in Fig. 2) in all ambient air chromatograms.
The IVOCs observed in this work likely encompass a portion of the total that is emitted. For example, IVOCs generated by combustion processes, such as aircraft engine exhaust, are comprised of alkanes, aromatics and oxygenated compounds (Cross et al., 2013). The use of a chromatographic column in this work biases the IVOC signal towards hydrocarbon-IVOCs, since oxygenated compounds (i.e., alcohols and acids) will not elute from the analytical column. Furthermore, the recovery of VOCs from the pre-concentration unit, while reproducible and likely complete for n-alkanes which bracket the bulk of IVOC emitted and whose calibration curves were linear, is not known for late-eluting compounds, but is assumed to be sufficiently reproducible to yield a semi-quantitative signal. "

We also added the following to the S.I.:

"In the field, there was no noticeable carry-over (i.e., memory effects) of IVOCs, which was occasionally evaluated by flooding the inlet with purified, VOC-free air.

Matrices of ions plotted against retention times for the total ion chromatograms (shown in Figure 2 in the main manuscript) are shown in Fig. S-1. In both cases, the greatest intensity is with masses are associated with alkanes (i.e., $m/z$ 55, 57, 67, 69, etc.).

[Figure]

Figure S-1. Scatter of ions as a function of retention time for the total ion chromatograms shown in Figure 2 of the main manuscript. Darker pixels represent a higher intensity than lighter pixels."

**3. In the light of the description of chemical analysis of IVOCs provided in this study, it is unclear whether the measurements capture the variability of atmospheric IVOCs, which is critical for PCA.**

We agree with the reviewer that variability is critical for PCA. The integrated IVOCs vary in time and this is depicted in Figure 3A.

As we stated above, the IVOC signature does not encompass oxygenated compounds and captures mainly primary IVOCs; we hence agree that we are therefore not capturing the variability of all atmospheric IVOCs. However, it is still acceptable as an input parameter for PCA, and we have not changed the manuscript in response to this comment.

**4. This manuscript mentioned that PCA is unable to determine the fractional contributions of sources to IVOCs in the section of "5. Summary and conclusions".**

We believe the reviewer is referring to lines 758-760: "For instance, PCA does not provide insight into the magnitudes of emissions, though it does capture what conditions change ambient concentrations the most."

We changed the wording as follows:
"For instance, PCA does not provide insight into  emission factors of individual facilities, though it does capture what conditions change ambient concentrations the most."

**5. Why is the PMF analysis not applied in this study? The number of samples is sufficient to provide a robust solution.**

Both PMF and PCA are commonly used for source apportionment, and either receptor model could have been used in this work. In PMF (Paatero and Tapper, 1994), the experimental data is decomposed using non-negative constraints (instead of orthogonality constraints that are used for PCA). This aides in the physical interpretation (i.e., quantification of source contributions), since emissions are then non-negative by default. For this reason, PMF has become more popular than PCA in recent years. The PMF approach does, however, benefit from knowledge about the sources as the solution is not necessarily unique, and solutions are affected by the chosen dimensionality (Paatero and Tapper, 1994). Studies in which data sets were analyzed using both PCA and PMF (Marie et al., 2009; Cesari et al., 2016; Tauler et al., 2009) note that results are similar but not always yield the same number of factors.

We chose PCA with varimax rotation because the main goal of this work was the identification of (orthogonal) source types. Going in, we had little information as to what source or sources would contribute to the IVOC signature. PCA is attractive in this context as it gives a unique solution and is particularly suited as an exploratory tool for identification of components without *a priori* constraints (Jolliffe and Cadima, 2016).

We added the following on line 89:

"PCA was chosen over the more popular positive matrix factorization (PMF) method (Paatero and Tapper, 1994) because it yields a unique solution and is particularly suited as an exploratory tool for identification of components without *a priori* constraints (Jolliffe and Cadima, 2016). "

**6. As mentioned above, ambient IVOCs includes oxygenated IVOCs, oxidation products of IVOCs and VOCs. However, the PCA in this study did not identify a component for oxygenated IVOCs.**

Please see our response to comment #1 above.

**7. Liggio et al. (2016) finds that the evaporation and atmospheric oxidation of low-volatility organic vapors from mined oil sands is directly responsible for the majority of observed SOA mass. Does this mean that the contribution of the component 2 to IVOCs shall be small given that this component is likely related to "Mine fleet and vehicle emissions". More discussion is needed.**

The transformation of IVOCs to organic aerosol mass is outside the scope of this paper, considering that we were close to sources and oxygenated IVOCs were not quantified. The aircraft study, in contrast, focused on the atmospheric oxidation of IVOC emissions that were entrained aloft, transported several hours downwind, at which time emissions from different sources would have merged into a single plume. At our ground site, the extent of oxidative processing is less and more components are distinguishable. Our analysis indicates that IVOCs emitted in component 2 were qualitatively different from those emitted by components 1 and 5, in that they were less associated with the LO-OOA organic aerosol mass loading, but we do not have sufficient information to comment on how IVOCs from these compounds transform and add to aerosol mass downwind.
No changes were made to the manuscript in response to this comment.

*This paper describes principle component analysis (PCA) of ambient air quality data set collected in the Canadian oil sands region over a month or so in the summer of 2013. The data collection was part of a large field project to characterize the impact of oil sand operations on local air quality and climate. This is interesting question given the magnitude of the facilities. Recent work suggests dramatic secondary organic aerosol (SOA) formation downwind of these facilities. This SOA formation was attributed to emissions of low volatility organics, IVOCs. This was described in the Nature paper of Liggio et al. 2016.*

*This paper is focused on better understand the sources of IVOCs. This is an important question that is of interest to readers of ACP. They have done this by performing PCA on an ambient dataset of mainly traditional pollutants (a handful of anthropogenic and biogenic VOCs plus other species). They decompose the data into 10 or 11 factors (depending on if they are looking at secondary species). Three of the factors have some association with IVOCs. The paper contains some relatively qualitative description of the sources of the factors that generally make sense.*

*1. Although I am interested in this topic, after reading it I did not find the paper particularly interesting and did not feel like it made much of contribution to our understanding of the sources.*

We are sorry that the reviewer did not find the paper particularly interesting, but appreciate the frank comment. We feel that we improved the understanding of the IVOC signature in the AOS region by identifying three components, all of which are associated with "the handling of raw bitumen, i.e., the unearthing, mining and transport of crude bitumen, and the disposal of processed material that contains residual bitumen in wet tailings ponds" (lines 744-746).

There has been a lot of emphasis in the literature on bitumen related pollution in tailings ponds (Small et al., 2015) and rivers (Kelly et al., 2010). In contrast, emissions of IVOCs from AOS operations to the gas-phase are understudied, in spite of their significant SOA formation potential (Liggio et al., 2016) and the very likely associated impact on human health. This data set and the analysis presented in this paper are important first steps, though we agree with the reviewer's sentiment that more needs to be done, as we indicated in the "Summary and conclusions" section of the manuscript.

No changes were made in response to this comment

*2. I felt like going in the authors' attributed the IVOCs to bitumen. It was not clear that the paper did anything to support that hypothesis.*

Figure 2 shows that the IVOC signature observed in ambient air is qualitatively similar to the bitumen head space vapor sample. We agree with the reviewer that there could have been other IVOC sources, which we don't believe to be significant at this site, however.

The following was added on lines 147-152:

"The major ions contributing to the unresolved signals in Figure 2 are associated with alkanes (i.e., *m/z* 55, 57, 67, 69, etc. – see Fig. S-1). In contrast, counts at masses associated with aromatics (i.e., *m/z* 115, $C_9H_7^+$, and *m/z* 91, $C_7H_7^+$) as reported by Cross et al. (2013) were negligible in both the bitumen head space and polluted day samples. The strong resemblance of the unresolved hydrocarbon feature in ambient air with the bitumen head space sample both in terms of volatility (i.e., elution time) and electron impact mass fragmentation suggests that bitumen vapors dominate the IVOCs at this site."

We agree that the hypothesis that IVOCs originate from bitumen is supported mainly by a consistency argument and more studies are needed to pinpoint the exact sources. To do this, we would require access to the mining operation sites. However, this is not feasible at the present time since the AOS companies do not allow such measurements to take place for a variety of reasons, including safety.

**4. There is nothing "wrong" with the paper, it is well written and has long descriptions.**

We thank the reviewer for pointing out the absence of technical errors in this manuscript.

**5. The paper would be much more interesting if it could quantitatively attribute IVOCs to sources (even the discussion of components seemed pretty speculative and qualitative).**

We agree with the reviewer that a more quantitative approach would be beneficial, for example, by establishing concrete emission factors for each of the many components of AOS mining operations. As we noted above in comment 3, to accomplish this, we would require access to the mining operation sites.

We stated on lines 758-760, "The PCA analysis in this study suffered from  several limitations. For instance, PCA does not provide insight into  emission factors of individual facilities, though it does capture what conditions change ambient concentrations the most."

No changes were made in response to the reviewer's comment.

**_Larger comments_**
**6. The paper uses a criteria of r>0.3 to indicate moderate correlation. I view 0.3 as almost no correlation. I think the threshold should be much higher.**

We agree. As reviewer #1 also expressed this concern, please see our response to major comment #3 of reviewer 1 and associated changes to the manuscript.

**7. Presumably the GC-ITMS technique used to measure the VOCs and IVOCs could measure a much broader suite of compounds (or even break the chromatogram into subareas) that might yield more insight into what the sources of the IVOCs.**

The GC-ITMS in its current configuration can quantify C9 and higher hydrocarbons. However, when IVOCs were observed, they did not resolve on the analytical column and produced many of the same ions ($m/z$ 57, 71, 69, etc.). As a result, identification let alone quantification of individual IVOCs were unfortunately out of the question.
The suggestion to break up the chromatograms into subareas is an interesting idea. Prior to writing this paper, we had (qualitatively) examined the ambient air total ion chromatograms for differences and had found that IVOC volatility distributions were fairly consistent, i.e. varied collectively in area (by more than 3 orders of magnitude) but not noticeably in the distribution of IVOCs into various volatility bins. In light of the reviewer's comment, we re-analyzed a subset of chromatograms (selected to encompass extreme events of the data set) and integrated the IVOC area prior to tridecane (an easily identified marker that was always present) and after tridecane. Unlike the total IVOC area, which varied by more than 3 orders of magnitude, the ratios of areas before tridecane to after tridecane varied by a relatively small amount, +/-50%. Changes in hydrocarbon volatility distributions are challenging to interpret as they will likely not only depend on the nature of IVOC sources but also on air mass "age" and chemical history (and possibly temperature for a source that outgasses IVOCs). We have therefore chosen not to differentiate between IVOC volatility bins to this manuscript, but have added the following to the conclusion section on line 784:
"Future studies should also investigate how IVOC volatility distributions vary with source type and chemical age."

**_Specific comments_**
**_1. Figure 1 – Including a wind rose for the study period as an inset panel on this figure would be very helpful._**

We considered adding wind rose plots (in lieu of the bivariate plots such as those shown in Figure 5) but decided against it since local wind speeds can be misleading as explained on lines 245-249
"... implies a linear relationship between local wind conditions and air mass origin, which may not be always the case (for example, during or after stagnation periods). In addition, local topography, such as the Athabasca river valley, complicates regional air flow patterns and limit the interpretability of polar plots in general and in particular to the E of AMS 13, where the river valley is located."

No changes were made to the manuscript in response to the reviewer's comment.

**_2. Line 125 – Please add one or two sentences on calibration and QA procedures._**

Details on how the instruments were operated are given in the supplemental information (lines 66-71). To make this clearer, we modified a sentence on lines 125-127.

"A dDetailed descriptions of these instruments and operational aspects such as calibrations are is given in the S.I. Sample observations of analytically unresolved hydrocarbons by GC-ITMS and how these data were used in the analysis are described in section 2.2.1 below."

**_3. Table 2. Who operated instruments is not that interesting. Time resolution would be more useful._**

We modified the table as requested by the reviewer, but note that all the measurements were averaged to the 10 minute preconcentration time of the GC-ITMS (stated on lines 217-218).

**_4. Line 164 – What is the recovery and calibration of the IVOCs?_**

We do not know what fraction of IVOCs is recovered from the preconcentration trap. We do know, however, that there is limited carry-over (i.e., if the instrument was sampling purified air, the chromatogram did not show IVOCs from preceding injections), and that calibration curves for the n-alkanes were linear (stated on line 97 of the S.I.), from which we deduce that the recovery (of alkanes) was reproducible and likely quantitative, i.e., not concentration dependent.

Since we do not know the identity of the individual compounds that make up the IVOCs, they were not calibrated for.

In response to the reviewer's question, we added the following to the S.I. on page 98:

"In the field, there was no noticeable carry-over (i.e., memory effects) of IVOCs, which was occasionally evaluated by flooding the inlet with purified, VOC-free air."

Please also see our response to question 2 of reviewer #2.

**5. Line 198 – What fraction of the data are below LOD?**

This value varied between species and instruments (see below).

| | % of data below LOD |
|---|---|
| **Anthropogenic VOCs** | |
| o-xylene | 10% |
| 1,2,3 - TMB | 27% |
| 1,2,4 - TMB | 8% |
| decane | 44% |
| undecane | 39% |
| **Biogenic VOCs** | |
| α-pinene | 0% |
| ß-pinene | 0% |
| limonene | 1% |
| **Combustion tracers** | |
| $NO_y$ | 0% |
| rBC | 40% |
| CO | 0% |
| $CO_2$ | 0% |
| **Aerosol species** | |
| pPAH | 39% |
| $PM_{10-1}$ | 0% |
| HOA | N/A |
| LO-OOA | N/A |
| **Sulfur species** | |
| Total sulfur (TS) | 35% |
| $SO_2$ | 81% |
| Total reduced sulfur (TRS) | 81% |

| **Other** | |
| --- | --- |
| IVOCs | N/A |
| $CH_4$ | 0% |
| $NH_3$ | 39% |

This information has been added to Table 3.

**6. Line 216 – I did not wade through the other solutions. Text on line 353 suggests they do not material change conclusions – maybe state that here.**

We have added the following to line 238:
"Components 1 through 4 were consistent regardless of the number of components retained."

**7. Line 345 – Moderately associated with IVOC (r=0.31) versus weakly associated with rBC (r=0.3). All these values seem very weak (to almost no) correlation.**

Please see our response to major comment #3 by reviewer #1.

**8. Figure 5. Add labels of directions to different facilities labeled in earlier table.**

We chose to leave the Figure as is since association of specific facilities with pollutants is outside of the scope of this manuscript and would over-interpret the data - after all, air does not necessarily move in straight lines (see our response to minor comment #1).

**9. Line 498 (and other locations) – diesel should not be capitalized.**

Done.

**10. Line 529 – This is not total PAH but particle bound PAH. Seems like surprising they are associated with combustion and not other components (e.g. 1 or 5).**

It is. We stated and discussed potential reasons on lines 556-561.

".... Given this diversity of known sources, the associations of PAHs with only a single component is surprising, though indicates that emissions from the mining fleet (which would include diesel and, perhaps, wind-blown emissions from petcoke that is being transported) gave rise to most of the variability in surface-bound PAH concentrations in this data set. The petcoke emissions identified in the studies mentioned above are likely mainly associated with gas-phase molecules or larger, supermicron sized particles, whose PAH content would not be detected by the pPAH measurement in this data set."

No changes were made.

***Line 544 – But diesel engines are not a major source of CO (gasoline engines are).***

The reviewer is correct, of course. Wang et al. (2016a) give emission factors for CO and $CO_2$ by large caterpillar trucks, which show CO emissions from the big trucks to be minor. We have changed the sentences in question (on lines 570-573) as follows:

"There is little to no association of component 2 with  $CO_2$ (r =  0.08). This is somewhat unexpected as the trucks are expected to release  $CO_2$ (Wang et al., 2016b) but could be due to significantly larger $CO_2$ sources in the area dominating the observed $CO_2$ variability at AMS 13 (e.g., components 3 and 6)."

***Line 552 – Associating the IVOCs bitumen seemed to be the hypothesis going in. It is not clear how this analysis reinforces or tests that hypothesis. It seems pre-conceived and they are just interpreting the data that way.***

Please see our response to major comment #2 above.

***Line 695 – What is the evidence for this claim?***

The sentence the reviewer is to referring reads "The analysis indicates that the strongest IVOC source (Component 5) has the largest impact on $PM_1$ (Table 7)."

We rephrased this sentence as follows:
"The analysis indicates that the component with the strongest IVOC  variability (Component 5) also has the  highest association with $PM_1$ (r = 0.7; Table 7)."

[revised manuscript text omitted]

Matrices of ions plotted against retention times for the total ion chromatograms (shown in Figure 2 in the main manuscript) are shown in Fig. S-1. In both cases, the greatest intensity is with masses are associated with alkanes (i.e., *m/z* 55, 57, 67, 69, etc.).

[Figure]

Figure S-1. Scatter of ions as a function of retention time for the total ion chromatograms shown in

Figure 2 of the main manuscript. Darker pixels represent a higher intensity than lighter pixels.

[revised manuscript text omitted]

---

## Author Response (AR2)

**Dr. HANS D. OSTHOFF**
Associate Professor, Department of Chemistry
University of Calgary, Faculty of Science
University Drive NW, Calgary, AB, Canada T2N 1N4, Canada
phone: 403-220-8689; e-mail: hosthoff@ucalgary.ca
http://homepages.ucalgary.ca/~hosthoff/

September 17, 2018

Re: manuscript ACP-2017-1026

Dear Dr. Russell:

Enclosed, please find a revised manuscript, titled "Principal component analysis of summertime ground site measurements in the Athabasca oil sands with a focus on analytically unresolved intermediate volatility organic compounds" for your consideration.

In our detailed rebuttal, we have outlined additional changes made to the manuscript.

Summarizing the main points: We have developed and described a new way to measure something indicative of the illusive total IVOC, which many, including Liggio et al. (2016), have shown to be important. Previous studies have only poorly captured the gas phase constituents contributing to SOA formation in this volatility bin. We sustained our new measurement for several weeks, creating a unique time series close to the sources, enabling a first look at where the IVOCs may originate. We have capitalized on a large suite of advanced concurrent measurements to help narrow down where IVOCs are coming from, using PCA and polar plots. We have tentatively identified some of its likely sources through PCA. This information is important for understanding this large contributor to PM mass associated with the oil sands mining operation found by Liggio et al. (2016), in terms of helping to determine the extent to which IVOCs emissions could be controlled to reduce oil sands impacts.

We hope that you will find the revised manuscript acceptable for publication and of interest to your readership.

Sincerely,

**Dr. Hans D. Osthoff**

We thank both reviewers and editor for their time and effort reviewing this manuscript. All reviewer comments are reproduced below in **bold, italicized font**. Our responses are shown in regular font. Changes to the text are indicated as underlined text for insertions or are  for deletions. Line numbers given below are for the revised version with all markups shown. We numbered the reviewer comments for easier cross-referencing.

*Co-Editor (Lynn M. Russell) received and published July 27, 2018*
*I agree with Reviewer 1 and the new Reviewer that more revisions to the manuscript are needed to fully address the previous comments.*

In the text below, we have responded to each of the concerns carefully and have made changes to the manuscript, where we believe they are warranted.

*Anonymous Referee #1 received and published June 25, 2018:*
*The authors have addressed most of my comments on the previous version. However, I still have major concern on the identification of IVOC source in the revised manuscript though I noted that it was impossible for them to resolve the IVOCs on the chromatographic column for the measurement campaign.*

*On one hand, in their response to me (Points 1d and 6), they claimed that "Table 5 suggests that the IVOCs in this study originated mainly from a standalone component (#5), were not associated with a biogenic source (#3) and only very weakly with vehicular sources (#2)", and "It is possible that the IVOCs included species formed by secondary processes.*
*However, given the close proximity to sources (and a bias of the measurement towards non-oxygenated hydrocarbons), it is reasonable to assume that most of what was observed in this study was primary".*
*In their response to reviewer #2 (point 2), the replied that "bitumen contains very little oxygen (Yoon et al., 2009) and the extent of oxidative processing was assumed to be negligible".*
*In their response to reviewer #3 (point 2), they responded that "there could have been other IVOC sources, which we don't believe to be significant at this site, however".*
*Based on their responses to all reviewers, the authors are quite confident that bitumen vapors dominate the IVOCs in this study though none of IVOC species was identified in this study.*

*On the other hand, the authors admitted in the response to reviewer #2 (point 2) that "oxygenated compounds will likely not elute from the analytical column. They are, therefore, not included in the IVOC signature detected".*
*In other word, the GC-ITMS technique cannot provide complete elution of oxygenated IVOCs from the GC column or bitumen indeed does not contain oxygenated compounds.*

*However, In Table 5, though IVOCs originated from a standalone component (#5) with factor loading of 0.74, the LO-OOA also had strong correlation with component #5 (0.72) while HOA had poor correlation with component #5, indicating that IVOCs were more related oxygenated organic aerosols (OOA) rather than hydrogen-like OA (HOA).*
*As such, it is not convincing that the IVOCs identified in this study were definitely caused by bitumen vapors.*

The reviewer is correct that we believe that the analytically unresolved IVOC feature is due to bitumen vapors. At the same time, one cannot be 100% certain in a receptor-based analysis but can only present the evidence and draw a conclusion, with caveats noted. Our opinion is based on the similar response of the lab head space analysis to what was observed in ambient air (e.g., Figure 2 and Figure S-1) and that no other source that produces such a response is known to us, nor is a credible alternate source suggested by the reviewer. We have also never observed an unresolved IVOC feature at locations other than the AB oil sands region, where bitumen is mined. At the same time, an unresolved hydrocarbon signal as we observed in this work will have multiple contributing sources. This does not, preclude, however, that one source dominates. We believe this to be bitumen vapors, as stated above.

In contrast, the reviewer's suggestion that the IVOC signature is not caused by bitumen vapors is less likely. The reviewer suggests that the association of IVOCs with the organic aerosol fractions LO-OOA and HOA is somehow in contradiction to bitumen vapors being the dominant source of IVOCs. We do not agree, as such correlations could arise from a fraction of IVOCs partitioning to the aerosol phase, perhaps assisted by oxidation. Furthermore, there are many heavy haulers emitting HOA driving all over the ground covered in bitumen that is dug up and carrying fresh bitumen. It would hence be surprising to not see a correlation between bitumen vapors and HOA. The fact that we do not see oxidation products in the GC-MS is irrelevant in this context.

However, we recognize that our line of reasoning is circumstantial and have made the following change to the text on lines 518-522:

"One possible explanation for the association of IVOCs with tailings ponds vapor  is the presence of bitumen in the ponds that was not separated from the sand during the separation stage (Holowenko et al., 2000). This semi-processed bitumen would be expected to emit similar IVOC vapors to those that were observed in the lab (Fig. 2)."

*Since the focus of this manuscript was about IVOC sources, it is important to make robust and reliable identification of IVOC sources. So far, it is still not certain.*

The identification of IVOC sources was indeed the goal of this work. Hence, as we stated on lines 93-95, "PCA was chosen ... as an exploratory tool for identification of components without a priori constraints." We agree with the reviewer that we fell somewhat short of this goal since there remains considerable uncertainty regarding the exact identification of sources, though we identified several classes of sources (i.e., components of the PCA analysis that are associated with IVOCs). The manuscript reflects this, as we qualified identifications as "hypothesized" (lines 366 and 390) and sources as "potential" (line 100) but we strengthened this point by adding "tentative" to assignments, e.g., on line 450: "  Tentative assignments of these components to source types in the oil sands are given in Table 6 and are discussed below. " and on line 754: "Three components correlated with the IVOC signature and were tentatively assigned to ...".

Having said this, the PCA presented in this paper clearly shows that the IVOCs are associated with emissions from anthropogenic activities, i.e., oil sands mining and upgrading operations. In the conclusion section on line 768 we state that further studies will be needed to pinpoint the exact sources:

"Direct measurements of emissions throughout the processing of raw bitumen are needed to pinpoint source contributions more accurately and aid in the development of potential mitigation strategies. "

In response to the reviewer's concern we have altered the title of the paper, de-emphasizing the source identification:

"Principal component analysis of summertime ground site measurements in the Athabasca oil sands: Sources ofwith a focus on analytically unresolved intermediate volatility organic compoundsIVOCs"

In addition, we have replaced "Sources" with "Components" in the titles of subsection 4.1 (line 469) and 4.2 (line 632) and on line 663 and changed the title of section 3.3 (line 417) from "Spatial distribution of IVOC sources" to "Bivariate polar plots".

***Question on the response 2a about PAC results: could the authors share the raw data for PCA analysis? We often encounter the problems of "collocated factors" when using PCA/APCS though we have many years experience using this tool.***

We have added the raw data as a supplemental file as requested.

We are not sure what "problems of collocated factors" are that the reviewer is referring to, since this is not standard terminology in the PCA literature. Perhaps the reviewer is referring to the general issue with a receptor analysis that different sources/mixtures in the same direction relative to the observing site are usually observed together and are hence challenging to separate. This is acknowledged on lines 772-773:

"Further, the receptor nature of PCA did not always discern between large source areas that may have many individual point sources coming together at the point of observation."

***Clarification:***

***My questions in the previous version from #13a to #17 in your response were related to the "Supplementary" file (not the "manuscript"). Hence, your replies are not relevant.***

We apologize for the earlier misunderstanding. We believe that our responses were relevant, with exception of question 13a (which we misunderstood) and question 14 (which referred to an empty line). We have restated the reviewer's questions and our responses below, making further clarifications as needed.

*13a. Lines 54-55. Again, what are the selection criteria for these specific VOCs?*

 Lines 55-57 in the supplemental states that the GC-ITMS quantified " o-xylene, decane, undecane, 1,2,3- and 1,2,4-trimethylbenzene (TMB), and several monoterpenes (i.e., α-pinene, ß-pinene and limonene)." As stated in our response to minor question #3, "we included all quantified non-methane hydrocarbons in the analysis".

In response to the reviewer's question, we added the following to line 57 of the supplemental: "The GC-ITMS primary responsibility was the quantification of monoterpenes. The remaining VOCs quantified were chosen because (a) they sufficiently resolved on the analytical column, and (b) response factors could be determined, either because the compounds of interest were part of the VOC standard used in the field (such as the aromatics o-xylene, 1,2,3- and 1,2,4-TMB, see below) or relative response factors were determined post-campaign."

*13b. Can GC-ITMS measure more VOCs such as C2-C10 HCs and so on for better source identification?*

In principle, yes. However, the GC-ITMS was on site to quantify monoterpenes and set up for measurements of C9 and higher hydrocarbons. Hydrocarbons up to an approximate volatility of toluene did not resolve on our column. It may be possible with column cooling, a different column and longer adsorption times that we might be able to see compounds as low as C2. However, this would be at the expense of C9 and higher time resolution and at the time of measurement was not feasible.

No changes were made to the manuscript in response to the reviewer's comment.

*14. Line 330. No solid evidence for this.*

It is still not clear what the reviewer is referring to here. In the version uploaded Jan 19 and published Jan 25, 2018, line 330 of the supplemental is an empty line. The preceding section deals with $NH_3$, the following section with CO.

We added a reference to the sentence on lines 326-329 "The lack of association of ammonia with other variables in this component and the bivariate polar plots (Figure S-9) are consistent with an $NH_3$-specific source profile, such as fugitive emissions from one or more point sources that emit independently from other activities (i.e., ammonia storage tanks) and natural emissions from soil and trees (Whaley et al., 2018)."

Line 331 was changed as follows in response to an earlier reviewer comment:
**"Component 9:  Incomplete hydrocarbon oxidation"**

*15. Figs S2-S4, S7-S11 caption: Table 4 should be Table 5 in the text?*

Thank you for noticing this. All captions have been corrected.

***16. Figure S5 caption: Table 4 should be other Table in the text?***

We believe the reviewer is referring to Table 7 in the main text. The following correction has been made: "Bivariate polar plots associated with component 4 for the optimum secondary pollutant solution (Table 7)"

***17. Figure S6 caption: Table 4 should be Table 5 in the text?***

We believe the reviewer is referring to Table 7 in the main text. The following correction has been made: "Bivariate polar plots associated with component 5 for the optimum secondary pollutant solution (Table 7)"

***Anonymous Referee #4 received and published July 27, 2018:***
***I have been asked to review this manuscript at this late stage as I understand that two of the initial reviewers were not available to review the revisions.***

We greatly appreciate referee #4's efforts, thank you.

***I find the authors have done a good job of responding to many specific comments, but that a number of issues were not addressed. While I also have some new suggestions/comments, I think the most important thing for the authors to do is more fully address the first round of comments, especially the more general comments of Reviewer 3 for which no changes were made in the manuscript (some of which I repeat and rephrase below).***

***1. Given the limitations on source sampling:***

***a. Is the subtitle "Sources of IVOCs" really appropriate? I suggest that if the main source is not really verifiable, it's not appropriate.***

We thank the reviewer for this constructive comment and have changed the title to the following:

"Principal component analysis of summertime ground site measurements in the Athabasca oil sands with a focus on analytically unresolved intermediate volatility organic compounds"

***b. Are there GCMS profiles of similar operations available in the literature that could be compared to, providing also a more substantial review of the relevant literature, and comparison of the results of this work to that? This is not that onerous and seems a minimum that the authors can do to provide appropriate context.***

This is an excellent suggestion. We found a handful of relevant papers (Stout and Wang, 2016; Payzant et al., 1980; Yang et al., 2011; Boczkaj et al., 2014; Stout and Wang, 2017) . One of them, Boczkaj et al.

(2014) state that a "literature search revealed that the information on volatile chemical compounds present in bitumen fumes is scarce."

The literature papers all extracted bitumen heavy crude oil and injected their extracts (usually after a clean-up step) via syringe. We did not find anything in the literature regarding direct air injection of bitumen (the method used in our work). Thus, the literature results are not directly comparable to our work since, for example, some components may be extracted into a solution but may not be volatile enough to be detected in a headspace analysis.

Having said this, Yang et al. (2011) report that "a large chromatographic hump of unresolved complex mixtures (UCMs) eluting between n-C10 to n-C40 is pronounced in all oil sands extracts and bitumen samples." in their analysis of AB oil sands samples, which is qualitatively similar to what we report. We have added the following to the main text on lines 147-149:

"The observed unresolved hydrocarbon feature is qualitatively similar to the "large chromatographic hump of unresolved complex mixtures" reported by Yang et al. (2011) during their analysis of bitumen extracts."

**2. Correlations:**

**a. I suggest using the scale and nomenclature of a standard stat reference rather than making one up.**

We're not certain which reference text to consider a standard, and there is no consensus in the literature as to what scale to use (and nomenclature varies somewhat between authors). For example, Henry and Hidy (1979) defined values of near zero -0.2 to +0.2 as "almost no correlation or dependence", < -0.9 and >+0.9 as "strong correlation", and intermediates values (0.2 to 0.9, or -0.2 to -0.9) as a "proportionally less strong correlation". Guo et al. (2004) chose to present only loadings with r > 0.3 in their PCA, whereas Harrison et al. (1996) present only loadings r > 0.23.

In response to this comment, since we found the lowest correlation in the literature that was interpreted being ±0.2, we have altered our criteria on line 365 as follows:

"Associations with r>0.7, r>0.3, and r>0.2r>0.1 are referred to as "strong", "weak", and "poor", respectively." and have updated the descriptions for the few instances where 0.2>r>0.1 from "poor correlation" to "no correlation" throughout the manuscript (lines 373, 376, 575, 583, 588, 603, 666, 668, 670, and 684) and in the supplement, noting that this change did not affect any of the conclusions. For the remaining associations, we note that the r value does give a quantitative measure of the degree of correlation, which supersedes the labels "strong", "weak", and "poor" anyhow.

**b. Have the authors investigated the degree of autocorrelation in the measurements?**

We had not investigated autocorrelation and thank the reviewer for bringing this issue to our attention. In any PCA, it is assumed that the data are independent and not autocorrelated. Autocorrelation, i.e., successive observations are serially correlated, can adversely affect a PCA if present (Vanhatalo and Kulahci, 2016). In a recent paper on this subject, Vanhatalo and Kulahci (2016) state that "the impact of autocorrelation on PCA ... is neither well understood nor properly documented in the literature" and that "there is no clear-cut recommendation on how to deal with autocorrelated data".

We do not believe that autocorrelation significantly affected our analysis. The measurements are independent (see the next question below), are spaced 60 min apart, and the data vary considerable in time (see the time series shown in Figure 3 in the main manuscript). Furthermore, the data were normalized ("standardized") in the PCA to remove relatively constant and large background present for, for example, $CO_2$, $CH_4$, and CO.

Since the reviewer has asked, we calculated the autocorrelation functions of the variables after normalization. In all cases, the autocorrelation function drops roughly exponentially from a value of 1 at lag 0 with a time constant of between 1 and 4 (hrs), with the only exceptions being TRS (time constant of <1) and CO (time constant of 5). At longer time constants, the autocorrelations appear random and show little trend. The only exception is $SO_2$ (and TS), whose autocorrelation function increases to 0.379 with a 7-day period, for reasons unknown though it may suggest a S-related process that is performed weekly at the source facility or facilities.

We found no mention of auto-correlation in any of the PCA papers in the literature and, as non-experts in chemometrics, decided not to comment on the degree of autocorrelation in this paper.

*Is each measurement independent? If so, the low r values are more meaningful.*

Yes, the measurements (Table 2) are independent, with two exceptions. The 22 variables were quantified by 12 independent instruments. Some instruments reported multiple variables, though quantified these variables independently (e.g., by resolving peaks in a chromatogram, or using different wavelengths to quantify CO, $CH_4$, and $CO_2$). The only exceptions are TRS, which is calculated by subtracting TS and $SO_2$, and (perhaps) the apportionment of organic aerosol by the AMS into HOA and LO-OOA.

*3. Is the paper new and interesting? (See general comments from Reviewer 3.)*

*a. I realize these are not specific comments that can be addressed easily, but I believe the authors can do a better job of more explicitly telling readers what is interesting and new about their work. Is it really the sources? If no, don't emphasize that in title. If yes, support it with any available literature.*

We thank the reviewer for this constructive suggestion. There are many interesting aspects of this new and unique data set that we can emphasize more. For instance, there are few independent data sets collected in the Athabasca oil sands region, one of the largest emitters of pollutants in Canada (NPRI, 2013). Further, the measurement suite encompassed the largest variety of collocated analytical instruments close to oil sands mining and upgrading operations to date and included a first, direct observation of airborne IVOCs, that is unique to this area and we have not observed elsewhere where we have made GC-ITMS measurements, i.e., in Calgary and on Vancouver Island (Tokarek et al., 2017). This paper presents this unique data set as a whole, which is analyzed by PCA with Varimax rotation. We show that the integrated IVOC signal is indeed associated with components that include known primary emissions, as suspected by Liggio et al., but not verified with direct IVOC measurements near the sources, as presented in this paper.

There are few source apportionment studies investigating pollution in the Athabasca oil sands (and none that include IVOCs) in the current literature. Existing studies usually had a fairly narrow focus: for example, Cho et al. investigated ground level $O_3$ and $PM_{2.5}$ (Cho et al., 2012), and VOCs (Bari and Kindzierski, 2018; Bari et al., 2016) and $PM_{2.5}$ (Bari and Kindzierski, 2017; Landis et al., 2017) impacting the nearby communities of Ft. McKay and Ft. McMurray were examined. Other studies included source apportionments of pollutants such as PAHs as they affect sediments  (Jautzy et al., 2013) or lichens (Landis et al., 2012). Two recent papers based in part on data collected during JOSM focused on $CH_4$ (Baray et al., 2018) and metals (Phillips-Smith et al., 2017).

We have modified three paragraphs in the introduction, starting on line 64:

"In August 2013,  a comprehensive air quality study as a part of the Joint Oil Sands Monitoring (JOSM) plan (JOSM, 2012), referred to here as the 2013 JOSM intensive study was conducted. This study was performed in northern Alberta at two ground sites in and near Fort McKay in close proximity (as close as 3.5 km) to oil sands mining operations and from a National Research Council of Canada (NRC) Convair 580 research aircraft to characterize oil sands emissions and their downwind physical and chemical transformations (Gordon et al., 2015; Liggio et al., 2016; Li et al., 2017).
One ground site, located at the Wood Buffalo Environmental Association (WBEA) air monitoring station (AMS) 13 (Fig. 1), was equipped with a comprehensive set of instrumentation to measure concentrations of a wide range of trace gases and aerosols (Table 1), yielding a unique and new data set, parts of which are presented in this paper for the first time. As part of this effort, a gas chromatograph equipped with an ion trap mass spectrometer (GC-ITMS) was deployed ..."

Continuing on line 86:

"In this paper, concurrent measurements of air pollutants at the AMS 13 ground site during the 2013 JOSM intensive study are presented. An analytically unresolved hydrocarbon signal was successfully integrated and quantified and is presented as a time series here for the first time.  This independent measurement, which represents an important source of SOA (Liggio et al.) is included directly in  analyzed  principal component analysis (PCA), along with a large number of other measurements, to  gain new insight into the possible origins of the  IVOCs in the Athabasca oil sands ."

On line 436 at the beginning of the introduction, we added the following:

"This work has added to the relatively few data sets of pollutants in the Athabasca oil sands region, one of the largest emitters of airborne pollutants in Canada (NPRI, 2013), that are available in the open literature. Earlier source apportionment studies in the region investigated ground level $O_3$ and $PM_{2.5}$ (Cho et al., 2012), examined VOCs (Bari and Kindzierski, 2018; Bari et al., 2016) and $PM_{2.5}$ (Bari and Kindzierski, 2017; Landis et al., 2017) impacting the nearby communities of Fort McKay and Fort McMurray, or investigated pollutants such as PAHs as they affect sediments  (Jautzy et al., 2013) or lichens (Landis et al., 2012). The  measurement suite in this work encompassed a larger variety of collocated analytical instruments closer to oil sands mining and upgrading operations than these earlier studies and included a first, direct observation of airborne IVOCs, that is unique to this area and we have not observed elsewhere where we have made GC-ITMS measurements, i.e., in Calgary and on Vancouver Island (Tokarek et al., 2017). "

***b. For example, additional and specific discussion in response to Reviewer 2 Comment 7 could add some depth to the information presented.***

We appreciate the suggestion. In comment 7, reviewer 2 asked for more discussion on the link between IVOCs and SOA production several hours downwind of the oil sands region.

"***7. Liggio et al. (2016) finds that the evaporation and atmospheric oxidation of low-volatility organic vapors from mined oil sands is directly responsible for the majority of observed SOA mass. Does this mean that the contribution of the component 2 to IVOCs shall be small given that this component is likely related to "Mine fleet and vehicle emissions". More discussion is needed.***"

Our response was:

"The transformation of IVOCs to organic aerosol mass is outside the scope of this paper, considering that we were close to sources and oxygenated IVOCs were not quantified. The aircraft study, in contrast, focused on the atmospheric oxidation of IVOC emissions that were entrained aloft, transported several hours downwind, at which time emissions from different sources would have merged into a single plume. At our ground site, the extent of oxidative processing is less, and more components are distinguishable. Our analysis indicates that IVOCs  loaded onto component 2 were qualitatively different from those loaded on components 1 and 5, in that they were less associated with the LO-OOA organic aerosol mass loading, but we do not have sufficient information to comment on how IVOCs from these compounds transform and add to aerosol mass downwind. No changes were made to the manuscript in response to this comment."

We would like to add that there is already some discussion of this subject in the paper, e.g., on line 738:

[revised manuscript text omitted]

VOCs quantified were chosen because (a) they sufficiently resolved on the analytical column, and (b)

response factors could be determined, either because the compounds of interest were part of the VOC

standard used in the field (such as the aromatics o-xylene, 1,2,3- and 1,2,4-TMB, see below) or relative response factors were determined post-campaign. Operation, calibration and performance of this instrument have been described elsewhere (Tokarek et al., 2017; Liggio et al., 2016). The GC-ITMS

sampled from a 3.6 m long stainless-steel inlet with an o.d. of 0.635 cm from a height of 5 m above ground. A 1 m long section of the inlet was heated to 110 °C and optimized to remove interference due to $O_3$ while avoiding decomposition of alkenes (Tokarek et al., 2017). The GC oven was programmed as follows: hold at 40 °C for 3.00 min, heat at 1.5 °C min$^{-1}$ to 70° C (reached at 23.00 min), heat at 5° C min$^{-1}$

to 200 °C (reached at 49.00 min) and hold for 4 min (total 53.00 min). This was followed by a 5 min recovery time to allow the oven and pre-concentration trap to cool back to 40 °C. The ion trap mass spectrometer was set to an *m/z* range of 50-425. After data reduction, the GC-ITMS generated 10- minute average concentrations of each VOC quantified every hour.

During the campaign, the GC-ITMS was calibrated in the field using an IONICON VOC standard (Table S-

1) containing (in addition to VOCs that the GC-ITMS did not detect) α-pinene and o-xylene at mixing ratios of ~ 1 ppmv and an uncertainty of 5% and 6%, respectively. A commercial calibrator assembly (IONICON, GCU Standard) was used to deliver diluted calibration mixtures. The instrument responses to the VOC standards were linear ($R^2 > 0.99$). The GC-ITMS was calibrated for other VOCs offline relative to

α-pinene. In the field, there was no noticeable carry-over (i.e., memory effects) of IVOCs, which was occasionally evaluated by flooding the inlet with purified, VOC-free air.

Matrices of ions plotted against retention times for the total ion chromatograms (shown in Figure 2 in the main manuscript) are shown in Fig. S-1. In both cases, the greatest intensity is with masses are associated with alkanes (i.e., *m/z* 55, 57, 67, 69, etc.).

[Figure]

Figure S-1. Scatter of ions as a function of retention time for the total ion chromatograms shown in

Figure 2 of the main manuscript. Darker pixels represent a higher intensity than lighter pixels.

[revised manuscript text omitted]

---

## Author Response (AR3)

We thank both reviewers and the editor for their time and effort reviewing this manuscript. All reviewer comments are reproduced below in **_bold, italicized font_**. Our responses are shown in regular font. Changes to the text are indicated as underlined text for insertions or are  for deletions. Line numbers given below are for the revised version with all markups shown. We numbered the reviewer comments for easier cross-referencing.

In the text below, we have responded to each of the concerns carefully and have made changes to the manuscript, where we believe they are warranted.

***Anonymous Referee #1:***

***I went through the authors' responses and was still not very convinced by their rebuttal or still confused by their explanation about the PCA analysis. I noticed that the authors provided more evidence to confirm the unsolved peaks were from bitumen. However, this still cannot well explain the good correlation of gas-phase IVOCs with particle-phase LO-OOA (0.72, factor 5, Table 5) and poor correlation with particle-phase HOA (0.25, factor 5, Table 5). Their argument of "We do not agree, as such correlations could arise from a fraction of IVOCs partitioning to the aerosol phase, perhaps assisted by oxidation" can't convince me because the oxidative processes of particle-phase IVOCs will most likely destroy the correlation between gas-phase IVOCs and particle-phase LO-OOA, no need to mention that not only IVOCs from bitumen but also IVOCs from other sources can partition to the aerosol phase and be oxidized, which could further ruin the correlation between IVOCs vapor and LO-OOA. In contrast, with regard to the poor correlation between IVOCs vapor and HOA, the authors provided confused response "there are many heavy haulers emitting HOA driving all over the ground covered in bitumen that is dug up and carrying fresh bitumen. It would hence be surprising to not see a correlation between bitumen vapors and HOA". In fact, the correlation between bitumen vapors and HOA is poor (r=0.25). Should we be surprised about it or not?***

***This actually brought in another issue which I mentioned earlier i.e. collocation of factors. Because most pollutants used by the authors are gas-phase pollutants including IVOCs while HOA and LO-OOA are actually particle-phase pollutants. There is argument in the community that it may not be suitable to run PCA using parameters with different physical and chemical characteristics because of factor collocation problems. Hence, the confused PCA results at least for Factor 5 in Table 5 may be due to the collocation of more than one source. If this is true, the PCA results could mislead the readers. How will the PCA results be if the HOA, LO-OOA and/or PM10-1 are removed from the PCA running?***

The reviewer is seeking clarification regarding (1) our interpretation of the relationship of IVOCs with particle-phase LO-OOA and HOA, and (2) the validity of using gas- and particle phase variables in the PCA.

Regarding question (1): The reviewer wonders about the relationship between the IVOCs observed and two of the factors observed by the aerosol mass spectrometer, the "hydrocarbon-like organic aerosol (HOA) ... included as a surrogate for fossil fuel combustion by vehicles (Jimenez et al., 2009) ... [and the] LO-OOA factor .... [which] appears to form rapidly after emission of precursors (Lee et al., 2018)." (lines 205-208 of the main manuscript).

The PCA shows that component 5 is strongly associated with IVOCs (r=0.74) and LO-OOA (r=0.72) but poorly with HOA (r=0.25). The HOA is associated mainly with component 2, a combustion source (likely diesel trucks).

A receptor-based analysis can only reveal such associations, but the physical interpretation(s) of these associations will always be less certain and, therefore, are presented in the "discussion" and not the "results" section. We interpret HOA in the conventional manner as a surrogate for fossil fuel combustion (Jimenez et al., 2009). The fact that the association of HOA with component 5 is poor suggests that IVOCs associated with this component are not dominated by diesel emissions and is consistent with bitumen vapors. We suggested (in our rebuttal letter) that it would have been reasonable to expect a correlation between bitumen vapors and HOA given that the trucks move bitumen on the mine sites; this turned out to be not (not strongly, anyhow) supported by the PCA.

We stated that "The correlation of LO-OOA with two of the three IVOC components in the main PCA and with $PM_1$ in the extended analysis is consistent with the high SOA formation potential of IVOCs", referring to the high SOA formation potential of IVOCs in general. Since the PCA does not give insight into the mechanism(s) of SOA formation or into the presence (or absence) of other organic compounds contributing to SOA formation, we have not commented on this issue in detail in this manuscript but note that the LO-OOA AMS factor will be examined in another manuscript (Lee et al., 2018), which was recently submitted to ACPD.

No changes were made to the manuscript in response to point (1), though we note that more detail is provided on the kinetics of IVOC oxidation in response to question (3) of reviewer #2 below.

Regarding point (2): The reviewer's concern is about using a combination of gas- and particle-phase variables in the analysis.

There have been many publications that have presented analyses with similar mixtures of variables (e.g., (Thurston and Spengler, 1985; Li et al., 1994; Statheropoulos et al., 1998)), so there is certainly precedence. Having said this, we thank the reviewer for the suggestion to perform a sensitivity run and have done so. The results have been added to the Supplemental information section as Table S-10 (reproduced below) and are presented as a 9-component solution, since the dust component associated with $PM_{10-1}$ cannot be generated in the PCA when its main variable is removed.

The pattern in Table S-10 resembles that in Table 5 of the main manuscript, in that the same nine components emerged in both solutions with similar magnitude r values for each of the variables, including the IVOC signature. The only difference is that components 2 and 3 as well as 5 and 6 have traded places (i.e., the relative magnitudes of their eigenvalues, which were of similar magnitude in Table 5, have switched), which is inconsequential.

**Table S-10.** The pattern without aerosol variables after Varimax rotation with 9 components.

| | 1 | 2 | 3 | 4 | 5 | 6 | 7 | 8 | 9 | Commu-nalities |
|---|---|---|---|---|---|---|---|---|---|---|
| **Anthropogenic VOCs** | | | | | | | | | | - |
| o-xylene | **0.88** | 0.02 | 0.04 | 0.09 | 0.12 | 0.10 | -0.03 | 0.17 | **0.32** | 0.94 |
| 1,2,3 - TMB | **0.94** | 0.07 | 0.12 | 0.04 | 0.11 | 0.03 | -0.02 | 0.18 | -0.02 | 0.95 |
| 1,2,4 - TMB | **0.94** | 0.01 | 0.11 | 0.10 | 0.08 | 0.08 | -0.02 | 0.19 | 0.13 | 0.98 |
| decane | **0.93** | -0.01 | 0.20 | 0.16 | 0.01 | 0.18 | 0.04 | 0.04 | 0.06 | 0.97 |
| undecane | **0.89** | -0.07 | 0.26 | 0.25 | -0.03 | 0.12 | 0.06 | -0.04 | -0.03 | 0.94 |
| **Biogenic VOCs** | | | | | | | | | | - |
| $\alpha$-pinene | -0.03 | **0.97** | -0.08 | -0.12 | 0.06 | 0.01 | -0.08 | 0.02 | 0.00 | 0.98 |
| ß-pinene | -0.02 | **0.97** | -0.08 | -0.12 | 0.05 | 0.00 | -0.08 | 0.00 | 0.01 | 0.98 |
| limonene | 0.08 | **0.92** | -0.04 | -0.08 | 0.27 | 0.10 | -0.11 | 0.03 | -0.06 | 0.95 |
| **Combustion tracers** | | | | | | | | | | - |
| $NO_y$ | 0.30 | -0.25 | **0.81** | 0.23 | -0.03 | 0.24 | 0.09 | -0.06 | 0.03 | 0.92 |
| rBC | **0.34** | 0.04 | **0.78** | 0.08 | 0.12 | **0.37** | 0.11 | 0.13 | -0.04 | 0.92 |
| CO | **0.42** | 0.04 | 0.16 | 0.03 | 0.10 | 0.05 | 0.05 | **0.87** | -0.01 | 0.98 |
| $CO_2$ | 0.10 | **0.46** | 0.06 | -0.10 | **0.84** | -0.02 | -0.13 | 0.06 | -0.05 | 0.96 |
| **Aerosol species** | | | | | | | | | | - |
| pPAH | 0.07 | -0.07 | **0.94** | -0.13 | 0.08 | -0.06 | 0.11 | 0.12 | 0.03 | 0.95 |
| **Sulfur** | | | | | | | | | | - |
| TS | 0.26 | -0.15 | 0.03 | **0.94** | -0.05 | 0.03 | -0.02 | 0.01 | 0.14 | 1.00 |
| $SO_2$ | 0.12 | -0.14 | 0.02 | **0.98** | -0.05 | -0.03 | -0.03 | 0.02 | -0.04 | 0.99 |
| TRS | **0.59** | -0.07 | 0.04 | 0.14 | -0.01 | 0.19 | 0.02 | -0.02 | **0.75** | 0.97 |
| **Other** | | | | | | | | | | - |
| IVOCs | **0.35** | 0.13 | **0.32** | -0.03 | 0.00 | **0.84** | -0.03 | 0.05 | 0.15 | 0.98 |
| $NH_3$ | 0.01 | -0.23 | 0.21 | -0.05 | -0.10 | -0.01 | **0.94** | 0.05 | 0.01 | 1.00 |
| $CH_4$ | **0.61** | 0.09 | **0.36** | -0.05 | **0.59** | 0.08 | 0.00 | 0.18 | 0.16 | 0.92 |
| **Eigenvalues** | 5.54 | 3.16 | 2.60 | 2.08 | 1.20 | 1.03 | 0.97 | 0.94 | 0.77 | |
| **% var.** | 29.15 | 16.63 | 13.68 | 10.96 | 6.33 | 5.40 | 5.11 | 4.96 | 4.03 | |
| **% Cum. var.** | 29.15 | 45.79 | 59.46 | 70.43 | 76.76 | 82.16 | 87.26 | 92.23 | 96.25 | |

This result is consistent with our assumption that the analytically unresolved IVOCs observed in this study were mainly of primary origin (assumed on the basis of the close proximity to sources and a bias of the measurement towards non-oxygenated hydrocarbons) and were likely due to  bitumen vapors (on the basis of the similar response of the lab head space analysis to what was observed in ambient air (e.g., Figure 2 and Figure S-1) and that no other source that produces such a response is on hand).

We recognize that it is possible that there were minor contributions to the observed IVOCs by secondary processes. Other sources of IVOCs that contribute to SOA are, for example, diesel emissions (Zhao et al.,

2015). The AB oil sands region is somewhat unusual in this regard, since primary emissions of IVOCs are an unusually large contributor to SOA formation (Liggio et al., 2016; Li et al., 2017). Making a distinction between primary and secondary IVOCs would have required more advanced instrumentation, not on hand during this study. However, the consistency of the results of Table S-10 and Table 5 suggests that the inclusion of HOA, LO-OOA, and $PM_{10-1}$ did not skew the PCA for the data in this paper, which should put the reviewer's concern to rest.

We have modified the paragraph starting on line 214 of the main manuscript as follows:

"To assess which components  impact  secondary product formation, a second PCA was performed which included variables mainly formed through atmospheric chemical processes and whose concentrations more strongly depend on air mass chemical age than those variables selected initially. In this PCA, odd oxygen ($O_x = O_3 + NO_2$), submicron aerosol $SO_4^{2-}{}_{(p)}$, $NO_3^-{}_{(p)}$, $NH_4^+{}_{(p)}$, a second, more-oxidized OOA factor (MO-OOA), and $PM_1$ volume were included, increasing the total number of variables to 28 (Table 4). Furthermore, since oxidation of IVOCs leads to formation of SOA (Robinson et al., 2007; Lee et al., 2018), and the photochemical conversion of IVOC to SOA may adversely affect the PCA, a PCA without secondary and aerosol variables is presented in the S.I. (Table S-10)."

We added the following to the supplementary information section (on line 281):

"**PCA without aerosol variables**

A sensitivity test was conducted by which all aerosol species were removed as variables. The results of this sensitivity test are shown in Table S-10 and are presented as a 9-component solution, since the dust component associated with $PM_{10-1}$ (component 7 in Table 5) cannot be generated when its main variable is removed.

The pattern in Table S-10 resembles that in Table 5 of the main manuscript, in that the same nine components emerged in both solutions with similar magnitude r values for each of the variables, including the IVOC signature. The only difference is that components 2 and 3 as well as 5 and 6 have traded places (i.e., the relative magnitudes of their eigenvalues, which were similar in Table 5, have switched), which is inconsequential. Furthermore, the correlation coefficients in Table S-10 are of similar magnitude (i.e., within ±0.1) as those in Table 5, which suggests that IVOC to SOA conversion does not adversely affect the PCA, likely because of the proximity of the receptor site to sources."

As such, a PCA by its very nature yields patterns, i.e., recurring structures, which are fusions of many single sources. Examples of factors/components comprised of numerous and somewhat dissimilar (point) sources are common in the literature; recurring examples include "biogenic" factors (originating from many different trees and tree species), "vehicle emission" or "traffic" factors (from a fleet of motor vehicles exhibiting a range of individual emission compositions), or "industrial sources" (from a range of activities) – see, for example (Bruno et al., 2001; Guo et al., 2004; Guo et al., 2007; Chavent et al., 2009; Lan et al., 2014; Cesari et al., 2016; Thurston and Spengler, 1985; Samara et al., 1994; Buhr et al., 1995). Emissions from these source types include a combination of trace gases ($NO_x$, $SO_2$, CO, $CO_2$, certain VOCs etc.) and particles (e.g., soot from a diesel vehicle, or OA from trees).

*Anonymous Referee #2*
*Suggestions for revision or reasons for rejection (will be published if the paper is accepted for final publication)*

*1. GC-ITMS: what type of column was used for measuring the IVOCs?*

We added this information to the S.I. on line 76: "Operation, calibration and performance of this instrument have been described elsewhere (Tokarek et al., 2017; Liggio et al., 2016). Briefly, the GC was operated with 30 m (length) × 0.25 mm (inner diameter) × 0.25 μm (film thickness) DB-5MS analytical column with helium carrier gas. The GC-ITMS sampled from ...."

*Would the IVOCs be limited to hydrocarbon species or would oxygenated species be detected too?*

This is discussed on line 167 of the main manuscript:

"The use of a chromatographic column in this work biases the IVOC signal towards hydrocarbon-IVOCs, since oxygenated compounds (i.e., alcohols and acids) will not elute from the analytical column. Furthermore, the recovery of VOCs from the pre-concentration unit, while reproducible and likely complete for n-alkanes which bracket the bulk of IVOC emitted and whose calibration curves were linear, is not known for late-eluting compounds, but is assumed to be sufficiently reproducible to yield a semi-quantitative signal. "

No changes were made in response to this comment.

*Depending on that answer, are the IVOC measurements discussed here primary and/or secondary species?*

We believe that the IVOC observed the GC-ITMS are mainly primary in origin, as stated on line 160 of the main manuscript:

"In the interpretation of the integrated IVOC signal, it is assumed that it is of primary origin, i.e., emitted directly from point sources in the vicinity of the measurement site."

Please see also our response to reviewer #1, where we corroborate the notion that IVOCs are mainly primary.

No changes were made in response to this comment.

**2. Line 160-167: This discussion would likely benefit from broadly citing and describing the IVOC work that has been done with fossil fuel sources other than aircraft. For example, Gentner et al. (PNAS, 2012) and Zhao et al. (ES&T, 2014, 2015, 2016) that look at IVOC emissions from evaporated fuel and mobile sources.**

We thank the reviewer for alerting us to these papers. We agree that diesel engine IVOC emissions are more relevant than those from aircraft and have modified the text (on line 165) as follows:

"The IVOCs observed in this work likely encompass a portion of the total that is emitted. For example, IVOCs generated by combustion processes, such as  diesel engine exhaust, are comprised of aliphatic alkanes, including cyclic and branched alkanes, and aromatics (Gentner et al., 2012; Zhao et al., 2015) ."

**3. The manuscript at one point (line 173-175) says that species are conserved between emission and measurement but at other points allude to the role of secondary formation and aging (e.g., PCA with secondary pollutants). A short paragraph that discusses the transport times between emission and measurement (based on distances in Table 1 and average wind speeds), average OH concentrations, and kOH of IVOCs would be useful to understand the potential influence of photochemical aging on the measurements.**

An excellent suggestion.

We have added the following text starting on line 116:

"A potentially important consideration is the photochemical aging of emissions between the points of emission and observation. During daytime, the average surface wind speed was 7.5 km hr$^{-1}$ (2.1 m s$^{-1}$). The average transit times were 0.5 hr to the edge of the closest mining operation, 1.6 hr to the 12.2 km distant Mildred Lake Plant site, and 3.2 hr to the Muskeg River Mine site located 23.7 km upwind."

We do not have direct measurements for hydroxyl radical concentrations during this campaign; Liggio et al. (2016) reported an estimated mid-day [OH] of $7\times10^6$ molecules cm$^{-3}$.

To corroborate the above OH concentration and transport times, we calculated photochemical age using the method outlined by Borbon et al. (2013), but substituting n-decane for benzene since the latter was not quantified.

The following text was added the supplemental on line 516:

"**Estimate of photochemical age**

Photochemical age was calculated using the method outlined by Borbon et al. (2013), but substituting n-decane for benzene since the latter was not quantified. The photochemical age of an air mass, Δt was calculated from the observed concentrations of 124-trimethylbenzene (124TMB) and n-decane using:

$$\Delta t = \frac{1}{[OH]\times(k_{124TMB}-k_{decane})} \times \left[\ln\left(\frac{[124TMB]}{[decane]}\right)_{t=0} - \ln\left(\frac{[124TMB]}{[decane]}\right)\right] \quad\quad (S\text{-}1)$$

where $k_{124TMB}$ = 3.25×10$^{-11}$ cm$^3$ molecule$^{-1}$ s$^{-1}$ and $k_{decane}$ = 1.10×10$^{-11}$ cm$^3$ molecule$^{-1}$ s$^{-1}$ are rate coefficients for reaction of OH with 124-TMB and n-decane (at 298 K), respectively, whose values were taken from Seinfeld and Pandis (2006). The ratio of [124TMB] to [decane] at the point of emission (time t = 0) was estimated from a plot of [124TMB] to [n-decane] (Figure S-13, left-hand side) and a straight-line fit to the nocturnal data (assumed to be unaffected by oxidation and shown in blue color). The slope of this line was 1.15±0.07 (r$^2$ = 0.84). Daytime data (color-coded by solar zenith angle, SZA) exhibit lower ratios of [124TMB]/[decane] as a result of the faster oxidation of 124TMB by OH.

Shown in Figure S-13 on the right-hand side is a plot of the photochemical age, calculated using equation (S-1) and an assumed [OH] of 7×10$^6$ molecules cm$^{-3}$ taken from Liggio et al. (2016), as a function of SZA (filtered for peak OH of 11:00 and 16:00 local time). The error bars indicate ages calculated using emission factors of 1.08 and 1.22, respectively. The average (±1 standard deviation) photochemical age is 1.0±0.4 hr. This photochemical age applies mainly to component 1; we assume that the photochemical ages of sources associated with other components were similar.

[Figure]

**Figure S-13. (A)** Plot of 124TMB mixing ratios against mixing ratios of n-decane, color-coded by solar zenith angle. The blue data points were collected at night. **(B)** Photochemical age calculating using equation S-1 plotted as a function of solar zenith angle.

In their analysis of IVOC photochemical aging, Zhao et al. (2014) estimated an average $k_{OH}$ for diesel-exhaust IVOCs of 1.8×10$^{-11}$ cm$^3$ molecule$^{-1}$ s$^{-1}$ (though their estimated rate coefficients varied and increased slightly with volatility bin between about 1 and 3 ×10$^{-11}$ cm$^3$ molecule$^{-1}$ s$^{-1}$). From this, we calculate a pseudo first-order lifetime of 130 min (2.17 hr) with respect to IVOC oxidation by OH during daytime. Using a photochemical age of 1.0±0.4 hr, we calculate that between 25% and 50% of the emitted IVOC is (potentially) oxidized during daytime. Photochemical aging will affect data collected during the daytime hours (from ~11 am to ~4 pm) or ~25% of the data (56 out of 218 data points) used in the PCA and likely resulted in partial conversion of IVOCs to SOA."

As stated in the main manuscript (line 168), oxygenated compounds (i.e., IVOC oxidation products) are unlikely to elute from the analytical column. Hence, the net effect is that the observed IVOC abundance is attenuated during daytime, which is not expected to substantially impact the PCA as some, if not most, of the starting material will remain. In contrast, IVOC emissions at night or during the morning and late afternoon would be by-and-large unaffected by OH oxidation.

How and on what time scale this photo-oxidation chemistry will translate into SOA formation is a complex issue (details of which are beyond the scope of this manuscript), suffice to say that Figure 1 of Liggio et al. (2016) shows (some) SOA production on a 1 hr time scale, and that some of the SOA generated from IVOCs is included in the LO-OOA observed by AMS (Lee et al., 2018).

We have added the following paragraph to the main manuscript on line 632:

"Component 5 correlates strongly with LO-OOA (r = 0.72), which is likely generated in part by photochemical aging of IVOCs.  A back-of-the-envelope calculation using a $k_{OH}$ of $1.8\times10^{-11}$ $cm^3$ $molecule^{-1}$ $s^{-1}$ based on that used for diesel exhaust IVOCs (Zhao et al., 2014) and an estimated mid-day OH concentration of $7\times10^6$ molecules $cm^{-3}$ (Liggio et al., 2016) gives a first-order lifetime of 130 min with respect to IVOC oxidation by OH during daytime. The photochemical age, estimated using relative concentrations of 124-TMB and n-decane and the method described by Borbon et al. (2013), during daytime was 1.0±0.4 hr; assuming similar photochemical ages, we estimate that between 25% and 50% of the emitted IVOC is (potentially) oxidized during daytime (see S. I.). This oxidation will contribute SOA growth (Kroll et al., 2011). Hence, we expect some formation and growth of organic aerosol associated with component 5."

We added the following paragraph to line 759 of the main manuscript:

"The relatively short distance to sources and young photochemical age suggests that IVOCs would experience a relatively small number of oxidation steps. Consistent with this interpretation, a correlation with the more-oxidized MO-OOA is not observed in component 5 (r = 0.10; Table 7). However, component 6, which is (poorly) anticorrelated with IVOCs (r = -0.23), is strongly correlated with MO-OOA (r = 0.92), consistent with the notion that this component is more photochemically processed and that IVOCs contribute to this SOA AMS factor."

***4. Line 223: As the authors are aware, there are nuanced approaches to dealing with values below the detection limit based on the distribution of the data. See for example, https://www.tandfonline.com/doi/pdf/10.1080/08940630.1989.10466534***

We thank the reviewer for bringing this reference (Cohen and Ryan, 1989) to our attention. Cohen and Ryan (1989) show numerical simulations that suggest that the log-fill-in method *may* provide more accurate results in some cases (depending on the geometric standard deviation and level of truncation of a variable) and suggest that a more tailored approach may benefit some analyses.

However, it is common practice in the literature (e.g., (Harrison et al., 1996; Polissar et al., 1998; Guo et al., 2004; Zhao et al., 2004; Mintz and McWhinney, 2008)) to use the "half of the LOD method" to replace values below an instrument's LOD with half of its value, which is what we followed.

We have expanded the references cited on lines 231 but have not made changes in direct response to this comment.

"When concentrations were below their respective limit of detection (LOD; values are given in Table 3), half the reported LOD was used to minimize bias (Harrison et al., 1996; Polissar et al., 1998; Zhao et al., 2004; Guo et al., 2004)."

***5. Section 2.3.2: Does log-transforming the data instead of normalizing help with the PCA?***

The reviewer asks an interesting and insightful question: whether a logarithmic transformation of the input data (in addition to, or as an alternative to, standardization to zero mean and unit variance to alleviate the effects of different measurement units) would improve the PCA.

We have chosen to follow common data pre-treatment practice in the field, which is to standardize the input to zero mean and unit variance (e.g., (Thurston and Spengler, 1985; Li et al., 1994; Statheropoulos et al., 1998; Bruno et al., 2001; Guo et al., 2004; Chan and Mozurkewich, 2007; Mintz and McWhinney, 2008; Jolliffe and Cadima, 2016)).

A logarithmic transformation is performed to reduce the influence of extreme values (or outliers) and may give more weight to lower concentration data points (Baxter, 1995) or when its based on a sound reason, such as taking a logarithm of an equilibrium constant, for which there are reasons nested in thermodynamics (Malinowski, 2002). The term "outlier" implies to us that a measurement value that was somehow made in error, which is not the case for this data set. Further, removing or dampening large concentration values is, in our opinion, not a good approach to pollution research.

No changes were made to the manuscript in response to the reviewer's question.

***6. Line 285-286: The source signatures for a single source could also change with environmental conditions and hence result in varying association of the IVOCs with VOCs.***

We thank the reviewer for this suggestion and have added the following text on line 291 of the manuscript:

"The IVOC magnitude also varied greatly and often increased and decreased in tandem with the other VOCs (e.g., on Aug 24, 16:30 UTC) but also increased independently from the other VOC abundances (e.g., on Aug 30, 01:20 UTC, and on the night of Aug 22). This behaviour suggests the presence of multiple sources with distinct signatures that are being sampled to a varying extent at different times or, perhaps, a single source whose emission profile varies. This, coupled with the intermittency of the highly elevated signals, presents an analysis problem frequently encountered in environmental analysis that is usually investigated through a factor or principal component analysis (Thurston et al., 2011; Guo et al., 2004)."

**7. Line 328: Stating the coefficient of variation (CoV) would help compare the variability between pollutants better since the CoV is a normalized metric.**

We modified the sentence in question as follows:

"Mixing ratios of $SO_2$ exhibited the most variability of all pollutants, as judged from the relative standard deviation of each of the measurements (Table 3)."

We made an analogous change to a sentence on line 360 of the main manuscript:

"Ammonia was not as variable as some of the other pollutants (e.g., the anthropogenic VOCs, sulfur species) as judged from its relative standard deviation (Table 3), which suggests a geographically more disperse source or sources similar to CO or $CH_4$, which have a "background"."

In Tables 3 and 4, we removed the "standard deviation" column and added a "relative standard deviation" column.

**8. Line 578-582: Wouldn't photochemical oxidation make the IVOCs less prone to aerosol formation since they would be more likely to fragment?**

The sentence in question reads: "whereas the IVOCs .... may have been ... oxidized to a greater extent and hence more prone to rapid aerosol formation."

Oxidation of organics is generally viewed as a source of atmospheric organic aerosol through secondary organic formation. This occurs because of increased functionality on the carbon skeleton, which reduces volatility (Kroll et al., 2011). The reviewer is correct that oxidation of hydrocarbons can also lead to their fragmentation. As pointed out by Kroll et al. (2011), small (four carbons or fewer) organic species are unlikely to contribute to aerosol formation, even though they might still form organic aerosol through oligomerization reactions. Oxidation reactions ("aging") of atmospheric organic aerosol are ultimately (after a few generations of oxidation) dominated by fragmentation reactions which then act as organic aerosol sinks, because oxidized organics may fragment and volatilize upon further oxidation (Kroll et al., 2011).

The IVOCs observed in this work are in the volatility range of $C_{11} - C_{17}$ hydrocarbons; even if these were to fragment during the initial oxidation step, at least one of the fragments would contain more than four carbons (and likely also oxygen) and hence can contribute to aerosol formation.

No changes were made in response to the reviewer's question.

**Also, would the GC-ITMS be able to detect them if the IVOCs were oxidized by the microbes? It's possible I have not understood the reasoning for differences in the IVOCs in components 1, 2, and 5 correctly. If that is the case, please consider rephrasing the last sections of that paragraph.**

We do not believe that the GC-ITMS can differentiate between IVOCs oxidized photochemically or biochemically, though both microbial and photochemical oxidation are plausible. We have rephrased the paragraph as follows:

"One reason for the difference could be that the bitumen that is transported by the mining fleet is relatively freshly exposed, whereas the IVOCs released from tailings ponds  or from mine faces (component 5) may have been  oxidized to a greater extent and hence more prone to rapid aerosol formation."

**Literature cited above**

Baxter, M. J.: Standardization and transfomration in principal component analysis, with applications to archaeometry, Appl. Stat.-J. R. Stat. Soc., 44, 513-527, 10.2307/2986142, 1995.

Borbon, A., Gilman, J. B., Kuster, W. C., Grand, N., Chevaillier, S., Colomb, A., Dolgorouky, C., Gros, V., Lopez, M., Sarda-Esteve, R., Holloway, J., Stutz, J., Petetin, H., McKeen, S., Beekmann, M., Warneke, C., Parrish, D. D., and de Gouw, J. A.: Emission ratios of anthropogenic volatile organic compounds in northern mid-latitude megacities: Observations versus emission inventories in Los Angeles and Paris, J. Geophys. Res.-Atmos., 118, 2041-2057, doi:10.1002/jgrd.50059, 2013.

Bruno, P., Caselli, M., de Gennaro, G., and Traini, A.: Source apportionment of gaseous atmospheric pollutants by means of an absolute principal component scores (APCS) receptor model, Fresenius J Anal Chem, 371, 1119-1123, 10.1007/s002160101084, 2001.

Buhr, M., Parrish, D., Elliot, J., Holloway, J., Carpenter, J., Goldan, P., Kuster, W., Trainer, M., Montzka, S., McKeen, S., and Fehsenfeld, F.: Evaluation of ozone precursor source types using principal component analysis of ambient air measurements in rural Alabama, J. Geophys. Res.-Atmos., 100, 22853-22860, doi:10.1029/95JD01837, 1995.

Cesari, D., Amato, F., Pandolfi, M., Alastuey, A., Querol, X., and Contini, D.: An inter-comparison of PM10 source apportionment using PCA and PMF receptor models in three European sites, Environm. Sci. Poll. Res., 23, 15133-15148, 10.1007/s11356-016-6599-z, 2016.

Chan, T. W., and Mozurkewich, M.: Application of absolute principal component analysis to size distribution data: identification of particle origins, Atmos. Chem. Phys., 7, 887-897, 10.5194/acp-7-887-2007, 2007.

Chavent, M., Guégan, H., Kuentz, V., Patouille, B., and Saracco, J.: PCA- and PMF-based methodology for air pollution sources identification and apportionment, Environmetrics, 20, 928-942, doi:10.1002/env.963, 2009.

Cohen, M. A., and Ryan, P. B.: Observations Less than the Analytical Limit of Detection: A New Approach, J. Air Waste Manag. Assoc., 39, 328-329, 10.1080/08940630.1989.10466534, 1989.

Gentner, D. R., Isaacman, G., Worton, D. R., Chan, A. W. H., Dallmann, T. R., Davis, L., Liu, S., Day, D. A., Russell, L. M., Wilson, K. R., Weber, R., Guha, A., Harley, R. A., and Goldstein, A. H.: Elucidating secondary organic aerosol from diesel and gasoline vehicles through detailed characterization of organic carbon emissions, Proceedings of the National Academy of Sciences, 109, 18318-18323, 10.1073/pnas.1212272109, 2012.

Guo, H., Wang, T., and Louie, P. K. K.: Source apportionment of ambient non-methane hydrocarbons in Hong Kong: Application of a principal component analysis/absolute principal component scores (PCA/APCS) receptor model, Environ. Pollut., 129, 489-498, 10.1016/j.envpol.2003.11.006, 2004.

Guo, H., So, K. L., Simpson, I. J., Barletta, B., Meinardi, S., and Blake, D. R.: C1–C8 volatile organic compounds in the atmosphere of Hong Kong: Overview of atmospheric processing and source apportionment, Atmos. Environm., 41, 1456-1472, 10.1016/j.atmosenv.2006.10.011, 2007.

Harrison, R. M., Smith, D. J. T., and Luhana, L.: Source Apportionment of Atmospheric Polycyclic Aromatic Hydrocarbons Collected from an Urban Location in Birmingham, U.K, Environm. Sci. Technol., 30, 825-832, 10.1021/es950252d, 1996.

Jimenez, J. L., Canagaratna, M. R., Donahue, N. M., Prevot, A. S. H., Zhang, Q., Kroll, J. H., DeCarlo, P. F., Allan, J. D., Coe, H., Ng, N. L., Aiken, A. C., Docherty, K. S., Ulbrich, I. M., Grieshop, A. P., Robinson, A. L., Duplissy, J., Smith, J. D., Wilson, K. R., Lanz, V. A., Hueglin, C., Sun, Y. L., Tian, J., Laaksonen, A., Raatikainen, T., Rautiainen, J., Vaattovaara, P., Ehn, M., Kulmala, M., Tomlinson, J. M., Collins, D. R., Cubison, M. J., E., Dunlea, J., Huffman, J. A., Onasch, T. B., Alfarra, M. R., Williams, P. I., Bower, K., Kondo, Y., Schneider, J., Drewnick, F., Borrmann, S., Weimer, S., Demerjian, K., Salcedo, D., Cottrell, L., Griffin, R., Takami, A., Miyoshi, T., Hatakeyama, S., Shimono, A., Sun, J. Y., Zhang, Y. M., Dzepina,

K., Kimmel, J. R., Sueper, D., Jayne, J. T., Herndon, S. C., Trimborn, A. M., Williams, L. R., Wood, E. C., Middlebrook, A. M., Kolb, C. E., Baltensperger, U., and Worsnop, D. R.: Evolution of Organic Aerosols in the Atmosphere, Science, 326, 1525-1529, 10.1126/science.1180353, 2009.

Jolliffe, I. T., and Cadima, J.: Principal component analysis: a review and recent developments, Philosophical Transactions of the Royal Society A: Mathematical, Physical and Engineering Sciences, 374, 10.1098/rsta.2015.0202, 2016.

Kroll, J. H., Donahue, N. M., Jimenez, J. L., Kessler, S. H., Canagaratna, M. R., Wilson, K. R., Altieri, K. E., Mazzoleni, L. R., Wozniak, A. S., Bluhm, H., Mysak, E. R., Smith, J. D., Kolb, C. E., and Worsnop, D. R.: Carbon oxidation state as a metric for describing the chemistry of atmospheric organic aerosol, Nature Chemistry, 3, 133, 10.1038/nchem.948, 2011.

Lan, C.-H., Huang, Y.-L., Ho, S.-H., and Peng, C.-Y.: Volatile organic compound identification and characterization by PCA and mapping at a high-technology science park, Environ. Pollut., 193, 156-164, 10.1016/j.envpol.2014.06.014, 2014.

Lee, A. K. Y., Adam, M. G., Liggio, J., Li, S.-M., Li, K., Willis, M. D., Abbatt, J. P. D., Tokarek, T. W., Odame-Ankrah, C. A., Huo, J. A., Osthoff, H. D., Strawbridge, K. B., and Brook, J. R.: A large contribution of anthropogenic organo-nitrates to secondary organic aerosol in Alberta oil sands, Atmos. Chem. Phys. Discuss., article number acp-2018-1177 2018.

Li, S.-M., Anlauf, K. G., Wiebe, H. A., and Bottenheim, J. W.: Estimating primary and secondary production of HCHO in eastern North America based on gas phase measurements and principal component analysis, Geophys. Res. Lett., 21, 669-672, doi:10.1029/94GL00643, 1994.

Li, S.-M., Leithead, A., Moussa, S. G., Liggio, J., Moran, M. D., Wang, D., Hayden, K., Darlington, A., Gordon, M., Staebler, R., Makar, P. A., Stroud, C. A., McLaren, R., Liu, P. S. K., O'Brien, J., Mittermeier, R. L., Zhang, J., Marson, G., Cober, S. G., Wolde, M., and Wentzell, J. J. B.: Differences between measured and reported volatile organic compound emissions from oil sands facilities in Alberta, Canada, Proceedings of the National Academy of Sciences, 114, E3756-E3765, 10.1073/pnas.1617862114, 2017.

Liggio, J., Li, S.-M., Hayden, K., Taha, Y. M., Stroud, C., Darlington, A., Drollette, B. D., Gordon, M., Lee, P., Liu, P., Leithead, A., Moussa, S. G., Wang, D., O'Brien, J., Mittermeier, R. L., Brook, J., Lu, G., Staebler, R., Han, Y., Tokarek, T. W., Osthoff, H. D., Makar, P. A., Zhang, J., Plata, D., and Gentner, D. R.: Oil Sands Operations as a Large Source of Secondary Organic Aerosols, Nature, 534, 91-94, 10.1038/nature17646, 2016.

Malinowski, E. R.: Factor analysis in chemistry, 3rd ed., Wiley, New York, 414 pp., 2002.

Mintz, R., and McWhinney, R. D.: Characterization of volatile organic compound emission sources in Fort Saskatchewan, Alberta using principal component analysis, J. Atmos. Chem., 60, 83-101, 10.1007/s10874-008-9110-5, 2008.

Polissar, A. V., Hopke, P. K., Paatero, P., Malm, W. C., and Sisler, J. F.: Atmospheric aerosol over Alaska: 2. Elemental composition and sources, J. Geophys. Res.-Atmos., 103, 19045-19057, doi:10.1029/98JD01212, 1998.

Robinson, A. L., Donahue, N. M., Shrivastava, M. K., Weitkamp, E. A., Sage, A. M., Grieshop, A. P., Lane, T. E., Pierce, J. R., and Pandis, S. N.: Rethinking organic aerosols: Semivolatile emissions and photochemical aging, Science, 315, 1259-1262, 2007.

Samara, C., Kouimtzis, T., and Katsoulos, G. A.: Characterization of airborne particulate matter in Thessaloniki, Greece, Toxicological & Environmental Chemistry, 41, 221-232, 10.1080/02772249409357977, 1994.

Seinfeld, J. H., and Pandis, S. N.: Atmospheric chemistry and physics: from air pollution to climate change, 2nd ed., Wiley, Hoboken, N.J., 2006.

Statheropoulos, M., Vassiliadis, N., and Pappa, A.: Principal component and canonical correlation analysis for examining air pollution and meteorological data, Atmos. Environm., 32, 1087-1095, 10.1016/S1352-2310(97)00377-4, 1998.

Thurston, G. D., and Spengler, J. D.: A quantitative assessment of source contributions to inhalable particulate matter pollution in metropolitan Boston, Atmos.Environm. (1967), 19, 9-25, 10.1016/0004-6981(85)90132-5, 1985.

Thurston, G. D., Ito, K., and Lall, R.: A source apportionment of U.S. fine particulate matter air pollution, Atmos. Environ., 45, 3924-3936, 10.1016/j.atmosenv.2011.04.070, 2011.

Tokarek, T. W., Brownsey, D. K., Jordan, N., Garner, N. M., Ye, C. Z., Assad, F. V., Peace, A., Schiller, C. L., Mason, R. H., Vingarzan, R., and Osthoff, H. D.: Biogenic Emissions and Nocturnal Ozone Depletion Events at the Amphitrite Point Observatory on Vancouver Island, Atmosphere-Ocean, 1-12, 10.1080/07055900.2017.1306687, 2017.

Zhao, W., Hopke, P. K., and Karl, T.: Source Identification of Volatile Organic Compounds in Houston, Texas, Environm. Sci. Technol., 38, 1338-1347, 10.1021/es034999c, 2004.

Zhao, Y. L., Hennigan, C. J., May, A. A., Tkacik, D. S., de Gouw, J. A., Gilman, J. B., Kuster, W. C., Borbon, A., and Robinson, A. L.: Intermediate-Volatility Organic Compounds: A Large Source of Secondary Organic Aerosol, Environm. Sci. Technol., 48, 13743-13750, 10.1021/es5035188, 2014.

Zhao, Y. L., Nguyen, N. T., Presto, A. A., Hennigan, C. J., May, A. A., and Robinson, A. L.: Intermediate Volatility Organic Compound Emissions from On-Road Diesel Vehicles: Chemical Composition, Emission Factors, and Estimated Secondary Organic Aerosol Production, Environm. Sci. Technol., 49, 11516-11526, 10.1021/acs.est.3b02841, 2015.

[revised manuscript text omitted]

[b] Average and relative standard deviation were calculated before zeros were replaced with 0.5×LOD.

[c] RSD = relative standard deviation

[d] LOD = limit of detection.

[e] ppt = parts-per-trillion by volume $(10^{-12})$

[f] N/A = data not available

[g] calculated using 3 × standard deviation at ambient background levels

To assess which components  impact  secondary product formation, a second PCA

was performed which included variables mainly formed through atmospheric chemical processes and whose concentrations more strongly depend on air mass chemical age than those variables selected initially. In this PCA, odd oxygen ($O_x$ = $O_3$ + $NO_2$), submicron aerosol $SO_4^{2-}{}_{(p)}$, $NO_3^-{}_{(p)}$, $NH_4^+{}_{(p)}$, a second, more-oxidized OOA factor (MO-OOA), and $PM_1$ volume were included, increasing the total number of variables to 28 (Table 4). Furthermore, since oxidation of IVOCs leads to formation of SOA (Robinson et al., 2007; Lee et al., 2018), and the photochemical conversion of IVOC to SOA may adversely affect the

PCA, a PCA without secondary and aerosol variables is presented in the S.I. (Table S-10).

[revised manuscript text omitted]

Component 5 correlates strongly with LO-OOA (r = 0.72), which is likely generated in part by photochemical aging of IVOCs.  A back-of-the-envelope calculation using a $k_{OH}$ of $1.8 \times 10^{-11}$ $cm^3$

$molecule^{-1}$ $s^{-1}$ based on that used diesel exhaust IVOCs (Zhao et al., 2014) and an estimated mid-day OH

concentration of $7 \times 10^6$ molecules $cm^{-3}$ (Liggio et al., 2016) gives a first-order lifetime of 130 min with respect to IVOC oxidation by OH during daytime. The photochemical age, estimated using relative concentrations of 124-TMB and n-decane and the method described by Borbon et al. (2013), during daytime was 1.0±0.4 hr; assuming similar photochemical ages, we estimate that between 25% and 50%

of the emitted IVOC is (potentially) oxidized during daytime (see S. I.).. This oxidation will contribute SOA

growth (Kroll et al., 2011). Hence, we expect some formation and growth of organic aerosol associated with component 5.

[revised manuscript text omitted]

Polissar, A. V., Hopke, P. K., Paatero, P., Malm, W. C., and Sisler, J. F.: Atmospheric aerosol over Alaska:

2. Elemental composition and sources, J. Geophys. Res.-Atmos., 103, 19045-19057, doi:10.1029/98JD01212, 1998.

Quagraine, E. K., Headley, J. V., and Peterson, H. G.: Is biodegradation of bitumen a source of recalcitrant naphthenic acid mixtures in oil sands tailing pond waters?, J. Environ. Sci. Health Part A-Toxic/Hazard.

Subst. Environ. Eng., 40, 671-684, 10.1081/ese-200046637, 2005.

Robinson, A. L., Donahue, N. M., Shrivastava, M. K., Weitkamp, E. A., Sage, A. M., Grieshop, A. P., Lane,

T. E., Pierce, J. R., and Pandis, S. N.: Rethinking organic aerosols: Semivolatile emissions and photochemical aging, Science, 315, 1259-1262, 2007.

[revised manuscript text omitted]

observations, J. Geophys. Res.-Atmos., 121, 1922-1934, 10.1002/2015jd024203, 2016.

Zhao, W., Hopke, P. K., and Karl, T.: Source Identification of Volatile Organic Compounds in Houston,

Texas, Environm. Sci. Technol., 38, 1338-1347, 10.1021/es034999c, 2004.

Zhao, Y. L., Nguyen, N. T., Presto, A. A., Hennigan, C. J., May, A. A., and Robinson, A. L.: Intermediate

Volatility Organic Compound Emissions from On-Road Diesel Vehicles: Chemical Composition,

Emission Factors, and Estimated Secondary Organic Aerosol Production, Environm. Sci. Technol., 49,

11516-11526, 10.1021/acs.est.3b02841, 2015.

**Supplementary information for**

**Principal component analysis of summertime ground site measurements in the Athabasca oil sands**

**with a focus on analytically unresolved intermediate volatility organic compounds**

Travis W. Tokarek[1], Charles A. Odame-Ankrah[1], Jennifer A. Huo[1], Robert McLaren[2], Alex K. Y. Lee[3, 4],

Max G. Adam[4], Megan D. Willis[5], Jonathan P. D. Abbatt[5], Cristian Mihele[6], Andrea Darlington[6],

Richard L. Mittermeier[6], Kevin Strawbridge[6], Katherine L. Hayden[6], Jason S. Olfert[7], Elijah. G. Schnitzler[8],

Duncan K. Brownsey[1], Faisal V. Assad[1], Gregory R. Wentworth[5, a], Alex G. Tevlin[5], Douglas E. J. Worthy[6],

Shao-Meng Li[6], John Liggio[6], Jeffrey R. Brook[6], and Hans D. Osthoff[1*]

[1] Department of Chemistry, University of Calgary, Calgary, Alberta, T2N 1N4, Canada

[2] Centre for Atmospheric Chemistry, York University, Toronto, Ontario M3J 1P3, Canada

[3] Department of Civil and Environmental Engineering, National University of Singapore, Singapore

117576, Singapore.

[4] NUS Environmental Research Institute, National University of Singapore, Singapore

[5] Department of Chemistry, University of Toronto, Toronto, Ontario, M5S 3H6, Canada

[6] Air Quality Research Division, Environment and Climate Change Canada, Toronto, Ontario, M3H 5T4,

Canada

[7] Department of Mechanical Engineering, University of Alberta, Edmonton, Alberta, T6G 1H9, Canada

[8] Department of Chemistry, University of Alberta, Edmonton, Alberta, T6G 2G2, Canada

[a] Now at: Environmental Monitoring and Science Division, Alberta Environment and Parks, Edmonton,

Alberta, T5J 5C6, Canada

* Corresponding author

**Descriptions of instrumentation used**

A Griffin 450 gas chromatograph equipped with a cylindrical ion trap mass spectrometer and electron impact ionization (GC-ITMS) was used to quantify selected VOCs including o-xylene, decane, undecane,

1,2,3- and 1,2,4-trimethylbenzene (TMB), and several monoterpenes (i.e., α-pinene, ß-pinene and limonene). The GC-ITMS primary responsibility was the quantification of monoterpenes. The remaining

VOCs quantified were chosen because (a) they sufficiently resolved on the analytical column, and (b)

response factors could be determined, either because the compounds of interest were part of the VOC

standard used in the field (such as the aromatics o-xylene, 1,2,3- and 1,2,4-TMB, see below) or relative response factors were determined post-campaign. Operation, calibration and performance of this instrument have been described elsewhere (Tokarek et al., 2017; Liggio et al., 2016). Briefly, the GC was operated with 30 m (length) × 0.25 mm (inner diameter) × 0.25 μm (film thickness) DB-5MS analytical column with helium carrier gas. The GC-ITMS sampled from a 3.6 m long stainless-steel inlet with an o.d.

of 0.635 cm from a height of 5 m above ground. A 1 m long section of the inlet was heated to 110 °C and optimized to remove interference due to $O_3$ while avoiding decomposition of alkenes (Tokarek et al.,

2017). The GC oven was programmed as follows: hold at 40 °C for 3.00 min, heat at 1.5 °C min$^{-1}$ to 70° C

(reached at 23.00 min), heat at 5° C min$^{-1}$ to 200 °C (reached at 49.00 min) and hold for 4 min (total

53.00 min). This was followed by a 5 min recovery time to allow the oven and pre-concentration trap to cool back to 40 °C. The ion trap mass spectrometer was set to an *m/z* range of 50-425. After data reduction, the GC-ITMS generated 10-minute average concentrations of each VOC quantified every hour.

During the campaign, the GC-ITMS was calibrated in the field using an IONICON VOC standard (Table S-

1) containing (in addition to VOCs that the GC-ITMS did not detect) α-pinene and o-xylene at mixing ratios of ~ 1 ppmv and an uncertainty of 5% and 6%, respectively. A commercial calibrator assembly (IONICON, GCU Standard) was used to deliver diluted calibration mixtures. The instrument responses to the VOC standards were linear ($R^2$ > 0.99). The GC-ITMS was calibrated for other VOCs offline relative to

α-pinene. In the field, there was no noticeable carry-over (i.e., memory effects) of IVOCs, which was occasionally evaluated by flooding the inlet with purified, VOC-free air.

Matrices of ions plotted against retention times for the total ion chromatograms (shown in Figure 2 in the main manuscript) are shown in Fig. S-1. In both cases, the greatest intensity is with masses are associated with alkanes (i.e., $m/z$ 55, 57, 67, 69, etc.).

[Figure]

Figure S-1. Scatter of ions as a function of retention time for the total ion chromatograms shown in

Figure 2 of the main manuscript. Darker pixels represent a higher intensity than lighter pixels.

[revised manuscript text omitted]

**PCA without aerosol variables**

A sensitivity test was conducted by which all aerosol species were removed as variables. The results of this sensitivity test are shown in Table S-10 and are presented as a 9-component solution, since the dust component associated with $PM_{10-1}$ (component 7 in Table 5) cannot be generated when its main variable is removed.

The pattern in Table S-10 resembles that in Table 5 of the main manuscript, in that the same nine components emerged in both solutions with similar magnitude r values for each of the variables, including the IVOC signature. The only difference is that components 2 and 3 as well as 5 and 6 have traded places (i.e., the relative magnitudes of their eigenvalues, which were similar in Table 5, have switched), which is inconsequential. Furthermore, the correlation coefficients in Table S-10 are of similar magnitude (i.e., within ±0.1) as those in Table 5, which suggests that IVOC to SOA conversion does not adversely affect the PCA, likely because of the proximity of the receptor site to sources.

[revised manuscript text omitted]

**Estimate of photochemical age**

Photochemical age was calculated using the method outlined by Borbon et al. (2013), but substituting n-decane for benzene since the latter was not quantified. The photochemical age of an air mass, Δt was calculated from the observed concentrations of 124-trimethylbenzene (124TMB) and n-decane using:

$$\Delta t = \frac{1}{[OH] \times (k_{124TMB} - k_{decane})} \times \left[ \ln\left(\frac{[124TMB]}{[decane]}\right)_{t=0} - \ln\left(\frac{[124TMB]}{[decane]}\right) \right] \tag{S-1}$$

where $k_{124TMB} = 3.25 \times 10^{-11}$ cm$^3$ molecule$^{-1}$ s$^{-1}$ and $k_{decane} = 1.10 \times 10^{-11}$ cm$^3$ molecule$^{-1}$ s$^{-1}$ are rate coefficients for reaction of OH with 124-TMB and n-decane (at 298 K), respectively, whose values were taken from Seinfeld and Pandis (2006). The ratio of [124TMB] to [decane] at the point of emission (time t = 0) was estimated from a plot of [124TMB] to [n-decane] (Figure S-13, left-hand side) and a straight-line fit to the nocturnal data (assumed to be unaffected by oxidation and shown in blue color). The slope of this line was 1.15±0.07 ($r^2$ = 0.84). Daytime data (color-coded by solar zenith angle, SZA) exhibit lower ratios of [124TMB]/[decane] as a result of the faster oxidation of 124TMB by OH.

Shown in Figure S-13 on the right-hand side is a plot of the photochemical age, calculated using equation (S-1) and an assumed [OH] of $7 \times 10^6$ molecules cm$^{-3}$ taken from Liggio et al. (2016), as a function of SZA (filtered for peak OH of 11:00 and 16:00 local time). The error bars indicate ages calculated using emission factors of 1.08 and 1.22, respectively. The average (±1 standard deviation) photochemical age is 1.0±0.4 hr. This photochemical age applies mainly to component 1; we assume that the photochemical ages of sources associated with other components were similar.

[Figure]

**Figure S-13. (A)** Plot of 124TMB mixing ratios against mixing ratios of n-decane, color-coded by solar zenith angle. The blue data points were collected at night. **(B)** Photochemical age calculating using equation S-1 plotted as a function of solar zenith angle.

In their analysis of IVOC photochemical aging, Zhao et al. (2014) estimated an average $k_{OH}$ for diesel- exhaust IVOCs of $1.8 \times 10^{-11}$ cm$^3$ molecule$^{-1}$ s$^{-1}$ (though their estimated rate coefficients varied and increased slightly with volatility bin between about 1 and 3 $\times 10^{-11}$ cm$^3$ molecule$^{-1}$ s$^{-1}$). From this, we calculate a pseudo first-order lifetime of 130 min (2.17 hr) with respect to IVOC oxidation by OH during daytime. Using a photochemical age of 1.0±0.4 hr, we calculate that between 25% and 50% of the emitted IVOC is (potentially) oxidized during daytime. Photochemical aging will affect data collected during the daytime hours (from ~11 am to ~4 pm) or ~25% of the data (56 out of 218 data points) used in the PCA and likely resulted in partial conversion of IVOCs to SOA.